# pH-responsive high stability polymeric nanoparticles for targeted delivery of anticancer therapeutics

L. Palanikumar[1], Sumaya Al-Hosani[1], Mona Kalmouni[1], Vanessa P. Nguyen [2], Liaqat Ali[3], Renu Pasricha[3], Francisco N. Barrera [2] & Mazin Magzoub [1✉]

The practical application of nanoparticles (NPs) as chemotherapeutic drug delivery systems is often hampered by issues such as poor circulation stability and targeting inefficiency. Here, we have utilized a simple approach to prepare biocompatible and biodegradable pH-responsive hybrid NPs that overcome these issues. The NPs consist of a drug-loaded polylactic-co-glycolic acid (PLGA) core covalently 'wrapped' with a crosslinked bovine serum albumin (BSA) shell designed to minimize interactions with serum proteins and macrophages that inhibit target recognition. The shell is functionalized with the acidity-triggered rational membrane (ATRAM) peptide to facilitate internalization specifically into cancer cells within the acidic tumor microenvironment. Following uptake, the unique intracellular conditions of cancer cells degrade the NPs, thereby releasing the chemotherapeutic cargo. The drug-loaded NPs showed potent anticancer activity in vitro and in vivo while exhibiting no toxicity to healthy tissue. Our results demonstrate that the ATRAM-BSA-PLGA NPs are a promising targeted cancer drug delivery platform.

---

[1] Biology Program, Division of Science, New York University Abu Dhabi, Abu Dhabi, UAE. [2] Department of Biochemistry & Cellular and Molecular Biology, University of Tennessee at Knoxville, Knoxville, TN, USA. [3] Core Technology Platforms, New York University Abu Dhabi, Abu Dhabi, UAE. ✉email: mazin.magzoub@nyu.edu

onventional chemotherapy drugs work primarily by interfering with DNA replication and mitosis in order to induce apoptosis in rapidly dividing cancer cells, thereby minimizing tumor growth and inhibiting metastatic progression[1,2]. However, conventional chemotherapeutics suffer from a number of limitations, including poor solubility and short in vivo circulation half-life. The majority of these drugs, which are administered systemically, are non-selective and therefore also target healthy cells[3]. Moreover, during treatment cancer cells often develop multidrug resistance (MDR) phenotypes, such as enhanced drug efflux, activation of nuclear DNA repair mechanisms and altered drug metabolism, which severely limit the clinical efficacy of chemotherapeutics[4,5]. Consequently, it becomes necessary to administer high doses of chemotherapeutics to ensure that a sufficient amount reaches the tumor to cause the desired effect in cancer cells. Unfortunately, the high chemotherapeutic doses, along with the various combinations of drugs used, ultimately lead to damage to healthy tissue, resulting in a range of side-effects that include nausea, hair loss, fatigue, decreased resistance to infection, infertility, and organ damage.

The issues associated with conventional chemotherapy have prompted the development of a wide-range of nanocarriers for safer and more effective delivery of chemotherapeutics. These include inorganic (e.g., silica[6] and silver[7]) and organic (e.g., polymers[8] and cellular membranes[9]) nanoparticles (NPs). Among these NPs, block copolymer micelles, which are composed of a stealth hydrophilic shell surrounding a hydrophobic core encapsulating a variety of water insoluble drugs, have garnered considerable interest[10]. Yet, to date, very few of these micellar systems have reached the clinical trial stage[11,12]. One of the few successes of this strategy so far is poly(ethylene glycol)-block-poly (D,L-lactide acid) (PEG-PDLLA) micelles loaded with paclitaxel (Genexol-PM), which is currently undergoing clinical trials[13]. Although they have been shown to reduce some of the adverse effects associated with chemotherapeutics, block copolymer micelles do not yield the expected enhancement in therapeutic outcome relative to free drugs[14]. This is attributed to loss of micellar integrity due to interactions with serum proteins[15]. The most common method for improving micelle stability and drug retention is to tune the polymer matrix miscibility and augment the loaded drug's hydrophobicity using various physical and chemical strategies[16]. However, these complex modifications of the polymeric structure often result in aggregation of the loaded drugs during the self-assembly process, which leads to attenuation of the therapeutic efficacy of the system[17,18].

Most NPs used for delivery of chemotherapeutics 'passively' target tumors[16,19]. The NPs are administered systemically and utilize the so-called enhanced permeability and retention (EPR) effect, which is a product of the leaky vasculature and impaired lymphatic drainage of tumors, to accumulate within tumor tissue[20]. However, even in cases of high EPR, <1% of passively targeted NPs accumulate in tumors[11]. An alternative strategy is to actively target NPs to cancer cells. This is achieved by functionalizing NPs with targeting ligands that bind to receptors or other membrane proteins overexpressed or clustered on surfaces of cancer cells[19,21]. Although this approach is expected to promote accumulation within tumors, so far no actively targeted NPs for cancer drug delivery have progressed beyond the clinical trial stage[19]. The lack of success of the active targeting strategy may be attributed, in part, to the formation of a serum protein corona on the surface of the NPs during in vivo blood circulation[22,23]. The protein corona not only hinders the ability of actively targeted NPs to interact with cancer cell surface receptors, but the adsorbed serum proteins are also associated with immune recognition and rapid blood clearance, all of which leads to poor tumor accumulation[17,23].

Once NPs reach the target tumor tissue, a barrier to successful delivery of their chemotherapeutic cargo is uptake into cancer cells[16,19]. Targeting ligands on NPs can promote their internalization in cancer cells by triggering receptor-mediated endocytosis[21]. However, most endocytosed NPs inevitably become entrapped in endocytic compartments where the ultimate fate of the loaded cargo is either exocytosis via recycling endosomes, or degradation in lysosomes. For instance, studies on NP-mediated delivery of siRNA revealed that endosomal escape efficiency was extremely low (1–2%)[24], with most of the internalized siRNA (~70%) undergoing exocytosis[25]. Thus, escape from endocytic compartments and delivery of the therapeutic payload to the appropriate intracellular site remains a major challenge for NP-based drug delivery strategies.

To address these challenges, we have designed hybrid NPs that consist of a polylactic-co-glycolic acid (PLGA) core 'wrapped' with a cross-linked bovine serum albumin (BSA) shell that is functionalized with the acidity-triggered rational membrane (ATRAM) peptide[26–28]. The pH-responsive hybrid ATRAM-BSA-PLGA NPs are a promising cancer drug delivery platform that combines high stability with effective tumor targeting and triggered release of chemotherapeutic agents in cancer cells.

## Results

### Synthesis and characterization of doxorubicin-triphenylphosphonium (Dox-TPP).
Mitochondria-targeting doxorubicin-triphenylphosphine (Dox-TPP), which is designed to overcome drug resistance in cancer cells (see Supplementary Note 1), was synthesized using published procedures (Supplementary Fig. 1a)[29]. The structure of the Dox-TPP molecule was confirmed by LC-MS (Supplementary Fig. 1b) and $^1$H NMR (Supplementary Fig. 2). The measured zeta potentials of the Dox and Dox-TPP solutions (40 µg mL$^{-1}$) were +3.35 mV and +22.18 mV, respectively (Supplementary Fig. 1c), which are consistent with reported values[29]. Cytotoxicity of Dox-TPP was evaluated using the CellTiter 96 AQueous One Solution (MTS) assay, which measures reduction of the tetrazolium compound MTS by mitochondrial reductases in living cells. Dox-TPP was significantly more toxic to human breast cancer MCF-7 cells compared to unmodified Dox (29 ± 4% vs 15 ± 2% cell viability for Dox vs Dox-TPP, respectively, at 2.0 µg mL$^{-1}$; Supplementary Fig. 3), confirming that Dox-TPP is a potent anticancer drug.

### Preparation and characterization of BSA-PLGA NPs.
Interactions with serum proteins are reported to cause substantial leakage of loaded drugs from self-assembled polymeric structures[17,18]. Moreover, the formation of a serum protein corona on receptor-targeting NPs during in vivo circulation adversely affects target recognition and results in nonspecific distribution[17,30]. To overcome these issues, we have designed biocompatible and biodegradable hybrid NPs that consist of a hydrophobic polylactic-co-glycolic acid (PLGA)[31] core 'wrapped' with a hydrophilic shell composed of BSA. The PLGA core can be readily loaded with a wide-range of hydrophobic drugs[32]. For the shell, BSA was chosen given its ready availability, low cost, ease of purification, high water solubility and long in vivo circulation half-life[33].

Dox-TPP loaded BSA-grafted PLGA NPs (loading capacity 1.8 wt%; Supplementary Table 1) were prepared by a simple two-step method (Supplementary Fig. 4). Glutaraldehyde (GA) was then used to covalently crosslink 72% of reactive lysine residues in BSA (Supplementary Fig. 5a), achieving a consistent polydispersity index of 0.02[34]. Such a high crosslinking density confers encapsulation stability[35], yielding a nanostructure that effectively avoids uncontrolled leakage of loaded cargo molecules.

Transmission electron microscopy (TEM) and scanning transmission electron microscopy (STEM) images clearly showed the BSA-modified PLGA NPs as a core-shell structure (Fig. 1b, d). BSA was enriched over the PLGA surface, forming a corona with a thickness of ~10 nm and increasing the hydrodynamic diameter of the NPs from 105 to 130 nm (Fig. 1c, Supplementary Table 2). The zeta potential of the PLGA NPs changed from −28 to −16 mV after conjugation with BSA (Fig. 1e, Supplementary Table 2). Fourier-transform infrared spectroscopy (FTIR) characterization confirmed the successful incorporation of BSA and PLGA molecules into the BSA-PLGA NPs (Supplementary Fig. 5b). Analysis using the bicinchoninic acid (BCA) protein assay revealed that the concentration of BSA conjugated to PLGA NPs was $72.9 \pm 2.7\ \mu g\ mg^{-1}$; in other words, BSA contributes ~7.3 wt% of the NPs. This is close to the reported 8 wt% of PEGylation over the surface of NPs required for a backfilling effect that reduces the formation of a serum protein corona[30]. However, such protective grafting may also strongly influence the surface charge and colloidal stability of the NPs[36].

The BSA-PLGA NPs showed good colloidal stability in 10 mM phosphate buffer solution (pH 7.4) and cell culture medium containing 10% fetal bovine serum (FBS) over 72 h (Fig. 1f, g). Moreover, the BSA-PLGA NPs maintained a similar surface charge in cell culture medium containing 10% FBS (Supplementary Fig. 5c). Next, we monitored the size of the BSA-PLGA NPs under conditions of high serum content and high ionic strength, both of which are reported to affect the colloidal stability of nanocarriers[37]. Increasing the serum content to 50% resulted in a modest increase of ~30 nm in the hydrodynamic diameter of the BSA-PLGA NPs (Supplementary Fig. 6a) relative to that in 10% FBS, suggesting that the NPs are able to maintain a reasonable size in circulation that will allow them to localize to tumors and then readily internalize into cancer cells to deliver their chemotherapeutic cargo. Likewise, increasing the ionic strength to the levels found in cells and higher did not noticeably alter the size of the BSA-PLGA NPs (Supplementary Fig. 6b), indicating that the NPs will remain stable and retain their chemotherapeutic cargo in cellular milieu until exposure to an appropriate stimulus (e.g., low pH). Taken together, these results show that the crosslinked BSA shell confers a high degree of stability on the NPs. This is likely due to thermodynamically favorable interactions between specific domains of BSA with the surface of the PLGA NPs, which are reinforced by crosslinking of the shell, leading to colloidal stabilization and acquisition of stealth properties (i.e., prevention of serum protein adsorption)[23,38,39].

Serum protein adsorption to the NPs was probed further using quantitative proteomics (Fig. 2). PLGA and BSA-PLGA NPs were incubated in cell culture medium containing 10% FBS for 72 h, and the serum proteins that adsorbed to the surface of the NPs were isolated by centrifugation and quantified using liquid chromatography tandem mass spectrometry (LC-MS/MS) with label free-quantification (LFQ). Analysis of the 26 most abundant serum proteins, selected after filtering the unavoidable mass spectrometric contaminants, revealed substantially higher adsorption to the PLGA NPs compared to the BSA-PLGA NPs. These results confirm that the crosslinked BSA shell successfully prevents the formation of a serum protein corona the surface of the NPs.

Finally, far-ultraviolet circular dichroism (far-UV CD) was used to determine the secondary structure of BSA conjugated to PLGA (Supplementary Fig. 5d). The measured molar ellipticities at 208 and 222 nm of the BSA-PLGA NPs, with and without crosslinking, were comparable to those reported for BSA alone[40]. This suggests that coupling of the BSA to the NPs, followed by crosslinking with GA, does not affect the properties of BSA.

**Encapsulation stability of BSA-PLGA NPs.** Förster resonance energy transfer (FRET) experiments were used to assess the encapsulation stability of PLGA, non-crosslinked BSA-PLGA and crosslinked BSA-PLGA NPs (Supplementary Fig. 7). The NPs were co-loaded with the FRET donor/acceptor pair DiO/DiI (loading capacity 1 wt%) (Supplementary Fig. 7a). In the case of PLGA and non-crosslinked BSA-PLGA NPs, the acceptor dye, DiI, fluorescence intensity at 565 nm remained constant in 10 mM phosphate buffer (Supplementary Fig. 7b, c), but decreased appreciably in cell culture medium containing FBS (Supplementary Fig. 7e, f, h, i). On the other hand, for crosslinked BSA-PLGA NPs we observed no time-dependent decrease in DiI fluorescence intensity in either 10 mM phosphate buffer (Supplementary Fig. 7d) or cell culture medium containing 10 or 20% FBS (Supplementary Fig. 7 g, j). The calculated time-dependent FRET efficiencies confirmed release of the FRET pair from PLGA and non-crosslinked BSA-PLGA NPs when incubated in serum-containing solutions due to destabilization of the carrier[18], whereas crosslinked BSA-PLGA NPs are stable under the same conditions (Supplementary Fig. 7k).

Cellular FRET results (Supplementary Fig. 8) were consistent with the solution experiments. Incubation of MCF-7 cells with the DiO/DiI co-loaded PLGA NPs resulted in intracellular accumulation of the fluorophores over 4 h (Supplementary Fig. 8a). However, no FRET signal was detectable, which indicates degradation of the NPs and release of the FRET pair. In contrast, treatment of with DiO/DiI co-loaded crosslinked BSA-PLGA NPs for 4 h led to a strong FRET signal from subcellular compartments, indicating close proximity of DiO and DiI within intact NPs (Supplementary Fig. 8b). As a control, MCF-7 cells treated with crosslinked BSA-PLGA NPs loaded with only the acceptor dye, DiI, did not show any FRET signal (Supplementary Fig. 8c). These results demonstrate the high encapsulation stability of the crosslinked BSA-PLGA NPs, which are able to retain loaded dyes even in the presence of high concentrations of serum proteins or in a cellular environment.

**Drug release profile of BSA-PLGA NPs.** In the absence of a stimulus, naked PLGA NPs showed a burst of Dox-TPP release within the first hour (Fig. 3a), which is attributed to dissociation of drug adhered to the surface of the NPs. The initial burst was followed by a continuous release of drug until 24 h, due to a combination of drug diffusion through pores within the NP structure and PLGA polymer erosion and degradation[41]. For BSA-PLGA NPs without shell crosslinking, there was a small stimulus-free release of Dox-TPP over 24 h, but the release rate was much slower than that of PLGA NPs ($24 \pm 5$ vs $52 \pm 2\%$ of Dox-TPP released from BSA-PLGA vs PLGA NPs, respectively) (Fig. 3a). This indicates that the presence of the BSA shell inhibits, but does not completely abolish, drug release from the PLGA core. Crosslinking of the BSA shell resulted in highly stable NPs that did not exhibit any stimulus-free leakage of loaded drug over 24 h (Fig. 3a). Hence, crosslinked BSA-PLGA NPs show very stable encapsulation, which is critical for preventing premature release and ensuring that the loaded drugs reach the target diseased cells[6].

Since endocytosis is a common cellular uptake route for nanocarriers[42], during their endocytic entry into cells NPs will be exposed to increasingly acidic environments as they are trafficked from weakly acidic early/maturing endosomes (pH 6.0–5.5) to more acidic late endosomes/lysosomes (pH 5.0–4.5)[43,44]. Therefore, we probed the effects of an acidic environment on the release of Dox-TPP from crosslinked BSA-PLGA NPs (Fig. 3b). At pH 6.5, negligible release of Dox-TPP was observed, indicating that the crosslinked BSA-PLGA NPs will effectively retain their

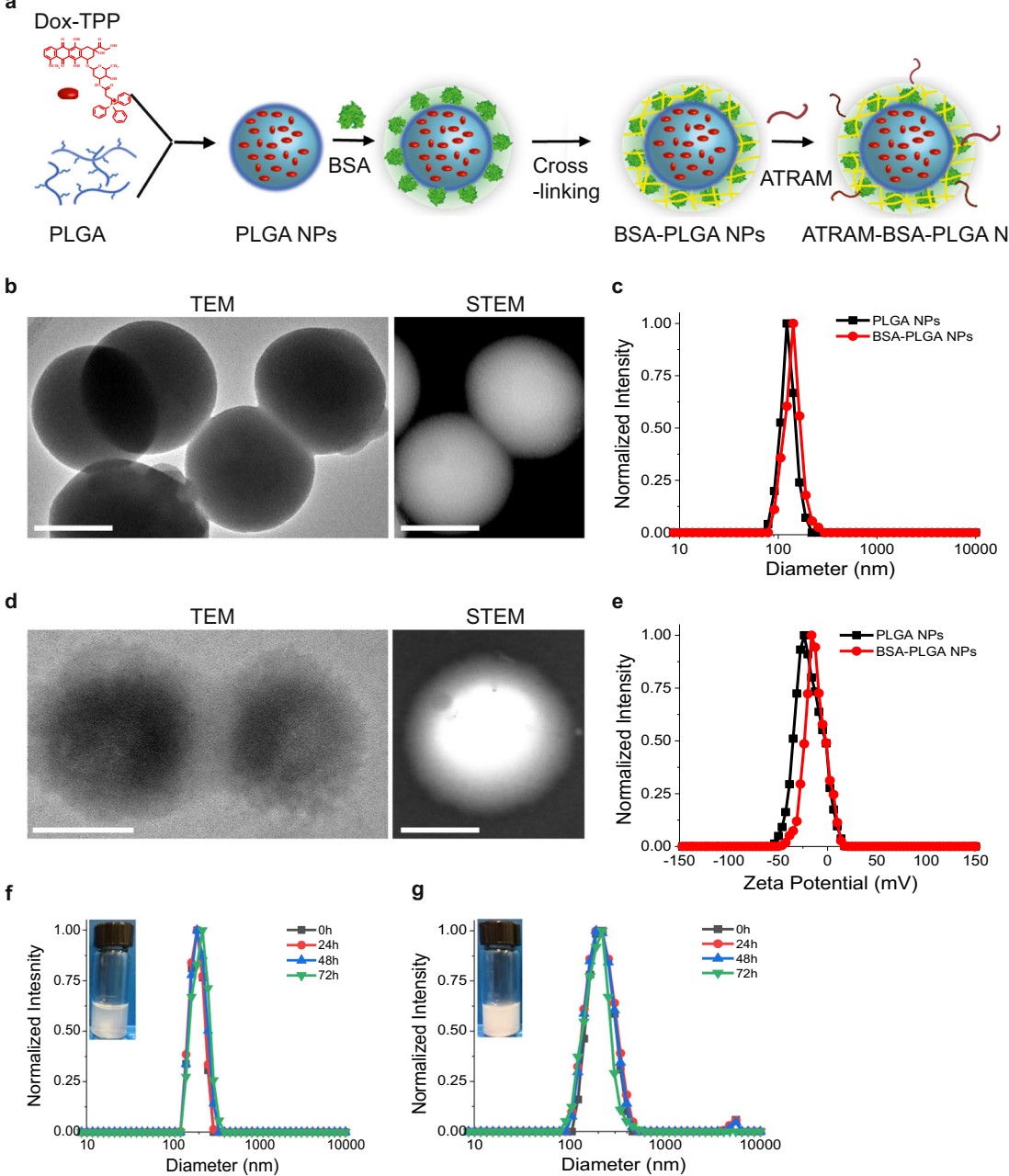

**Fig. 1 Characterization of the BSA-PLGA nanoparticles (NPs). a** Schematic representation of preparation of Dox-TPP loaded ATRAM-conjugated BSA-PLGA nanoparticles. **b**, **d** Transmission electron microscopy (TEM) and scanning transmission electron microscopy (STEM) images of PLGA (**b**) and BSA-PLGA (**d**) NPs. Scale bar = 50 nm. **c**, **e** Size analysis (**c**) and zeta potential measurements (**e**) for PLGA and BSA-PLGA NPs in 10 mM phosphate buffer at pH 7.4. **f**, **g** Colloidal stability analysis for BSA-PLGA NPs in 10 mM phosphate buffer at pH 7.4 (**f**) and cell culture medium containing 10% fetal bovine serum (FBS) (**g**). Inset: images of higher concentrations of nanoparticles dispersed in buffer (**f**) and serum (**g**) solutions to show the colloidal dispersity after 72 h.

chemotherapeutic cargo within the mildly acidic microenvironment of malignant solid tumors (pH 6.5–6.9)[45]. Lowering the pH to within the range reported for early endosomes (pH 5.8) triggered release of the Dox-TPP cargo (45 ± 5% release at 24 h). The release under acidic conditions occurs due to degradation of the PLGA core (as a result of hydrolysis of the ester bonds in the polymer chains)[46]. Additionally, as the pH decreases, dissociation of the increasingly hydrophilic Dox-TPP from the hydrophobic PLGA core likely contributes to the acid-triggered release of the drug[45,47]. Lowering the pH even further to that of late endosomes/lysosomes (pH 5.0), led to much higher release of Dox-TPP (79 ± 4% at 24 h)

(Fig. 3b). As a control, we monitored release of the hydrophobic dye rhodamine B from crosslinked BSA-PLGA NPs (Fig. 3c). Negligible release of rhodamine B was observed at pH 7.4 or 5.8. However, at pH 4.0, where rhodamine B becomes cationic due to protonation of its carboxylic acid group (p$K_a$ ~4.2)[48], a substantial amount of the dye is released from the NPs (44 ± 2% release at 24 h). This supports the notion that protonation of the cargo contributes to its acid-triggered release. Thus, the acidic microenvironment of endocytic compartments is expected to facilitate the efficient release of the Dox-TPP cargo following internalization of the crosslinked BSA-PLGA NPs into cancer cells.

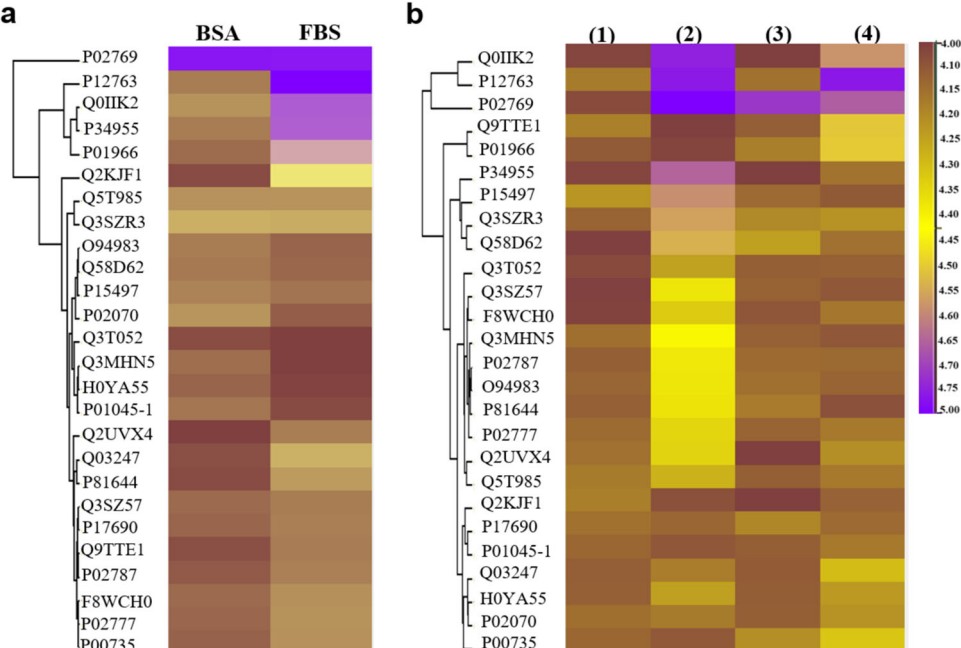

**Fig. 2 Quantitative proteomic analysis of serum protein adsorption to the NPs. a** Heat map representation of identified serum proteins in the control BSA (2 mg mL$^{-1}$ solution) and FBS (10% in cell culture medium) samples. **b** Heat map representation of serum proteins adsorbed to the NPs under the following conditions: PLGA (**1**) and BSA-PLGA NPs (**3**) incubated in 10 mM phosphate buffer for 72 h; PLGA (**2**) and BSA-PLGA NPs (**4**) incubated in cell culture medium containing 10% FBS for 72 h. The digests for both the control and NP samples were analyzed by liquid chromatography tandem mass spectrometry (LC-MS/MS) and protein abundance was determined using the label-free quantification (LFQ) intensities. As shown in the color scale bar, the purple and yellow (gold) colors indicate high and low LFQ intensities (log2 (LFQ))[85], respectively, while dark brown indicates that the protein concentration is below the detection limit. The proteins corresponding to the UniProt Knowledgebase (UniProtKB) accession numbers shown in the figure are given in Supplementary Table 3.

The levels of cytosolic antioxidant glutathione (GSH) are several fold higher in tumors cells compared to healthy cells (see Supplementary Note 2)[41,45]. Therefore, we tested whether GSH represents a convenient stimulus for destabilizing the NPs and releasing their cargo within cancer cells. Exposure of the crosslinked BSA-PLGA NPs to 0.5 mM GSH, which is within the range reported for healthy mammalian cells[49], triggered the release of a modest amount of the Dox-TPP cargo (18 ± 3% release at 24 h) (Fig. 3d). However, addition of 10 mM GSH at pH, which is within the range reported for malignant cells[41,50], led to release of Dox-TPP from the NPs (47 ± 3% release at 24 h). As GSH does not adversely affect PLGA[51], this suggests that the target of GSH is the BSA shell, particularly since GSH is known to have a high affinity for BSA[52]. Although the mechanism of GSH-mediated disruption of the crosslinked BSA shell is not yet clear (see Supplementary Note 2), the results strongly suggest that the elevated levels of GSH within cancer cells will disrupt the BSA shell of the NPs and release the Dox-TPP cargo intracellularly.

**Conjugation of pH-responsive ATRAM peptide to BSA-PLGA NPs**. In order to specifically target tumor cells, the BSA-PLGA NPs were functionalized with the ATRAM peptide, which inserts into lipid bilayers under acidic conditions[26]. The measured membrane insertion p$K_a$ of ATRAM is 6.5 (see Supplementary Note 3)[26–28], making the peptide ideal for targeting cancer cells in the acidic microenvironment of solid tumors[45,53], which results from the high glycolytic rates of tumor cells[54]. Indeed, ATRAM was shown to efficiently target tumors in mice[55].

We previously established that ATRAM inserts into membranes via its C-terminus[27]. Therefore we conjugated ATRAM to the surface of the BSA-PLGA NPs by covalently coupling the remaining lysine residues of BSA in the NPs to the N-terminal amino group of the peptide via carbodiimide chemistry[56], and the attachment was evaluated by release of the urea byproduct at 232 nm (Supplementary Fig. 9)[57]. Dynamic light scattering measurements showed that conjugation to the ATRAM peptide did not noticeably alter the hydrodynamic size of the BSA-PLGA NPs (Fig. 4a; Supplementary Table 2). Conjugation to ATRAM was further confirmed by change in the zeta potential of the NPs at pH 7.4 from −16 mV (for crosslinked BSA-PLGA) to −5 mV (for ATRAM-BSA-PLGA) (Fig. 4b, c; Supplementary Table 2), which is within the range of zeta potential values reported for other highly stable NPs at physiological pH[23,45]. At pH 6.5, due to protonation of the glutamic acid residues that drives the pH-responsiveness of ATRAM, the zeta potential increased further to +10 mV, suggesting that the ATRAM-BSA-PLGA NPs would be efficiently internalized by cancer cells within the acidic tumor microenvironment[19,45].

Development of peptides for therapeutic purposes has been hampered by their short in vivo half-life due to degradation by proteases and fast renal clearance[58]. A common strategy for prolonging the half-life of peptides in vivo is to modify them in order to increase their affinity for long-lived serum albumin[59,60], the most abundant protein in human blood plasma (constituting ~60% of the total plasma protein pool)[61]. The effectiveness of this approach has been demonstrated by the FDA-approved diabetes therapeutics, insulin detemir, insulin degludec and liraglutide, which are fatty acid-modified peptides that have increased affinity for albumin, and consequently prolonged circulation half-life, compared to their unmodified counterparts[62].

Interestingly, ATRAM exhibits a longer circulation half-life compared to peptides of similar molecular weight[27]. Moreover, ATRAM was shown to accumulate within acidic tumor tissue in mice[55]. These results indicate that ATRAM is able to evade rapid

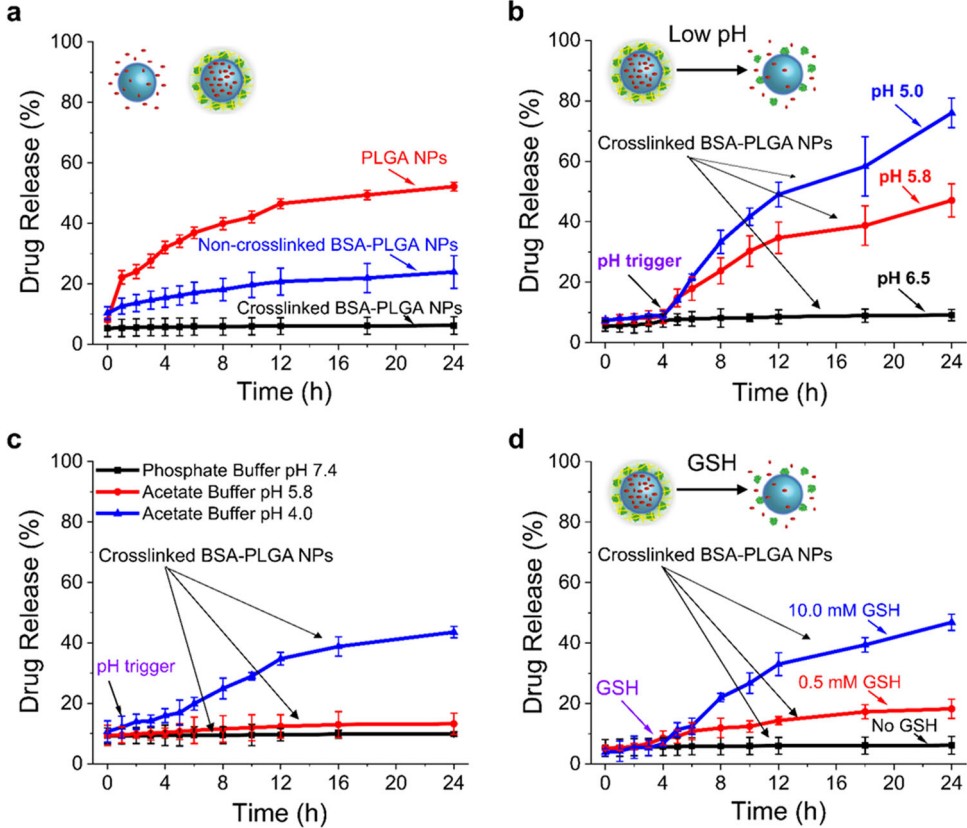

**Fig. 3 Drug release profiles of the BSA-PLGA NPs in the absence and presence of a stimulus. a** Stimulus-free release from Dox-TPP loaded PLGA or BSA-PLGA NPs (without and with shell crosslinking) in 10 mM phosphate buffer (pH 7.4) ($n = 3$). **b**, **c** pH-triggered release from Dox-TPP loaded (**b**) and rhodamine B-loaded (**c**) BSA-PLGA NPs with shell crosslinking ($n = 3$). **d** Glutathione (GSH) mediated Dox-TPP release from crosslinked BSA-PLGA NPs in 10 mM phosphate buffer (pH 7.4) ($n = 3$).

proteolysis and clearance in vivo. ATRAM was determined to bind reversibly to human serum albumin with an affinity of 1.1 ± 0.4 μM[27]. This suggests that once it is administered intravenously, ATRAM can use serum albumin as a temporary carrier in order to avoid rapid proteolysis, thereby explaining the observed long circulation time. Here, ATRAM was N-terminally conjugated to BSA on the surface of the NPs. Using fluorescence anisotropy studies of NBD-labeled ATRAM, we determined that ATRAM binds to BSA with an affinity of 294.8 ± 85.5 nM (Supplementary Fig. 10). We propose that the BSA to which ATRAM is anchored on the NP surface may confer similar protection to the peptide against rapid proteolysis, ensuring that ATRAM remains intact until the NPs reach the target tumor tissue.

**Cellular uptake of ATRAM-BSA-PLGA NPs.** The uptake of ATRAM-BSA-PLGA NPs in cancer cells was assessed using confocal fluorescence microscopy. Human breast cancer MCF-7 cells were incubated with Dox-TPP loaded ATRAM-BSA-PLGA NPs for 1 or 4 h at 37 °C (Fig. 4d). Considerably higher cellular internalization of the Dox-TPP cargo was observed at pH 6.5 compared to pH 7.4 at both 1 and 4 h incubation times. Moreover, the greater amount of intracellular Dox-TPP at acidic pH showed strong colocalization with mitochondria. Similarly, in human cervical cancer HeLa cells we observed a marked enhancement in cellular uptake, and mitochondrial localization, of the Dox-TPP cargo of ATRAM-BSA-PLGA NPs at acidic pH compared to physiological pH (Supplementary Fig. 11). Quantification of cellular uptake using flow cytometry (an example of the gating strategy is shown in Supplementary Fig. 12) confirmed the confocal microscopy results, with higher cellular

internalization of ATRAM-BSA-PLGA NPs in MCF-7 cells measured at acidic pH relative to physiological pH (amount of internalized NPs was >5- and >7-fold higher at pH 6.5 compared to pH 7.4 at 1 and 4 h incubation, respectively) (Fig. 4f, g). As a control, cellular uptake of Dox-TPP loaded BSA-PLGA NPs in MCF-7 cells was assessed using confocal fluorescence microscopy (Supplementary Fig. 13) and flow cytometry (Fig. 4e, g). In the absence of the ATRAM peptide, poor uptake of the NPs was observed at both pHs (7.4 and 6.5) and incubation times (1 and 4 h). Thus, the pH-dependent membrane insertion of ATRAM facilitates cellular internalization of the coupled NPs preferentially in cells that reside within a mildly acidic environment.

The cellular uptake of pH-responsive peptides such as ATRAM is proposed to involve pH-dependent insertion into the cell membrane—where the peptide undergoes a conformation change from a membrane surface adsorbed unstructured peptide at physiological pH to a transmembrane α-helix at acidic pH—followed by translocation of the peptide across the lipid bilayer[26,27]. However, there is evidence that, following pH-dependent insertion, a fraction of these peptides is also taken up by endocytosis, particularly when the peptide is coupled to a sizeable cargo, such as a liposome or a nanoparticle[27]. To determine whether cellular uptake of the ATRAM-BSA-PLGA-NPs occurs via an active, energy-dependent process such as endocytosis, or an energy-independent direct translocation mechanism, the uptake of Dox-TPP loaded NPs was assessed at 4 °C where all energy-dependent uptake processes are inhibited. Lowering the temperature to 4 °C markedly decreased the amount of intracellular Dox-TPP (Supplementary Fig. 14). Likewise, depleting the cellular ATP pool, by pre-incubation of the cells

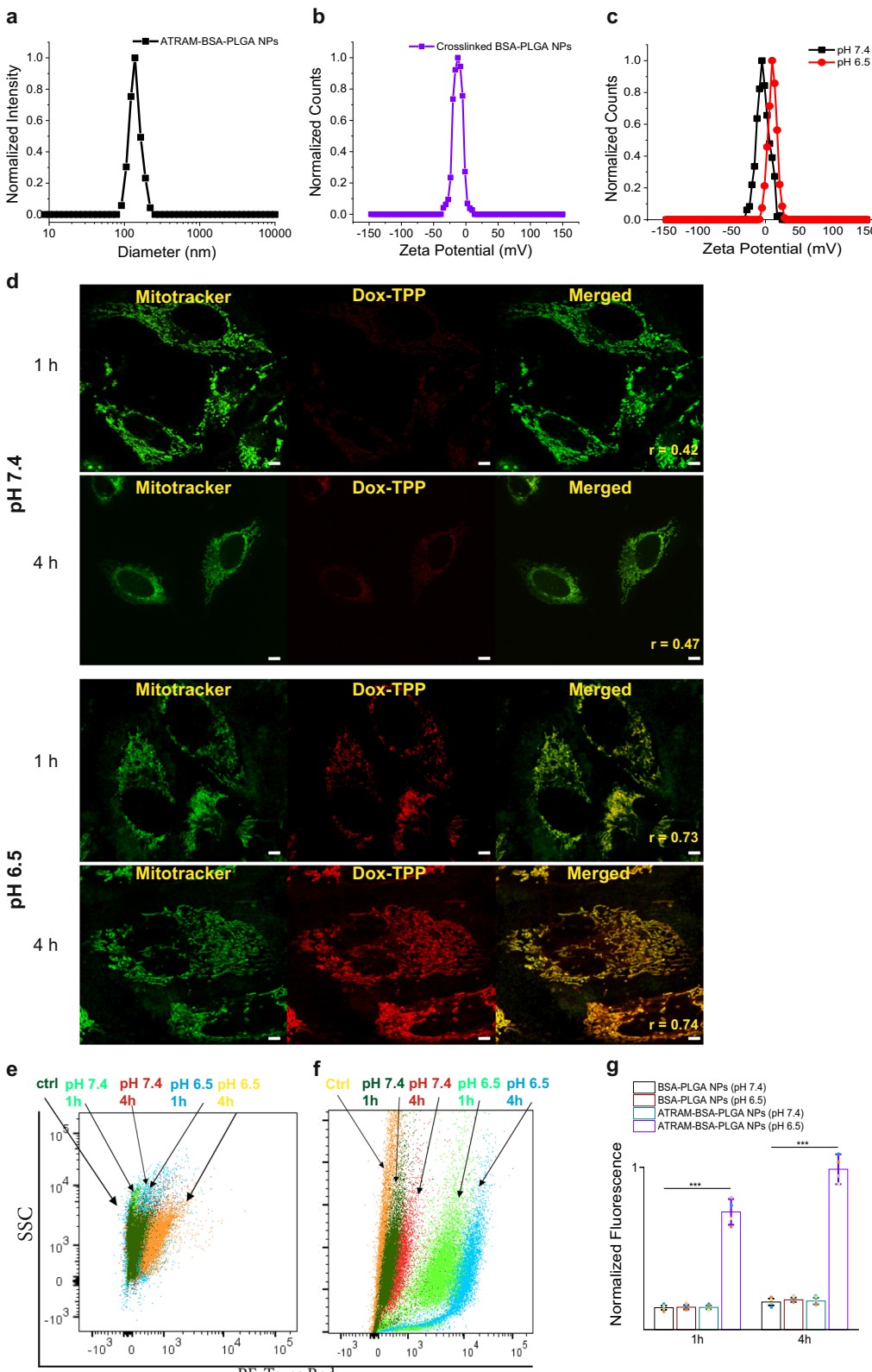

**Fig. 4 Cellular uptake of ATRAM-BSA-PLGA NPs is strongly pH-dependent. a** Size analysis for ATRAM-BSA-PLGA NPs in aqueous solution. **b**, **c** Zeta potential measurements of BSA-PLGA (**b**) and ATRAM-BSA-PLGA (**c**) NPs in aqueous solution. **d** Confocal laser scanning microscopy images of MCF-7 cells incubated for 1 or 4 h with Dox-TPP loaded ATRAM-BSA-PLGA NPs at physiological or acidic pH. Quantification of colocalization of the Dox-TPP cargo with mitochondria using Pearson's correlation coefficient (r). Scale bar = 10 μm. **e**–**g** Flow cytometry analysis of cellular uptake of Dox-TPP loaded BSA-PLGA and ATRAM-BSA-PLGA NPs in MCF-7 cells: plot of side scatter (SSC) vs fluorescence signal for MCF-7 cells that were either untreated (ctrl), or treated with Dox-TPP loaded BSA-PLGA (**e**) or ATRAM-BSA-PLGA (**f**) NPs for 1 or 4 h at pH 7.4 or 6.5; **g** quantification of cellular uptake of BSA-PLGA and ATRAM-BSA-PLGA NPs at different incubation times and pHs from the flow cytometry analysis ($n = 4$). ***$P < 0.001$ compared with NPs at pH 7.4.

with sodium azide/deoxyglucose[63], led to a reduction in cellular internalization of the NPs to $53 \pm 4\%$ of controls (Fig. 5a, b, g). Our results therefore show that the cellular internalization of the ATRAM-BSA-PLGA NPs occurs by both energy-dependent and energy-independent uptake mechanisms.

In order to determine whether the energy-dependent uptake mechanism of the peptides involves a specific endocytic pathway, MCF-7 cells were pretreated with endocytosis inhibitors. Chlorpromazine, which inhibits clathrin-coated pit formation[64], reduced the amount of internalized Dox-TPP to $49 \pm 1\%$ of controls (Fig. 5c, g), indicating that clathrin-mediated endocytosis contributes to the cellular internalization of the ATRAM-BSA-PLGA-NPs. Clathrin-mediated endocytosis, the best characterized endocytosis pathway, has been reported as the cellular uptake mechanism for a wide-range of nanocarriers[42]. On the other hand, treatment with methyl-β-cyclodextrin to remove cholesterol from the plasma membrane and disrupt several lipid raft-mediated endocytic pathways, including caveolae-dependent endocytosis[65], had little effect on the cellular uptake of the NPs, indicating that internalization is lipid raft-independent (Fig. 5d, g). This was confirmed by treating the cells with filipin, an inhibitor of lipid raft-mediated caveolae endocytosis[66], which again had a negligible effect on the uptake of the NPs (Fig. 5e, g). Likewise, amiloride, a specific inhibitor of the Na$^+$/H$^+$ exchange required for micropinocytosis[67], did not significantly inhibit uptake of the NPs (Fig. 5f, g). Taken together, these results show that pH-dependent cellular uptake of ATRAM-BSA-PLGA NPs occurs by both direct translocation and clathrin-mediated endocytosis.

Translocation across the plasma membrane provides the ATRAM-BSA-PLGA NPs with direct access to the cytosol, where the elevated levels of GSH would dissociate the crosslinked BSA shell and release the Dox-TPP cargo (Fig. 3d). In the case of uptake by clathrin-mediated endocytosis, the acidification of endocytic compartments during endosomal maturation would result in degradation of the PLGA core (Fig. 3b). Moreover, the low lumenal pH of late endosomes would promote endosome membrane insertion and destabilization by ATRAM, as has been reported for other endocytosed pH-triggered peptides[68,69], thereby releasing the entrapped Dox-TPP cargo into the cytosol. Thus, following pH-dependent cellular uptake by both mechanisms, direct translocation and clathrin-mediated endocytosis, the ATRAM-BSA-PLGA NPs would be degraded and the cargo released intracellularly (Fig. 5h).

**Cytotoxic effects of drug-loaded ATRAM-BSA-PLGA NPs.** The effectiveness of ATRAM-BSA-PLGA NPs in delivering chemotherapeutics was assessed using the MTS assay. PLGA NPs without drug did not have an adverse effect on MCF-7 cell viability (Fig. 6a). This is in agreement with the reported biocompatibility of PLGA, which is approved by the FDA for use in drug formulations[46]. Likewise, treatment with drug-free BSA-PLGA up to a reasonably high concentration of $75\,\mu g\,mL^{-1}$ did not result in a significant loss of MCF-7 cell viability. Furthermore, drug-free ATRAM-BSA-PLGA NPs did not exhibit cytotoxicity up to a concentration of $75\,\mu g\,mL^{-1}$ at pH 7.4 (Fig. 6a) or 6.5 (Supplementary Fig. 15), which is consistent with the previously reported lack of toxicity of the ATRAM peptide at either physiological or acidic pH[55]. Taken together, the cell viability results suggest that the designed ATRAM-BSA-PLGA NPs are nontoxic and compatible for use as a pH-responsive drug delivery system.

Treatment of human cancer cell lines, MCF-7, HeLa and pancreatic carcinoma MIA PaCa-2 with the Dox-TPP loaded NPs resulted in substantially greater loss of viability at pH 6.5

compared to pH 7.4, particularly at higher Dox-TPP concentrations ($>0.5\,\mu g\,mL^{-1}$) (Fig. 6b–d). Likewise, incubation of mouse cancer cell lines, neuroblastoma Neuro-2a and breast cancer 4T1, with Dox-TPP loaded NPs led to significantly greater toxicity at pH 6.5 compared to 7.4 (Fig. 6e, f). As a control, we measured the effects of Dox-TPP loaded BSA-PLGA NPs on MCF-7 and HeLa cell viability (Supplementary Fig. 16). BSA-PLGA NPs loaded with up to $2\,\mu g\,mL^{-1}$ Dox-TPP were not toxic at either pH 7.4 or 6.5. This confirms that the pH-responsive ATRAM is required for the both the pH-dependent uptake (Fig. 4d–g) and cytotoxicity of the coupled NPs (Fig. 6b–f). Although the Dox-TPP loaded ATRAM-BSA-PLGA NPs induced pH-dependent toxicity in all cell lines tested, the greatest effect was observed in MCF-7 and MIA PaCa-2 cells, suggesting that the pH-responsive hybrid NPs could be particularly effective in targeting human breast and pancreatic cancers.

To determine whether the observed cytotoxicity occurs via apoptosis or necrosis, we used Alexa 488-conjugated annexin V/propidium iodide (PI) staining (Fig. 6g–i). Treatment of MCF-7 cells with ATRAM-BSA-PLGA NPs loaded with $0.5\,\mu g\,mL^{-1}$ Dox-TPP for 12 h at pH 6.5 resulted in $74 \pm 2\%$ of the cells undergoing early apoptosis. Thus, the pH-dependent cancer cell death due to treatment with ATRAM-BSA-PLGA NPs occurs via apoptosis mediated by the Dox-TPP cargo.

To ascertain the generality of the ATRAM-BSA-PLGA NPs as a delivery platform for anticancer drugs, the NPs were loaded with paclitaxel (PTX; Supplementary Table 1), a hydrophobic chemotherapeutic commonly used to treat a wide range of cancers. Paradoxically, many of the side-effects associated with PTX are attributed to its delivery vehicle, the polyoxyethylated castor oil Cremophor EL[70]. MCF-7 cells were treated with PTX-loaded NPs at pH 7.4 and 6.5, and the cell viability was again quantified using the MTS assay. As observed with Dox-TPP-loaded NPs, treatment with PTX-loaded NPs induced markedly higher cytotoxicity at pH 6.5 compared to pH 7.4 (Supplementary Fig. 17). Taken together, these results show that the ATRAM-BSA-PLGA NPs exhibit selectivity for cancer cells in an acidic environment, such as that of tumors, compared to normal cells under physiological conditions. Moreover, the ATRAM-BSA-PLGA NPs may help to mitigate the side-effects due to some of the delivery vehicles of current hydrophobic chemotherapeutics.

**Macrophage recognition and immunogenicity of ATRAM-BSA-PLGA NPs.** Opsonization of NPs and their subsequent uptake by monocytes and macrophages of the mononuclear phagocyte system (MPS) leads to accumulation of the NPs in healthy organs (e.g., spleen and liver) rather than at the target solid malignant tumors (tumor-associated macrophages are discussed in Supplementary Note 4)[71]. To overcome this issue, the surface of NPs is often functionalized with neutral molecules, e.g., PEG, known to resist protein adsorption and MPS clearance[72]. However, studies have reported that surface modification with compounds such as PEG may trigger an immune reaction[62].

Cellular uptake of the NPs in differentiated human monocytic leukemia THP-1 cells, a widely used model of monocyte/macrophage activation[73], was assessed using confocal fluorescence microscopy. Treatment of THP-1 cells with naked PLGA NPs at pH 7.4 led to accumulation of a substantial amount of the Dox-TPP cargo intracellularly over time, indicating that the PLGA NPs are readily taken up by macrophages (Fig. 7a). In contrast, minimal intracellular Dox-TPP was observed for THP-1 cells treated with crosslinked BSA-PLGA (Supplementary Fig. 18) or ATRAM-BSA-PLGA (Fig. 7b) NPs at pH 7.4 for the duration of the experiment. Quantification of cellular uptake using flow

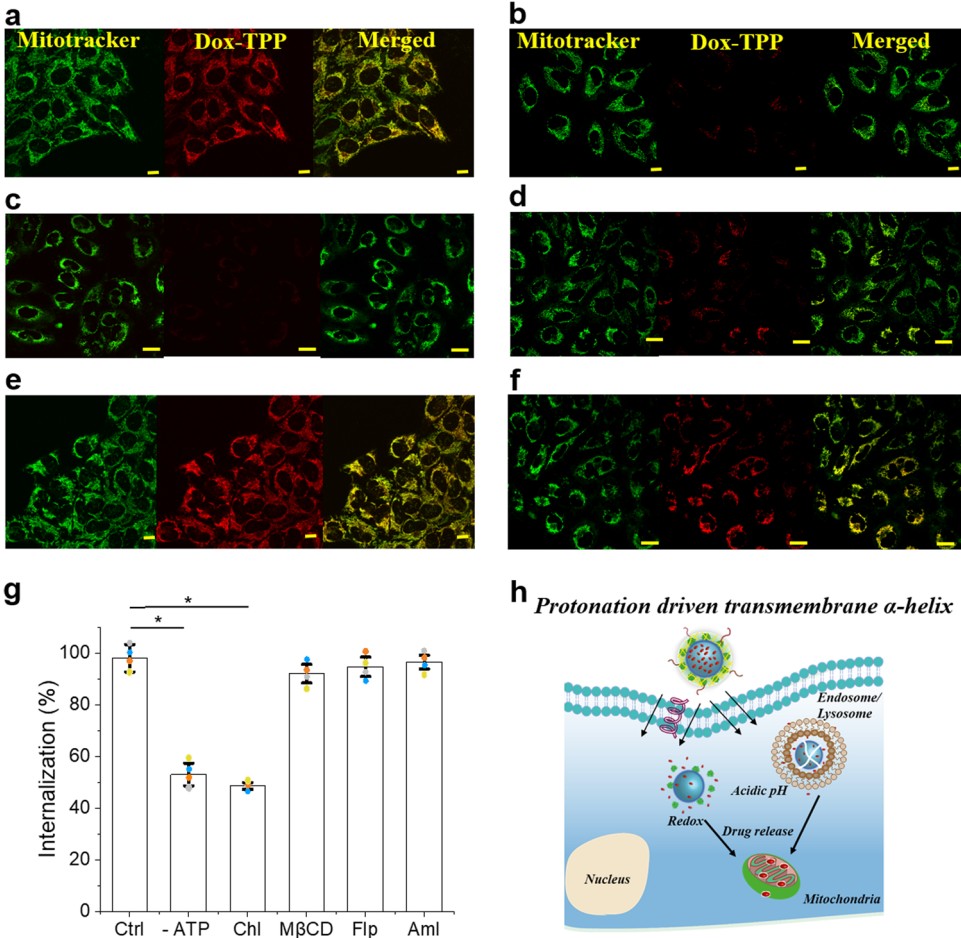

**Fig. 5 Determination of cellular uptake mechanisms of ATRAM-BSA-PLGA NPs. a–f** Confocal laser scanning microscopy images of MCF-7 cells treated with Dox-TPP loaded ATRAM-BSA-PLGA NPs for 1 h, at acidic pH 6.5. Uninhibited uptake in control cells (**a**) was compared to uptake in cells that were pretreated with sodium azide and 2-deoxy-D-glucose to deplete cellular ATP (**b**). Alternatively, the cells were pretreated with endocytosis inhibitors: chlorpromazine (Chlor; clathrin-dependent endocytosis) (**c**), methyl-β-cyclodextrin (MβCD; lipid raft-mediated endocytosis) (**d**), filipin (Flp; caveolae-dependent endocytosis) (**e**) or amiloride (Aml; macropinocytosis inhibitor) (**f**). Scale bar = 10 μm. **g** Flow Cytometry analysis of cellular uptake of Dox-TPP loaded ATRAM-BSA-PLGA NPs in MCF-7 cells under conditions in (**a–f**) ($n = 4$). $*P < 0.05$ compared with controls. **h** Schematic representation of efficient cellular uptake, by both energy-independent and -dependent mechanisms, of the ATRAM-BSA-PLGA NPs at acidic tumoral pH, followed by intracellular release of the Dox-TPP cargo.

cytometry showed that internalization of crosslinked BSA-PLGA or ATRAM-BSA-PLGA NPs in THP-1 cells at pH 7.4 was ~5 fold lower compared to PLGA NPs (Fig. 7c, d). Moreover, exposure to Dox-TPP loaded PLGA NPs at pH 7.4 induced considerable toxicity in THP-1 cells, whereas Dox-TPP loaded crosslinked BSA-PLGA or ATRAM-BSA-PLGA NPs did not affect THP-1 cell viability (Fig. 7f).

To assess the inflammatory potential of ATRAM-BSA-PLGA NPs, we quantified the production of tumor necrosis factor-alpha (TNF-α) and interleukin-1 beta (Il-1β) by differentiated human THP-1 monocytes exposed to the NPs (Fig. 7g). Treatment of THP-1 cells with PLGA NPs loaded with 0.5 μg mL$^{-1}$ Dox-TPP resulted in production of ~39 and ~58 pg mL$^{-1}$ of TNF-α and Il-1β, respectively. This was comparable to the ~40 and ~60 pg mL$^{-1}$ of TNF-α and Il-1β, respectively, produced by THP-1 cells due to exposure to the positive control LPS. In contrast, Dox-TPP loaded crosslinked BSA-PLGA and ATRAM-BSA-PLGA NPs induced production of negligible levels (<5 pg mL$^{-1}$) of TNF-α and Il-1β by THP-1 cells, which demonstrates the lack of immunogenicity of the hybrid NPs. Taken together, these results show that ATRAM-BSA-PLGA NPs are able to effectively evade recognition and uptake by macrophages (Fig. 7e).

**In vivo tumor inhibition by Dox-TPP loaded ATRAM-BSA-PLGA NPs.** The Dox-TPP loaded NPs showed a much longer in vivo circulation half-life[74] ($t_{1/2} = $~7 h) compared to free Dox ($t_{1/2} < 30$ min) (Fig. 8a). Moreover, the drug was detected in the plasma up to 48 h after treatment with the Dox-TPP loaded NPs, whereas free Dox was completely eliminated from the bloodstream in <16 h. The extended in vivo circulation time of ATRAM-BSA-PLGA NPs should facilitate greater accumulation of the chemotherapeutic cargo in tumor tissue and increased antitumor activity.

Next, we assessed the antitumor efficacy of the Dox-TPP loaded ATRAM-BSA-PLGA NPs (Fig. 8b). As expected, drug-free BSA-PLGA and ATRAM-BSA-PLGA NPs did not affect tumor growth (Fig. 8c–e and Supplementary Fig. 19a). Free Dox lead to a moderate slowing of tumor growth, with the tumor volume and weight following treatment ~78% and 55%, respectively, those of the saline treated controls (Fig. 8c–e). A more pronounced effect was observed with the Dox-TPP loaded BSA-PLGA NPs (tumor volume and weight after treatment were ~60% and 38%, respectively, those of controls) (Fig. 8c–e). However, the most effective treatment was the Dox-TPP loaded ATRAM-BSA-PLGA NPs, which completely inhibited tumor growth. Indeed, Dox-TPP

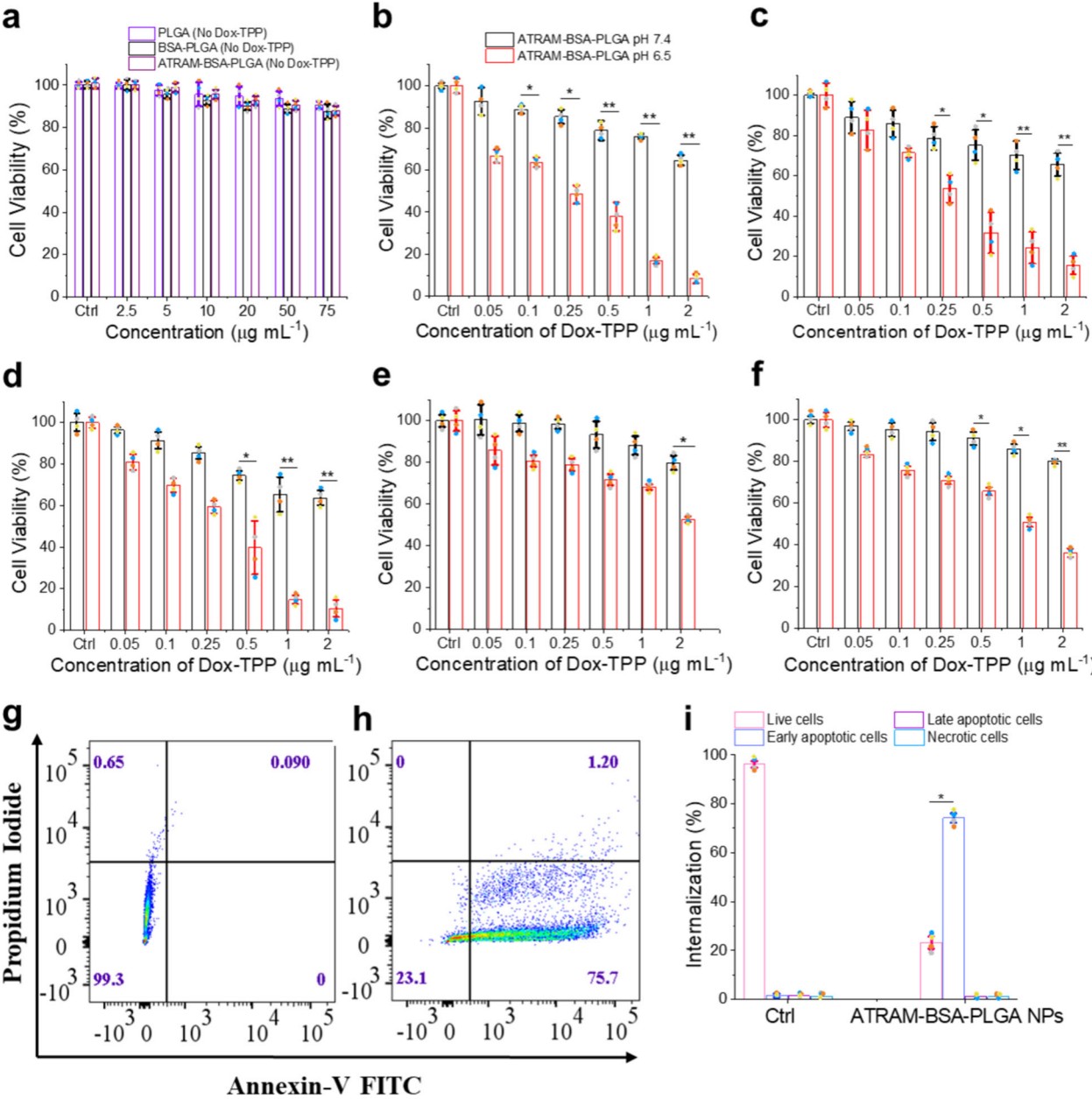

**Fig. 6 Mechanism of pH-dependent cytotoxicity of Dox-TPP loaded ATRAM-BSA-PLGA NPs. a** Cell viability of MCF-7 cells treated with PLGA, BSA-PLGA or ATRAM-BSA-PLGA NPs (all without Dox-TPP) for 48 h ($n = 4$). **b–f** Effect of Dox-TPP loaded ATRAM-BSA-PLGA NPs on viability MCF-7 (**b**), HeLa (**c**), MIA PaCa-2 (**d**), Neuro-2a (**e**), and 4T1 (**f**) cancer cells after 48 h incubation at pH 7.4 or 6.5. Cell viability in (**a–f**) was assessed using the MTS assay, with the % viability determined form the ratio of the absorbance of the treated cells to the control cells ($n = 4$). *$P < 0.05$, **$P < 0.01$ compared with pH 7.4. **g, h** Flow cytometry analysis of annexin V/propidium iodide (PI) staining of MCF-7 cells that were either untreated (control; **g**), or treated with ATRAM-BSA-PLGA NPs loaded with 0.5 µg mL$^{-1}$ Dox-TPP for 12 h at pH 6.5 (**h**). The bottom left quadrant (annexin V-/PI-) represents live cells; bottom right (annexin V+/PI−), early apoptotic cells; top right (annexin V+/PI+), late apoptotic cells; and top left (annexin V−/PI+), necrotic cells. **i** A summary of the incidence of early/late apoptosis and necrosis in the MCF-7 cells treated with Dox-TPP loaded ATRAM-BSA-PLGA NPs determined from the flow cytometry analysis of annexin V/PI staining ($n = 4$). *$P < 0.05$ compared with controls.

loaded ATRAM-BSA-PLGA NPs reduced the tumors from an initial volume of $26 \pm 2$ mm$^3$ to $23 \pm 2$ mm$^3$ after treatment (Fig. 8c), while the tumor mass was ~24% that of the saline treated controls (Fig. 8d, e). Histological analysis of tumor tissues using hematoxylin and eosin (H&E) staining confirmed the enhanced antitumor efficacy of the Dox-TPP ATRAM-BSA-PLGA NPs compared to the other treatment groups (Fig. 8f). Importantly, treatment with Dox-TPP loaded ATRAM-BSA-PLGA NPs greatly prolonged survival compared to free Dox (Fig. 8g) over the 90-day duration of the experiment.

Similar to drug-free ATRAM-BSA-PLGA NPs (Supplementary Fig. 19b), Dox-TPP loaded ATRAM-BSA-PLGA NPs did not adversely affect the body weight of treated mice (Supplementary Fig. 20), and H&E-stained lung, liver, spleen, heart, and kidney sections showed no apparent abnormalities or lesions (Supplementary Fig. 21). Taken together, these results demonstrate that the ATRAM-BSA-PLGA NPs effectively target tumor tissue, thereby enhancing the efficacy of their chemotherapeutic cargo, while also minimizing the non-specific toxicity that is common to conventional chemotherapy.

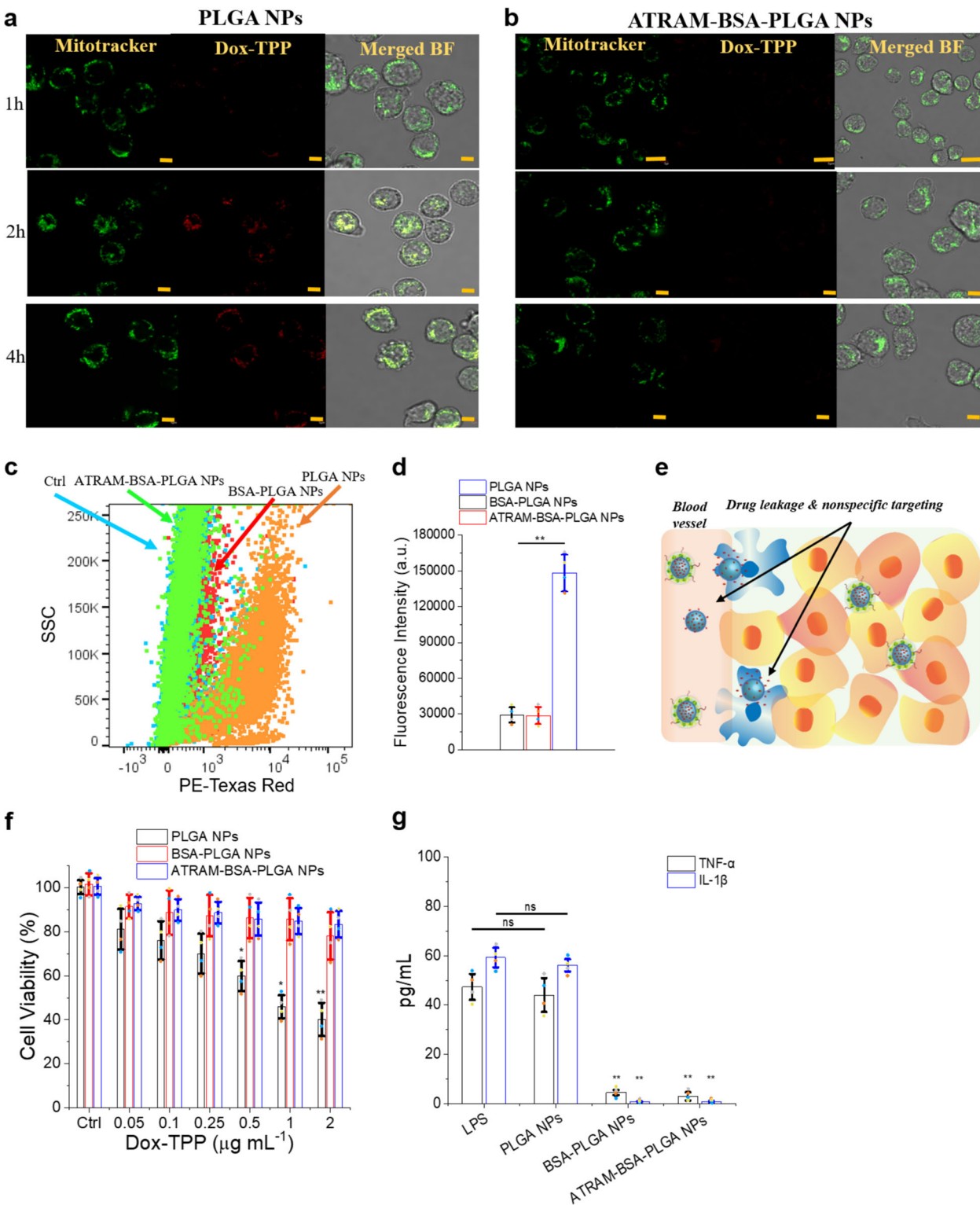

**Conclusions**. We have designed a biocompatible, biodegradable, and economical nanocarrier that combines high encapsulation stability with targeted delivery of chemotherapeutic drugs to cancer cells. The hybrid NPs were prepared by covalently 'wrapping' BSA over the surface of a PLGA core using simple carbodiimide chemistry in mild reaction conditions. The PLGA core can readily be loaded with a variety of chemotherapeutics, such as the mitochondria-targeting Dox-TPP, which is designed to overcome drug resistance in cancer cells. The BSA shell enables the NPs to evade recognition by neutrophils and macrophages and avoid interactions with blood serum components that inhibit target recognition and result in nonspecific distribution. The BSA shell is crosslinked to provide high circulation stability (i.e., retention of loaded cargoes in the presence of high concentrations of serum proteins) combined with degradation and drug release due to stimuli present in the unique intracellular environment of cancer cells (e.g., elevated levels of GSH). To facilitate specific tumor cell targeting, the hybrid NPs are functionalized with the

**Fig. 7 ATRAM-BSA-PLGA NPs evade uptake by differentiated human monocytic leukemia THP-1 cells. a, b** Confocal laser scanning microscopy images of THP-1 cells incubated with Dox-TPP loaded PLGA (**a**) or ATRAM-BSA-PLGA (**b**) NPs for 1–4 h at pH 7.4. Scale bar = 5 μm. **c, d** Flow cytometry analysis of cellular uptake of Dox-TPP loaded PLGA, crosslinked BSA-PLGA and ATRAM-BSA-PLGA NPs in THP-1 cells: **c** plot of side scatter (SSC) vs fluorescence signal for THP-1 cells that were either untreated (ctrl), or treated with Dox-TPP loaded PLGA, crosslinked BSA-PLGA or ATRAM-BSA-PLGA NPs for 2 h at pH 7.4; **d** quantification of cellular uptake of the NPs from the flow cytometry analysis (n = 4). **e** Schematic representation of sequestration of PLGA NPs, but not crosslinked BSA-PLGA or ATRAM-BSA-PLGA NPs, by macrophages (e.g., monocytes). **f** Cell viability of THP-1 cells treated with Dox-TPP loaded PLGA, crosslinked BSA-PLGA or ATRAM-BSA-PLGA NPs for 48 h at pH 7.4. Cell viability was assessed using the MTS assay, with the % viability determined from the ratio of the absorbance of the treated cells to the control cells (n = 4). **g** Release of inflammatory cytokines, tumor necrosis factor-alpha (TNF-α) and interleukin-1 beta (IL-1β), by THP-1 cells exposed to Dox-TPP loaded PLGA, crosslinked BSA-PLGA or ATRAM-BSA-PLGA NPs for 24 h at pH 7.4. Cells treated with lipopolysaccharide (LPS) were used as a positive control for inflammation. TNF-α and IL-1β levels in the culture medium were assayed using a commercial ELISA kit (n = 4). *P < 0.05, **P < 0.01 or non-significant (ns, P > 0.05) compared with controls.

pH-responsive ATRAM peptide. Dox-TPP loaded ATRAM-BSA-PLGA NPs showed highly efficient pH-dependent cellular uptake, by both energy-independent and -dependent internalization mechanisms, and remarkable cytotoxicity in a wide range of cancer cell lines. Importantly, ATRAM-BSA-PLGA NPs extended the in vivo circulation half-life of the Dox-TPP cargo, and treatment with Dox-TPP loaded NPs effectively reduced tumor volume and mass and prolonged survival, while exhibiting no obvious toxicity to healthy tissue. Taken together, our studies underline the potential of the ATRAM-functionalized high stability hybrid BSA-PLGA NPs as a platform for targeted cancer drug delivery.

## Methods

**Reagents**. Acetone, ammonium bicarbonate, BSA, N,N″-dicyclohexylcarbodiimide, N,N-dimethylformamide (DMF), 1,4-dithiothreitol (DTT), doxorubicin hydrochloride (Dox), 1-ethyl-3-(3-dimethylaminopropyl)carbodiimide hydrochloride (EDC), endocytosis inhibitors (amiloride, chlorpromazine, cytochalasin D and filipin), N-(2-hydroxyethyl) piperzine-N″-(2-ethanesulfonic acid) (HEPES), H&E, lipopolysaccharide (LPS), 2-(N-morpholino)ethanesulfonic acid (MES), N-hydroxysuccinimide (NHS), PTX, phosphotungstic acid, phosphate buffered saline (PBS), phorbol 12-myristate 13-acetate, PLGA, polyvinyl alcohol (PVA; MW = 30,000–70,000), Resomer RG 502 (lactide:glycolide 50:50; $M_w$ = 7000–17,000), sodium acetate buffer, 2,4,6-trinitrobenzenesulfonic acid solution (TNBS), (3-carboxypropyl)triphenylphosphonium bromide (TPP) and trimethylamine were all purchased Sigma (St. Louis, MO). Dead Cell Apoptosis kit, enzyme-linked immunosorbent assay (ELISA) kits (for detection of tumor necrosis factor-alpha (TNF-α) and interleukin-1 beta (Il-1β), FRET pair 3,3′-dioctadecyloxacarbocyanine perchlorate (DiO) and 1,1′-dioctadecyl-3,3,3′,3′-tetramethylindocarbocyanine perchlorate (DiI)), MitoTracker Green FM, N,N″-dimethyl-N-(iodoacetyl)-N″′-(7-nitrobenz-2-oxa-1,3-diazol-4-yl)ethylenediamine (NBD), Pierce BCA Protein Assay kit, Pierce Trypsin/Lys-C Protease Mix and rhodamine B were obtained from Thermo Fisher (Waltham, MA). CellTiter 96 AQueous One Solution (MTS) Cell Proliferation Assay kit were from Promega (Madison, WI).

**Synthesis of Dox-TPP**. Dox-TPP was synthesized using published procedures[29]. Briefly, 45 mg (0.10 mmol) TPP was added to 10 mL anhydrous DMF in a round-bottom flask, and the mixture was stirred until TPP dissolved completely. To this 25 mg (0.012 mmol) DCC and 14 mg (0.12 mmol) NHS were added and stirred for 4 h at room temperature. The mixture was then centrifuged to remove dicyclo-hexylurea and the supernatant was collected. Next, 60 mg (0.10 mmol) Dox was dissolved in 15 mL anhydrous DMF, and 15 μL (0.11 mmol) triethylamine (0.11 mmol) was added dropwise to the Dox solution while stirring. The supernatant of activated TPP was then added dropwise to the Dox solution and stirred overnight in the dark. The reaction mixture was precipitated with cold ether and centrifuged to obtain the crude product, which was dissolved in chloroform and extracted with deionized water to remove impurities from the chloroform solution in a separating funnel. Finally, the organic phase was dried and the product was purified completely using high performance liquid chromatography (HPLC). The Dox-TPP product was characterized using $^1$H NMR (Bruker AVANCE III HD 600 MHz NMR spectrometer), mass spectrometry (Agilent 6538 QToF LC/M) and zeta potential measurements (Malvern Zetasizer Nano ZS).

**Synthesis and characterization of BSA-PLGA NPs**. Dox-TPP loaded PLGA NPs were prepared by a simultaneous emulsion solvent diffusion method[56]. Briefly, a Dox-TPP/Resomer solution was prepared by mixing 10 mg Resomer with 200 μg Dox-TPP in 1 mL acetone. The Dox-TPP/Resomer solution was added dropwise to a 0.5% PVA solution in water and stirred at 1000 rpm at room temperature. The resulting solution was then transferred to 100 mL DI water and stirred overnight

for the dissipation/removal of the organic phase and hardening of the dox-loaded PLGA nanoparticles. Thereafter, the NPs were centrifuged at 15,000 rpm, washed thrice and finally redispersed in 2.5 mL MES buffer (0.11 M, pH 5.8) by bath sonication to get a clear dispersion.

For the shell of the NPs, BSA was chosen as an economical alternative to human serum albumin (HSA). Although BSA and HSA are of similar size (BSA and HSA are composed of 583 and 585 amino acids, respectively), and share a number of properties (including high solubility in water, long in vivo circulation half-life, biocompatibility and biodegradability), BSA costs a fraction of HSA[75,76]. BSA was separately conjugated to the surface of Dox-TPP loaded PLGA NPs by a two-simple step process, which involves activating carboxyl groups on the NP surface using an 1-ethyl-3-(-3-dimethylaminopropyl) carbodiimide/N-hydroxysuccinimide (EDC/NHS) mixture, followed by covalently coupling the activated carboxyl groups to the amino groups of BSA[77]. One milliliter of activating solution (0.1 M EDC and 0.07 M NHS in MES buffer, pH 5.8) was added to the NP dispersion (2.5 mL) and the mixture was incubated for 30 min under continuous stirring. To this mixture, 5 mL PBS (pH 7.4) was added to terminate the activation reaction, and the reactants (EDC/NHS) were removed by dialysis. The NP dispersion was concentrated to 1 mg mL$^{-1}$ by microcentrifuge dialysis and then incubated with 0.5 mL BSA solution (1 mg mL$^{-1}$ in PBS) for 12 h in the dark under continuous stirring. Finally, unbound BSA was removed by dialysis and excess coupling sites were blocked by incubating the NP dispersion with 0.5 mL glycine solution (10 mg mL$^{-1}$ in PBS) for 30 min, after which excess glycine was removed by dialysis. The concentration of BSA conjugated to PLGA NPs was determined using the BCA protein assay[78].

BSA on the surface of the NPs was crosslinked with bifunctional compound GA[79] for 30 min, creating a structure that is more stable than that attained by physical aggregation of BSA. The reaction is simple and does not require metal-catalysts, addition of salts, organic solvents, or non-ionic polymers that may cause toxicity[6]. Excess and unreacted GA was removed by dialyzing three times against phosphate buffer. The extent of crosslinking density was analyzed using the trinitrobenzenesulfonic acid (TNBS) assay[80], in which the free primary amine groups of proteins—in this case, crosslinked BSA on the NP surface—react with the sodium salt of TNBS. BSA was used as a standard. Crosslinked BSA-PLGA NP samples and BSA standards were diluted in 0.1 M borate buffer containing 0.15 M NaCl (pH 8.1). Fifty microliters of aqueous TNBS solution (30 mM) was then added dropwise to 2 mL sample mixtures (50 μg mL$^{-1}$) and stirred for 30 min at room temperature. Finally, the optical density/absorbance of the TNBS reaction product was measured at 420 nm (Synergy H1MF Multi-Mode Microplate-Reader; BioTek, Winooski, VT), with borate buffer as a reagent blank. The degree of crosslinking of the BSA-PLGA NPs, in terms of modified lysine residues, was calculated using the following formula:

$$\text{Modified fraction/crosslinking density (\%)} = (A - B/A) \times 100, \quad (1)$$

where A and B are the slopes of the curves for the BSA standard and crosslinked BSA-PLGA NPs, respectively, as determined from the absorbance at 420 nm in the linear regime.

The synthesized crosslinked BSA-PLGA NPs were characterized using TEM and STEM (FEI Talos F200X Transmission Electron Microscope). The beam energy was regulated so as to minimize damage to the samples while imaging in TEM and STEM modes. TEM images were acquired using a 200 kV beam with spot size 5, gun lens 6 and a dose of 1.13–1.16 A/m$^2$. STEM images were acquired in HAADF (high-angle annular dark-field) mode with spot size 9, gun lens 4 and a screen current of less than 0.2 nA. This mode was helpful for clear depiction of the core-shell structure of the BSA-PLGA NPs, as all of the inelastically scattered beam was collected for the image formation. Further characterization of the NPs was done using dynamic light scattering (DLS), Fourier-transform infrared spectroscopy (FTIR, Agilent Cary 600 Series FTIR Spectrometer) and zeta potential measurements.

The Dox-TPP loading capacity of the crosslinked BSA-PLGA NPs was determined using the following formula:

$$\text{Loading capacity (\%)} = (M_0 - M_s)/W_0 \times 100 \quad (2)$$

where $M_0$ and $M_s$ represent the initial mass of Dox-TPP mass and mass of Dox-TPP loaded in the NPs, respectively.

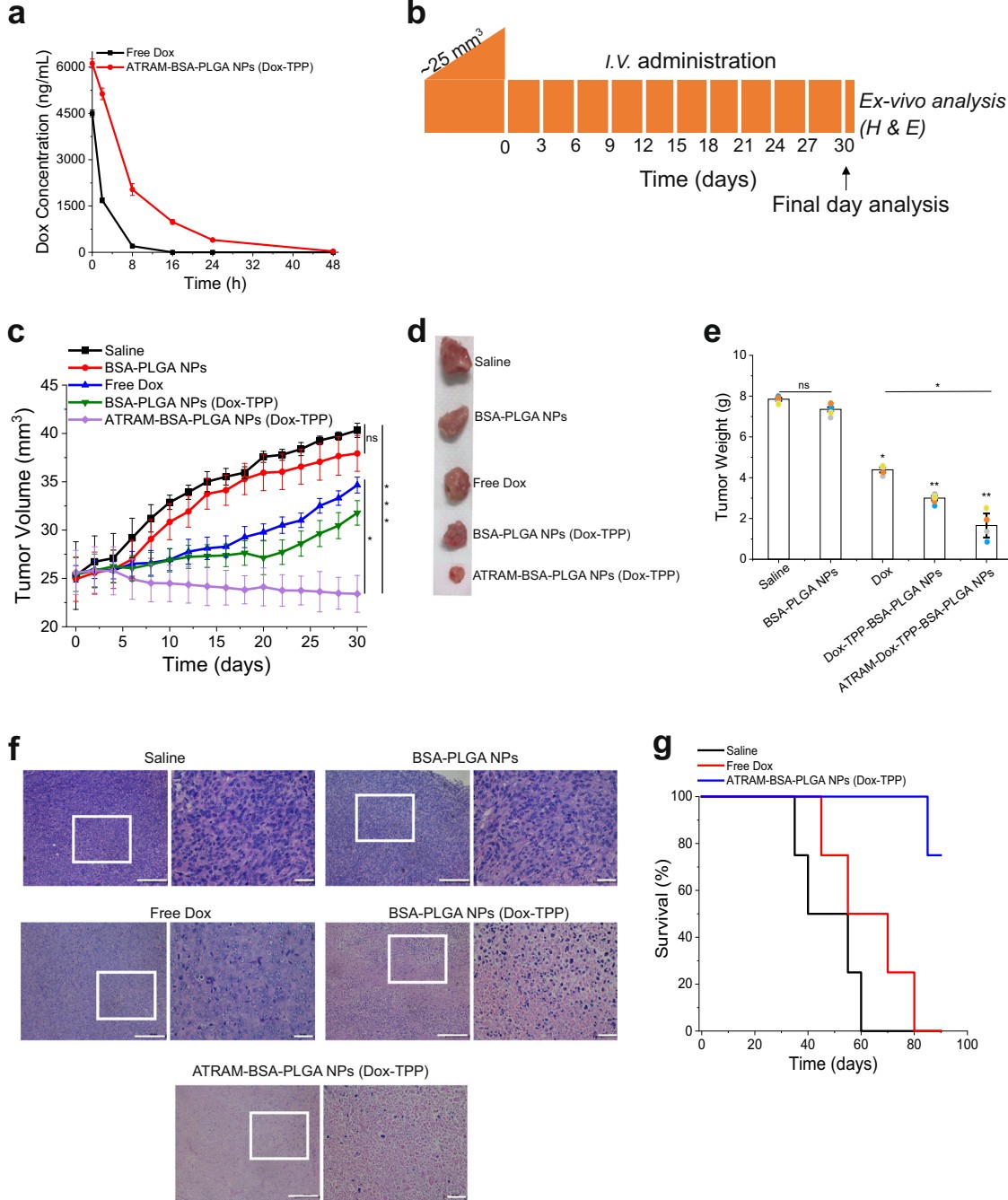

**Fig. 8 Inhibition of 4T1 tumor growth by Dox-TPP loaded ATRAM-BSA-PLGA NPs. a** In vivo pharmacokinetics of Dox-TPP loaded in ATRAM-BSA-PLGA NPs. The Dox concentration in plasma of mice ($n = 4$ per group) treated with free Dox (2 mg kg$^{-1}$) or Dox-TPP loaded BSA-PLGA NPs (115 mg kg$^{-1}$, with the loading capacity of Dox-TPP at 1.8 wt%) was quantified using HPLC[96] at different time points over 48 h. **b** Treatment schedule for the tumor reduction studies. Once the tumor volume reached ~25 mm$^3$, the mice were randomized into the different treatment groups ($n = 8$ per group), which were injected intravenously with: (1) saline; (2) BSA-PLGA NPs (115 mg kg$^{-1}$); (3) free Dox (2 mg kg$^{-1}$); (4) Dox-TPP loaded BSA-PLGA NPs (115 mg kg$^{-1}$, with the loading capacity of Dox-TPP at 1.8 wt%); (5) Dox-TPP loaded ATRAM-BSA-PLGA NPs (115 mg kg$^{-1}$, with the loading capacity of Dox-TPP at 1.8 wt%). Injections were done every 3 days for a total of 10 doses, with the first day of treatment defined as day 0. **c** Tumor volume growth curves for the different treatment groups over 30 days of treatment ($n = 8$ per group). **d, e** Tumor mass analysis for the different treatment groups. After 30 days of treatment, four mice per treatment group were sacrificed and the tumor tissues were isolated and imaged (**d**) and subsequently weighed to determine the tumor mass (**e**). **f** Hematoxylin and eosin (H&E)-stained images of tumor sections from the different treatment groups following 30 days of treatment. Images on the right are magnified views of the boxed regions in the images on the left. Scale bar = 50 μm. **g** Survival curves for the saline, free Dox and Dox-TPP loaded ATRAM-BSA-PLGA NPs treatment groups ($n = 4$ per group) over 90 days. *$P < 0.05$, **$P < 0.01$, ***$P < 0.001$ or non-significant (ns, $P > 0.05$) for comparison with controls and amongst the different treatment groups.

Encapsulation stability and drug release kinetics of the NPs were measured on PerkinElmer LS-55 Fluorescence Spectrometer. Encapsulation stability was assessed using FRET[81]. The NPs (PLGA, non-crosslinked BSA-PLGA and crosslinked BSA-PLGA) were co-loaded with the FRET donor (3,3′-dioctadecyloxacarbocyanine perchlorate, DiO) and acceptor (1,1′-dioctadecyl-3,3,3′,3′-tetramethylindocarbocyanine perchlorate, DiI) pair. Release of the dyes from the NPs, which results in decreased FRET, was monitored under different conditions (10 mM phosphate buffer, or cell culture medium containing 10% or 20% FBS) by measuring the acceptor DiI fluorescence ($\lambda_{em} = 565$ nm) upon excitation of the donor DiO ($\lambda_{ex} = 450$ nm). Cargo (Dox-TPP or rhodamine B) release from the NPs, in the absence or presence of a stimulus—e.g., pH or glutathione (GSH)—was monitored using the dialysis method[82]. Briefly, 1 mL of 200 μg mL$^{-1}$ Dox-TPP loaded NPs (PLGA, non-crosslinked BSA-PLGA or crosslinked BSA-PLGA) was placed in a dialysis bag (molecular-weight cutoff: 3 kDa), with or without GSH (0.5 or 10 mM), and fully submerged into 25 mL of release medium—10 mM phosphate buffer (pH 7.4 or 6.5) or 10 mM acetate buffer (pH 5.8, 5.0 or 4.0)—followed by stirring at 100 rpm. At the designated time points, 1 mL of the release medium was removed for analysis, and replenished with the same volume of fresh buffer. Dox-TPP fluorescence in the release sample was measured ($\lambda_{ex/em} = 480/580$ nm) and amount of Dox-TPP released was determined using a standard calibration curve.

**Quantitative proteomic analysis**. In order to investigate the interaction of serum proteins with PLGA and BSA-PLGA NPs, 1 mg mL$^{-1}$ of the NPs were incubated in 10 mM phosphate buffer or cell culture medium containing 10% FBS for 72 h. Thereafter, the serum proteins that adsorbed to the NPs were isolated by centrifugation following published methods[23,83]. Finally, the isolated serum proteins were de-N-glycosylated and analyzed by mass spectrometry (MS)[23,83]. BSA (1 mg mL$^{-1}$ in PBS) and FBS (10% in cell culture medium) were used as controls.

To prepare the digests, isolated serum proteins (~100 μg) were reduced (10 mM DTT, 85 °C, 30 min), and alkylated (25 mM IAA, 1 h, RT in the dark). Protease specific pH was achieved by diluting the sample with 50 mM ammonium bicarbonate buffer using a spin filter with a 30-kDa cutoff and digested with 1:50 (w/w) MS-grade trypsin/Lys-C protease mix for 24 h at 37 °C. The peptide digests were offline enriched using reverse-phase liquid chromatography, dried and re-substituted in 20 μL of a 2% acetonitrile/0.1% formic acid solution prior to online reverse-phase LC-MS/MS.

For LC-MS/MS analysis, the LC was done on an UltiMate 3000 RSLCnano System (Dionex) fitted with a C18 column (inner diameter = 75 μm, length = 50 cm; PepMap RSLC). Mobile phases were 0.1% formic acid (solvent A) and 80% acetonitrile/0.08% formic acid (solvent B). Samples were loaded in solvent A and eluted as follows: 10% B for 5 min, gradient to 55% B over 40 min then to 85% B over 5 min, 85% B for 6 min, followed by 14 min re-equilibration with 2% B. The LC system was coupled to a QTOF Impact II mass spectrometer (Bruker) equipped with an Easy Spray ion source and operated in positive ion mode. The spray voltage was set to 1.5 kV, with 3.0 L min$^{-1}$ dry gas and 165 °C dry temperature, and the full scans were acquired in a TOF MS mass analyser over $m/z$ 200–2200 at a spectral rate of 2.0 Hz. The auto MS/MS analyses with a fixed precursor cycle time of 3 s were performed using collision induced dissociation (CID). The precursor was released after 0.3 min. The raw files, converted to mgf format by the DataAnalysis software (Bruker Daltonik), were searched against reported proteomes using the ProteinScape software with an in-house Mascot search engine (Matrix Science Inc., Boston, MA). The search parameters were set as follows: peptide tolerance, 20 ppm; MS/MS tolerance, 0.5 Da; enzyme, trypsin; 2 missed cleavage allowed; and fixed carbamidomethyl modifications of cysteine. Oxidation of methionine and protein N-terminal acetylation were used as variable modifications.

Label-free protein quantification was done using the MaxQuant software (version 1.6.5.0) with default parameters. The raw data was searched against the Universal Protein Resource (UniProt)[84,85] database using the Andromeda search engine[86].

**Preparation of ATRAM-BSA-PLGA NPs**. The ATRAM peptide (N$_t$-GLA-GLAGLLGLEGLLGLPLGLLEGLWLGLELEGN-C$_t$) was synthesized by Selleck Chemicals (Houston, TX) using standard Fmoc methods. The peptide was purified in house by reverse-phase HPLC (Waters 2535 QGM HPLC), and purity was subsequently verified using mass spectrometry (Agilent 6538 QToF LC/MS). ATRAM was covalently coupled to the surface of the BSA-PLGA NPs using a simple carbodiimide (EDC) coupling reaction[87]. Briefly, 5 mg BSA-PLGA was dissolved in 5 mL phosphate buffer (50 mM) containing EDC (at a molar ratio of 2:1 to BSA in the NPs) at pH 5.5 and stirred for 5 min at 4 °C to activate the carboxyl groups on the BSA, and unreacted EDC was removed by dialysis. One milligram of ATRAM was dissolved in 50 mM phosphate buffer and added to the carboxyl activated BSA-PLGA NPs, and the mixture was continuously stirred for 6 h at 4 °C in order to covalently couple the peptide to BSA. Unconjugated peptide was removed by dialysis, and peptide conjugation was confirmed by release of the urea byproduct at 232 nm[57].

**Serum albumin binding assay**. NBD was conjugated to a C-terminal cysteine ATRAM variant (N$_t$-GLAGLAGLLGLEGLLGLPLGLLEGLWLGLELEC̲N-C$_t$). Free

dye and unlabeled peptide were separated from labeled peptide by gel filtration through a PD-10 column (GE Healthcare Bio-Sciences, Marlborough, MA) and reverse-phase HPLC (Agilent, Santa Clara, CA), respectively. Peptide labeling was confirmed by MALDI-TOF (Bruker, Billerica, MA).

BSA and ATRAM-NBD were prepared in PBS. A constant concentration of ATRAM-NBD (0.4 μM) was added to increasing concentrations of BSA (0.25–15 μM). Fluorescence anisotropy was measured on a Fluorolog-3 Spectrofluorometer (Horiba, Edison, NJ) at room temperature with the excitation and emission wavelengths set to 460 and 535 nm, respectively. Both excitation and emission slits were set to 6 nm. The data were fitted to determine the dissociation constant using the following equation:

$$<r> = <r>_b + (<r>_f - <r>_b)\frac{([L]+K_d+[P]) - \sqrt{([L]+K_d+[P])^2 - 4[L][P]}}{2[P]}$$

(3)

where $<r>$ is the measured NBD anisotropy, $<r>_f$ and $<r>_b$ are the anisotropy for the free and the completely bound fluorescent ligand, respectively. [L] and [P] represent the total fluorescent ligand (ATRAM-NBD) and protein (BSA) concentrations, respectively[27,60].

**Cell culture**. All cell lines were purchased from American Type Culture Collection (ATCC). Prior to use, the cells were authenticated and tested for mycoplasma contamination by Charles River Laboratories (Margate, United Kingdom). Human breast cancer MCF-7 cells (ATCC no. HTB-22), human cervical HeLa cells (ATCC no. CCL-2), human pancreatic carcinoma MIA PaCa-2 cells (ATCC no. CRL-1420) and mouse neuroblastoma Neuro-2a cells (ATCC no. CCL-131) were cultured in DMEM (Sigma) supplemented with 10% FBS (GE Healthcare Life Sciences, Logan, UT), 4 mM L-glutamine, 1 mM sodium pyruvate and 1% penicillin/streptomycin (all from Sigma), in 5% CO$_2$ at 37 °C. Mouse breast cancer 4T1 cells (ATCC no. CRL-2539) were cultured in RPMI 1640 (Sigma) supplemented with 10% FBS, 4 mM L-glutamine, 1 mM sodium pyruvate and 1% penicillin/streptomycin, in 5% CO$_2$ at 37 °C. Human monocytic leukemia THP-1 cells (ATCC no. TIB-202) were cultured in RPMI 1640 supplemented with 10% FBS, 10 mM HEPES, 4 mM L-glutamine, 1 mM sodium pyruvate, 0.05 mM 2-mercaptoethanol (Sigma) and 1% penicillin/streptomycin, in 5% CO$_2$ at 37 °C. THP-1 cells were differentiated into a macrophage-like phenotype by incubating them with 10 ng mL$^{-1}$ PMA in complete medium for 72 h in 5% CO$_2$ at 37 °C.

**Intracellular imaging and colocalization**. Intracellular imaging was done according to a previously published protocol[63]. Briefly, cells (MCF-7, HeLa or differentiated THP-1) were seeded at a density of $2 \times 10^5$ cells/well in 500 μL complete medium in 4-chambered 35 mm glass bottom Cellview cell culture dishes (Greiner Bio-One, Monroe, NC). After culturing for 24 h, the medium was replaced fresh medium (pH 7.4 or 6.5) containing 2.5 μg mL$^{-1}$ Dox-TPP loaded PLGA, BSA-PLGA or ATRAM-BSA-PLGA NPs and incubated for 1–4 h. For some experiments, MCF-7 cells were pre-incubated for 1 h at 4 °C in serum-free DMEM, pre-treated for 1 h at 37 °C with 10 mM sodium azide and 6 mM 2-deoxy-D-glucose in serum- and glucose-free DMEM, or pretreated for 30 min at 37 °C with the following drugs in serum-free DMEM: 10 μM chlorpromazine; 5 mM methyl-β-cyclodextrin; 5 μM filipin; or 5 μM amiloride. After addition of the NPs, the cells were maintained for 1 h at 4 °C or in presence of inhibitors at 37 °C. Thirty minutes prior to imaging, the medium was replaced with fresh medium containing 50 nM MitoTracker Green or vehicle. Finally, immediately prior to imaging, the medium was once again replaced with fresh medium to remove any extracellular markers. Imaging was done on an Olympus Fluoview FV-1000 confocal laser scanning microscope, using a 63× Plan-Apo/1.3 NA oil immersion objective with DIC capability. Image processing was done using the Fiji image processing software[88].

**Quantification of cellular uptake**. Cellular uptake of Dox-TPP loaded ATRAM-BSA-PLGA NPs at pH 7.4 or 6.5 was measured using a previously published flow cytometry assay[63]. Briefly, cells (MCF-7 or differentiated THP-1) were seeded at a density of $2 \times 10^4$ cells/well in 500 μL complete medium in 24-well plates. After culturing for 24 h, the cells were washed with PBS at 37 °C and the medium was replaced with fresh medium (pH 7.4 or 6.5) containing 2.5 μg mL$^{-1}$ Dox-TPP loaded PLGA, BSA-PLGA or ATRAM-BSA-PLGA NPs and incubated for 1–4 h at 37 °C. Subsequently, the cells were washed three times with ice-cold PBS to remove the extracellular NPs, and then treated with trypsin-EDTA for 5 min to detach the cells and remove cell surface-bound peptide. Finally, the cells were centrifuged (1000 × $g$ for 5 min at 4 °C) and re-suspended in 500 μL ice-cold PBS with 10% FBS. Data collection (10,000 cells/sample, gated on live cells by forward/side scatter and propidium iodide (PI) exclusion) was done immediately afterwards on a BD FACSAria III cell sorter (BD Biosciences, San Jose, CA), and analysis was performed using the BD FACSDiva software.

To elucidate the cellular internalization pathways of Dox-TPP loaded ATRAM-BSA-PLGA NPs, MCF-7 cells were pre-incubated for 1 h at 4 °C in serum-free DMEM, or pretreated for 1 h at 37 °C with 10 mM sodium azide/6 mM 2-deoxy-D-glucose in serum- and glucose-free DMEM, or pretreated for 30 min at 37 °C with the endocytosis inhibitors. After addition of 2.5 μg mL$^{-1}$ Dox-TPP loaded

ATRAM-BSA-PLGA NPs, the cells were maintained for 1 h at 4 °C or in the presence of inhibitors at 37 °C. Thereafter, the cells were washed three times with ice-cold PBS, trypsinized, centrifuged and re-suspended in 500 μL ice-cold PBS with 10% FBS, and fluorescence was measured by flow cytometry. Cells treated with NPs without inhibitors at 37 °C were used as control, and cells treated with vehicle alone served as background. The uptake efficiency was determined from the ratio of Dox-TPP fluorescence of cells treated with NPs under different inhibition conditions to the control cells.

**Cellular FRET**. MCF-7 cells were seeded at a density of $2 \times 10^5$ cells/well in 500 μL complete DMEM in 4-chambered 35 mm glass bottom Cellview cell culture dishes. After culturing for 24 h, the medium was replaced fresh DMEM containing 0.1 mg mL$^{-1}$ NPs (PLGA or crosslinked BSA-PLGA) co-loaded with 0.61% DiO/DiI or loaded with 0.70% DiI, and incubated for 4 h. Immediately prior to imaging, the medium was replaced with fresh DMEM to remove any extracellular NPs. FRET images were acquired on an Olympus Fluoview FV-1000 confocal laser scanning microscope, using a 63× Plan-Apo/1.3 NA oil immersion objective with DIC capability. Images were recorded in the DiO channel ($\lambda_{ex} = 488$ nm, $\lambda_{em} = 543$ nm; DiI channel ($\lambda_{ex} = 543$ nm, $\lambda_{em} = 633$ nm); FRET channel ($\lambda_{ex} = 488$ nm, $\lambda_{em} = 633$ nm). The FRET ratio was calculated as: FRET ratio = $I_{FRET}/(I_{FRET} + I_{DiO})$. The exposure time was 200 ms for the DiO and FRET channels and 100 ms for the DiI channel.

**Cell viability/toxicity assays**. Cell viability/toxicity was measured using two complementary assays as previously published:[63] (i) CellTiter 96 AQueous One Solution (MTS) assay, which measures reduction of the tetrazolium compound MTS (3-(4,5-dimethylthiazol-2-yl)-5-(3-carboxymethoxyphenyl)-2-(4-sulfophenyl)-2H-tetrazolium, inner salt) to soluble formazan, by dehydrogenase enzymes, in living cells;[89,90] (ii) Dead Cell Apoptosis assay, in which Alexa 488-conjugated annexin V is used a sensitive probe for detecting exposed phosphatidylserine in apoptotic cells[91], and red-fluorescent PI, a membrane impermeant nucleic acid binding dye, assesses plasma membrane integrity and distinguishes between apoptosis and necrosis[92].

Cells were seeded at a density of $2 \times 10^4$ cells/well in 100 μL complete medium in standard 96-well plates. After culturing for 24 h, the medium was replaced with fresh medium (pH 7.4 or 6.5) containing 2.5–75 μg mL$^{-1}$ drug-loaded NPs, or NPs loaded with 0.05–2 μM Dox-TPP or PTX, and incubated for 48 h at 37 °C. Thereafter, the medium was replaced with fresh medium, and 20 μL MTS reagent was added to each well. The MTS reagent was incubated for 4 h at 37 °C, and absorbance of the soluble formazan product ($\lambda = 490$ nm) of MTS reduction was measured on a BioTek Synergy H1MF Multi-Mode Microplate-Reader, with a reference wavelength of 650 nm to subtract background. Wells treated with peptide-free carrier were used as control, and wells with medium alone served as a blank. MTS reduction was determined from the ratio of the absorbance of the treated wells to the control wells.

For the Dead Cell Apoptosis assay, MCF-7 cells were treated with ATRAM-BSA-PLGA NPs (loaded with 1 μg mL$^{-1}$ Dox-TPP) for 12 h at pH 6.5. Subsequently, the cells were washed with ice-cold PBS, harvested by trypsinization, centrifuged and re-suspended in 1× annexinbinding buffer (10 mM HEPES, 140 mM NaCl, 2.5 mM CaCl₂, pH 7.4) to a density of ~1 × 10⁶ cells per mL. The cells were then stained with 5 μL Alexa 488-conjugated annexin V and 0.1 μg PI per 100 μL of cell suspension for 15 min at room temperature. Immediately afterwards, fluorescence was measured using flow cytometry, and the fractions of live (annexin V−/PI−), early and late apoptotic (annexin V+/PI− and annexin V+/PI+, respectively), and necrotic (annexin V−/PI+) cells were determined.

**Inflammatory cytokine assay**. Tumor necrosis factor-alpha (TNF-α) and interleukin-1 beta (IL-1β), are inflammatory cytokines primarily produced by macrophages/monocytes during acute inflammation[93].

Differentiated THP-1 cells were seeded at a density of $2 \times 10^4$ cells/well in 100 μL complete medium in standard 96-well plates. After culturing for 24 h, the medium was replaced with medium containing PLGA, BSA-PLGA or ATRAM-BSA-PLGA NPs loaded with 0.5 μg mL$^{-1}$ Dox-TPP and incubated for 24 h at 37 °C. Cells treated with the macrophage activator lipopolysaccharide lipopolysaccharide (LPS)[94] were used as positive control, while untreated cells served as a negative control. Thereafter, the cell culture medium was assayed for secretion of TNF-α and IL-1β using commercial ELISA kits. The total TNF-α and IL-1β levels were determined from the absorbance ($\lambda = 450$ nm) measured on a BioTek Synergy H1MF Multi-Mode Microplate-Reader using a standard TNF-α concentration calibration curve.

**In vivo tumor inhibition studies**. All animal experiments were approved by the NYU Abu Dhabi Institutional Animal Care and Use Committee (NYUAD-IACUC; Protocol No. 18-0001), and were carried out in accordance with the Guide for Care and Use of Laboratory Animals[95]. Female C3H/HeJ mice (The Jackson Laboratory, Bar Harbor, ME) were bred in-house by the NYU Abu Dhabi Vivarium Facility in a 12 h light/dark schedule.

For pharmacokinetics, a simple and rapid high performance liquid chromatographic method (HPLC) method[96] was used to quantify the concentration of Dox

in plasma of mice treated with Dox-TPP loaded ATRAM-BSA-PLGA NPs. Following intravenous injection with one dose of Dox (2 mg kg$^{-1}$) or Dox-TPP loaded BSA-PLGA NPs (115 mg kg$^{-1}$, with the loading capacity of Dox-TPP at 1.8 wt%), blood was drawn at different time points over 48 h. Hundred microliters of plasma was separated from whole blood by centrifugation (2500 rpm for 15 min), then spiked with Dox (1 μg mL$^{-1}$). This was followed by addition of 100 μL Tris buffer (of 1 M, pH 8) to the isolated plasma, and extraction of Dox was performed thrice by dilution in 3 mL chloroform/methanol (9:1 v/v) and vortexing for 10 min, followed by centrifugation (11,000 rpm for 10 min). The organic phase was collected and evaporated to dryness under a N₂ stream. Thereafter, the dry residue was dissolved in 60 μL acetonitrile and centrifuged (11,000 rpm for 5 min), and the supernatant was collected and filtered using a 0.2-μm syringe filter. Finally, ~20 μL of the supernatant was injected into the HPLC and peaks were analyzed for Dox concentration against a standard calibration curve.

For tumor inhibition studies, $5 \times 10^6$ viable breast cancer 4T1 cells (ATCC no. CRL-2539) were injected subcutaneously into the right flank of each mouse at age 6–8 weeks. Mice were assessed daily for overt signs of toxicity. Tumor volume was measured via high-precision calipers (Thermo Fisher) using the following formula:

$$\text{Tumor volume}\left(\text{mm}^3\right) = \left(W^2 \times L\right)/2 \qquad (4)$$

where W and L are tumor width and length in mm, respectively[97]. Mice were euthanized once tumor volume approached burden defined by NYUAD-IACUC.

Once the tumor volume reached ~25 mm³, the mice were randomized into six treatment groups ($n = 8$ per group), which were injected intravenously (once every 3 days, for a total of 10 doses) with: (1) saline; (2) drug-free BSA-PLGA NPs (115 mg kg$^{-1}$); (3) drug-free ATRAM-BSA-PLGA NPs (115 mg kg$^{-1}$); (4) free Dox (2 mg kg$^{-1}$); (5) Dox-TPP loaded BSA-PLGA NPs (115 mg kg$^{-1}$, with the loading capacity of Dox-TPP at 1.8 wt%); (6) Dox-TPP loaded ATRAM-BSA-PLGA NPs (115 mg kg$^{-1}$, with the loading capacity of Dox-TPP at 1.8 wt%). The dose of doxorubicin administered was based on previously published work[18]. Body weight and tumor volume were recorded every 2 days, and survival ($n = 4$ per group) was monitored for a total of 90 days.

After the 30 days of treatment, four mice per treatment group were sacrificed and the tumor tissues were isolated to determine the tumor mass. For histological analysis, tumors and vital organs were dehydrated with 95% ethanol twice for 30 min, and then soaked in xylene for 1 h at 60–70 °C followed by paraffin for 12 h. The paraffin embedded tissues were sectioned into 7-μm slices, dewaxed on microscope slides, and finally stained with H&E using standard procedures[98]. The tissue sections were imaged on a NIKON LV-Dia Metallurgical Microscope.

**Statistics and reproducibility**. For in vitro studies, investigators were blinded for all parts of the experiments (treatment, data acquisition and data analysis), and a different investigator carried out each part. For in vivo studies, power calculation was used to select sample sizes from the NYU Abu Dhabi Institutional Animal Care and Use Committee (NYUAD-IACUC) Protocol (Protocol No. 18-0001). Confidence intervals in this work represent the standard deviation across at least three biological replicates (i.e., $n \geq 3$). Statistical analysis was performed using the Prism 7.0 software (GraphPad Software, Inc., La Jolla, CA, USA). Statistical significance between two groups was assessed by an unpaired $t$-test, and among three or more groups by two-way analysis of variance (ANOVA) followed by Tukey's post hoc test. $P < 0.05$ was considered to be statistically significant.

**Reporting summary**. Further information on research design is available in the Nature Research Reporting Summary linked to this article.

## Data availability

Source data for Figs. 1 and 3–8 are available in Supplementary Data 1. The proteomics data (Fig. 2) has been deposited to ProteomeXchange Consortium via the MassIVE repository (accession code: MSV000084848). All the datasets are also available from the authors on reasonable request.

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

## Acknowledgements
The authors thank the Center for Genomics and Systems Biology (CGSB) at NYU Abu Dhabi for use of their BD FACSAria III (BD Biosciences) for flow cytometry measurements. Imaging (confocal and TEM), as well as CD, FTIR, LC-MS/MS and zetasizer measurements, were carried out using the Core Technology Platform resources at NYU Abu Dhabi. This work was supported by funding from NYU Abu Dhabi and an ADEK Award for Research Excellence grant (AARE17-089) to M.M. and from the National Institutes of Health (R01GM120642) to F.N.B.

## Author contributions
L.P. and M.M. conceived the project. M.M. and F.N.B. provided funding and supervised the project. L.P., M.M., and F.N.B. designed the experiments. L.P., S.A.-H., M.K., V.P.N., L.A., and R.P. performed the experiments. L.P. and M.M. wrote the manuscript. All authors reviewed and edited the manuscript.

## Competing interests
The authors declare no competing interests.
