## [Peer Review File · Communications Biology]

Reviewers' comments:

Reviewer #1 (Remarks to the Author):

Palanikumar et al report here on the development of novel nanoparticles (NPs) for the targeted delivery of the cytotoxic agent doxorubicin to cancer cells. The NPs are based on FDA-approved PLGA, have a cross-linked BSA corona to avoid unspecific protein binding and opsonization and are decorated with a peptide that binds the cell membrane at low pHs as those in the tumor microenvironment. The results presented merit publication, especially given the demonstrated antitumor efficacy of the NPs in a mouse model in vivo, and will be very interesting for the field. However, a number of issues need to be answered by the authors before I can recommend publication.

Major comments:

1. Quantitative proteomics analysis (Figure 2): it is not clear from the explanation in the text what the FBS and BSA controls consist of. Should FBS not contain a high amount of protein?
2. Cellular internalization studies have only been reported at one time point (1 hour after exposure). The authors report that in such time frame, internalization is significantly higher at low pH. However, have the authors performed any tests at later time points? Most NPs may require longer times to interact and internalize and in fact 1 hour is a very short time. Internalization at higher pHs could be underestimated simply because of this short time of interaction.
3. Figure 4 (e,d): the authors affirm that the higher uptake at low pHs is caused by the ATRAM peptide, which would be explained by a higher affinity to the cell membrane. However, in order to be able to affirm that, a control of the same NP without the peptide is needed.
4. Cytotoxic effects of NPs (Figure 6a). Cytotoxicity of ATRAM-BSA-PLGA NPs loaded with dox is demonstrated. However, the empty NP controls utilized here are not correct, since they do not have the ATRAM peptide which enhances internalization. While previous studies indicate that this peptide is not cytotoxic, the increased NP uptake that it induces could lead to cytotoxic events. A control of empty ATRAM-BSA-PLGA NPs should be included.
5. NP uptake in macrophages (Figure 7). At which pH were these experiments performed? A condition at low pH should be included, because it is possible that at the lower pH of the tumor microenvironment the NPs are taken up by resident macrophages, which could be a limitation or an opportunity depending on the drug delivered, but in any case should be addressed.
6. Figure 8, antitumor efficacy in vivo: as in the in vitro cytotoxicity studies, a control with the exact same drug-free NP (including the peptide) should be desirable.

Minor comments:

1. The z potential values reported for the different NP formulations do not align with colloidal stability (too close to 0). Can the authors comment on how with such values the NPs are still stable for a period of 72 hours.
2. Size values from DLS experiments should also be reported on a table format. It is difficult to appreciate the specific values from the graphs.
3. The color code utilized in the heatmaps displayed in Figure 2 are not color-blind friendly. This should be changed to ensure all readers can interpret the data.
4. The term Fluorescence Activated Cell Sorting (FACS) is used incorrectly here. To my understanding, the authors have performed flow cytometry, but have not sorted the cells.

Reviewer #2 (Remarks to the Author):

Palanikumar et al. present the generation of a BSA coated nanoparticle (NP) that contains a pH activated ATRAM peptide for enhancing delivery into solid tumors due to the lower pH environment. The manuscript is well-written and exceptionally well-reasoned study on enhancing NP delivery of

chemotherapeutics by incorporating multiple molecular design elements. Impressively, the introduction elaborately (and politely) lays out the failures of the NP field to harness the presumed EPR in humans (vs. rodent models where it is much more prevalent). Overall, the studies are well controlled and the conclusions supported by the primary data. I have several concerns listed below that, given the high caliber of this group, the authors should be able to readily address. In conclusion, this is an exciting study on enhancing delivery of NPs in vivo that after addressing most of my concerns below should greatly exceed the bar for publication in CB.

Concerns:

1. Fig. 1e/f. The NPs become quite large when exposed to 10% FBS, likely ~200+ nm diameter, and that will dramatically negatively impact their delivery into solid tumors where they are swimming upstream against the tumor produced water flow. This is a significant concern with all NPs. However, to their credit, the authors are showing data that most NP groups do not. Question: Can you keep the size of the NPs to ~100 nm in serum, especially 100%?
2. Fig.3. The cross-linked BSA-PLGA NPs need to be added to panel 3a to have all of the NPs comparable on the same graph. Likewise, panel 3c needs the pH 5-0 control added here. Plus the authors need to add labels of " cross-linked BSA-PLGA NPs" to panels 3b/c/d. Panel 3d needs to no GSH control added.
3. Fig 4. ATRAM is a highly hydrophobic peptide and it is rather remarkable that the only two charged Glu residues in ATRAM have a pKa of 6.5 vs. the expected ~4.2. Beyond their prior publications, the authors need to comment and/or preferably add pKa data to support this point.
4. Fig 4b needs the zeta potential of the non-ATRAM conjugated control cross-linked BSA-PLGA NP.
5. Fig 6b-f. TO test for pH induced toxicity from the NP vs. solely from the DOX-TPP cargo, the authors need to repeat these experiments using the control of ATRAM conjugated cross-linked BSA-PLGA NPs that are not loaded with the DOX-TPP.
6. Fig 7. The control of cross-linked BSA-PLGA NP is missing and needs to be added into this figure.
7. Fig 8f should be updated to the current length of the experiment since the manuscript was first submitted.

Reviewer #3 (Remarks to the Author):

The manuscript "pH-Responsive High Stability Polymeric Nanoparticles for Targeted Delivery of Anticancer Therapeutics" from Palanikumar, L., et. al. highlights the development of biocompatible and biodegradable pH-responsive nanoparticles that display: 1) low non-specific release of drugs, 2) high selectivity towards cancer cells, 3) higher circulation stability, 4) selective triggered release of therapeutics, and 5) high applicability for in vivo treatment of tumors, decreasing their volume and improving survival rate; all key characteristics of an ideal delivery system. The authors also looked into the specifics of the uptake mechanism of this particular system by cancer cells, the cytotoxic pathway leading to cancer cell eradication, and the structural characteristics of the nanoparticles that lead to their anticancer properties, providing a complete assessment of the proposed nanoparticle design.

The project has been carried out with great attention to detail, and extensive tests to prove the advantages of these pH-responsive nanoparticles for cancer treatment, including their actual application in an in vivo scenario, and as such I consider it significantly relevant for a publication in Communications Biology. However, I do have many concerns and comments that I think should be addressed prior publication, to strengthen the scientific value of the manuscript.

1) The contents from Scheme 1 should be split into the actual figures of the manuscript at the points where the different topics are discussed. For instance, panel a and c can be made into one panel, and added to Figure 1, panel b to Figure 3, and panel d to Figure 5. I find it quite confusing to see a Scheme with many panels right up front, and then it being referred to repeatedly throughout the manuscript to explain different things.

2) Page 7:

a. How exactly the Dox loading was performed? And where is it expected to be located in the particle: at the surface, on the inside, or both? If Dox can also be exposed on the surface of the nanoparticles, then it can also react during the EDC coupling as it also contains a carboxylic acid. How can this affect the total efficiency of this system (Dox that is not exposed or reacts with EDC vs Dox that reacts with EDC and which efficacy can lower due to the modification)?

b. It is clear that there is a high crosslinking density of the BSA layer, as measured, but what is the pore size of this shell? Does it completely avoid leakage because the pores are smaller than the molecules or controls the leakage because of diffusion delay through a crowded network?

c. In the TEM pics of the PLGA+BSA layer (Figure 1, page 8), I can indeed see one particle with a layer, but the ones around it do not seem to contain such layer, so it will be nice to see a picture with more than one particle showing the layer. For a symmetrical system like this one, this should be visible in many particles using TEM. Also, for particle's layers/shells made with proteins, the TEM beam might be too destructive to give accurate structural information; CryoTEM characterization is preferred to get more details of the structure. In terms of BSA coating, and based on the pictures provided, how uniform is particle coating, are all particles coated?

d. Furthermore, in relation to size there does not seem to be an agreement between the TEM pictures and the DLS data, because I can see polydispersity in TEM, while DLS gives a sharp peak. Why is this? If the real sample actually has smaller and larger sizes, like the TEM pictures, then this could also affect the particle uptake mechanism and even display two different internalization pathways, just like the results observed by the authors, which will be dependent on the size.

e. Could you comment further as to why the colloidal stability of particles with BSA on the surface is higher than PEG coated systems? I would have thought that particles covered with proteins will be less stable due to both steric and electrostatic effects

f. The cell environment has high ionic strength, which is known to significantly affect both the zeta potential and the colloidal stability of particles. Even though the stability of the PLGA-BSA particles was measured in presence of proteins and 10 mM PBS, it will be also good to show how increasing ionic strength to values near those found in cells will affect the colloidal properties of this particular system.

3) Page 10:

a. First sentence of "Encapsulation Stability...", change "to assess the stability..." to "to assess the encapsulation stability...".

b. It is mentioned that "Cellular FRET results (Figure S7) were consistent with the solution experiments.", however different conditions are tested (BSA-PLGA with FRET pair and

PLGA with FRET pair versus BSA-PLGA FRET pair and BSA-PLGA no complete FRET pair), not the same exact particles, which does not represent consistency in results. Yes, this new data shows that the FRET pair or DiI is kept inside the particles when they are exposed to cells, and that is positive, but it does not show improvement of dye retention in BSA coated vs non-coated like the previous data. A better control would have been to have the DiI loaded particle exposed to cells with DiO in the surrounding solution, if there was leakage, because you need close proximity of the dyes to have FRET, you should see FRET signal, if there is no leakage, you should see little to no signal. Another control could have been using PLGA-FRET pair just like in the solution experiments.

c. The encapsulation stability experiments are done comparing PLGA-BSA and PLGA particles, but why not showing too results for PLGA-BSA non-crosslinked vs crosslinked? That will be more conclusive as to show the need for the crosslinking step.

4) Page 11:

a. Figure 3: why there is no release profile for extracellular cancer cell pH 6.0-6.5? This condition is shown for the ATRAM peptide but not for the release. It will be more consistent to show release in normal pH, pH outside cancer cell and pH inside cancer cell.

b. More explanation as to why the protonation of Dox-TPP leads to release at lower pH? Until this point in the manuscript, I only got the impression of higher release because of PLGA degradation, and not because of electrostatic repulsion of the drug with the carrier. This could be explained in the section related to Dox-TPP synthesis. I also guess that the conditions for encapsulation of Dox have to take this into account, and this is not mentioned. Also, the control (Rhodamine B) used is not non-pH-sensitive, it is pH sensitive but maybe in the range studied it stays the same. Rhodamine B is a cation at pH lower than 4, and a zwitterion at pH higher than 4, which means that for these experiments it was zwitterionic. However, there was not much release with this molecule, which is essentially charge-neutral, but good release with Paclitaxel, which is neutral and hydrophobic, after explaining the importance of the protonation step of Dox for release (besides PLGS degradation)? This is not clear. An explanation for no Rhodamine release vs Paclitaxel release is needed. Maybe even show with a negative drug what happens. Also, is loading efficiency of Rhodamine B and Paclitaxel comparable to Dox-TPP?

5) Page 12:

a. What is the pH of the solutions that contain GSH?

b. How GSH-BSA complexes can disrupt a covalently crosslinked shell, that is covalently attached to the particle? If the BSA was crosslinked via S-S bridges only, then I would understand this statement; but that is not the case, so how exactly that disruption occurs? Can this be proved by looking at NMR, IR of BSA crosslinked or not crosslinked in presence of GSH?

c. In the GSH experiments, how is it known that the PLGA is not damaged during GSH exposure, and is only BSA shell damage leading to release?

6) Page 14:

a. Figure 4: Why the cellular uptake was not shown with BSA-PLGA as control?

7) Page 19:

a. What is the release mechanism in the case of Paclitaxel?

8) Page 21:

a. The controls used for tumor inhibition, can also be used as control for the studies with different cancer cell lines, particularly Dox loaded BSA-PLGA nanoparticles.

Response to Reviewer #1

Palanikumar et al report here on the development of novel nanoparticles (NPs) for the targeted delivery of the cytotoxic agent doxorubicin to cancer cells. The NPs are based on FDA-approved PLGA, have a cross-linked BSA corona to avoid unspecific protein binding and opsonization and are decorated with a peptide that binds the cell membrane a low pHs as those in the tumor microenvironment. The results presented merit publication, especially given the demonstrated antitumor efficacy of the NPs in a mouse model in vivo, and will be very interesting for the field. However, a number of issues need to be answered by the authors before I can recommend publication.

Response: We thank the reviewer for the positive comments regarding the manuscript. We are grateful for the feedback provided by the reviewer and have worked diligently to address all of their concerns. In the following pages, we respond point-by-point to the reviewer's comments. We believe, and hope the reviewer agrees, that the paper is much improved in content and clarity.

Major Concerns

Concern #1. Quantitative proteomics analysis (Figure 2): it is not clear from the explanation in the text what the FBS and BSA controls consist of. Should FBS not contain a high amount of protein?

Response: We thank the reviewer for their careful reading of the manuscript! As the reviewer correctly points out, the FBS control should indeed contain a high amount of protein (compared to the BSA control). In preparing the figure, we had inadvertently switched the FBS and BSA labels on the heat maps for the control samples. This has now been corrected in the updated **Figure 2** (please see response to the reviewer's **Minor Concern #3**). The BSA control is 1 mg/mL in PBS, which is the solution we use to prepare the BSA-PLGA NPs, while the FBS control is 10% in cell culture medium (our standard solution for culturing the cells). This information has now been added to the updated **Figure 2** legend and included in the **Experimental Section**:

Supporting Information; Experimental Section; Quantitative Proteomic Analysis:

“In order to investigate the interaction of serum proteins with PLGA and BSA-PLGA NPs, 1 mg/mL of the NPs were incubated in 10 mM phosphate buffer or cell culture medium containing 10% FBS for 72 h. Thereafter, the serum proteins that adsorbed to the NPs were isolated by centrifugation following published methods.(Ali et al. 2014; Oh et al. 2018) Finally, the isolated serum proteins were de-N-glycosylated and analyzed by mass spectrometry (MS).(Ali et al. 2014; Oh et al. 2018) **BSA (1 mg/mL in PBS) and FBS (10% in cell culture medium) were used as controls”.**

Concern #2. Cellular internalization studies have only been reported at one time point (1 hour after exposure). The authors report that in such time frame, internalization is significantly higher at low pH. However, have the authors performed any tests at later time points? Most NPs may require longer times to interact and internalize and in fact 1 hour is a very short time. Internalization at higher pHs could be underestimated simply because of this short time of interaction.

Response: As suggested by the reviewer, we have carried out additional confocal microscopy and flow cytometry controls experiments to assess the cellular internalization of Dox-TPP loaded ATRAM-BSA-

PLGA NPs in MCF-7 cells at a later time point (**Figure 4d–f**). As expected, increasing the incubation time from 1 h (original measurement) to 4 h (additional time point), resulted in greater cellular uptake of the NPs. Consistent with the original measurement at 1 h, at 4 h the amount of internalized ATRAM-BSA-PLGA NPs in MCF-7 cells at pH 6.5 was far higher than that at pH 7.4. Indeed, at 1 h incubation internalization at pH 6.5 was > 5-fold higher than at pH 7.4, whereas after 4 h the difference was even more pronounced (amount of NPs internalized at pH 6.5 was > 7-fold higher than that at pH 7.4). This confirms our original conclusion that the pH-dependent membrane insertion of ATRAM facilitates uptake of the coupled NPs preferentially in cells that reside within a mildly acidic environment (such as that of solid tumors), and the new data extends the efficacy and time range where the desired effect is expected. Our results are in agreement with previous reports of enhanced cellular internalization of pH responsive drug delivery systems at acidic pH compared to physiological pH at all measurement time points (*Biochemistry* **2015**, *54*, 6567-6575; *Scientific Reports* **2017**, *7*, 46540).

Concern #3. Figure 4 (e,d): the authors affirm that the higher uptake at low pHs is caused by the ATRAM peptide, which would be explained by a higher affinity to the cell membrane. However, in order to be able to affirm that, a control of the same NP without the peptide is needed.

Response: To address the reviewer's concern, we have performed additional control experiments. Cellular uptake in MCF-7 cells of Dox-TPP loaded BSA-PLGA NPs (without ATRAM) was assessed using confocal fluorescence microscopy (**Figure S12**) and flow cytometry (**Figure 4e,f**). In the absence of the ATRAM peptide, poor uptake of BSA-PLGA NPs was observed at both pHs (7.4 and 6.5) and incubation times (1 and 4 h). This confirms that the pH-dependent membrane insertion of ATRAM facilitates uptake of the coupled NPs into cancer cells at low pH.

The results of the new control experiments suggested by the reviewer's **Major Concerns #2 and #3** are presented in the new **Figure S12** and updated **Figure 4d–f**.

Figure S12. Confocal laser scanning microscopy images of MCF-7 cells incubated with Dox-TPP loaded BSA-PLGA NPs for 1 h at pH 7.4 (*top panels*) and pH 6.5 (*lower panels*). Scale bar = 10 μm.

Figure 4. Cellular uptake of ATRAM-BSA-PLGA NPs is strongly pH-dependent. **(a)** Size analysis for ATRAM-BSA-PLGA NPs in aqueous solution. **(b,c)** Zeta potential measurements of BSA-PLGA **(b)** and ATRAM-BSA-PLGA **(c)** NPs in aqueous solution. **(d)** Confocal laser scanning microscopy images of MCF-7 cells incubated for 1 or 4 h with Dox-TPP loaded ATRAM-BSA-PLGA NPs at physiological (*top panels*) or acidic (*lower panels*) pH. Quantification of colocalization of the Dox-TPP cargo with mitochondria using Pearson's correlation coefficient (r ; *right panels*). Scale bar = 10 μm . **(e,f)** Flow cytometry analysis of cellular uptake of Dox-TPP loaded BSA-PLGA and ATRAM-BSA-PLGA NPs in MCF-7 cells: **(e)** plot of side scatter (SSC) vs fluorescence signal for MCF-7 cells that were either untreated (ctrl), or treated with Dox-TPP loaded BSA-PLGA (*left panel*) or ATRAM-BSA-PLGA (*right panel*) NPs for 1 or 4 h at pH 7.4 or 6.5; **(f)** quantification of cellular uptake of BSA-PLGA and ATRAM-BSA-PLGA NPs at different incubation times and pHs from the data in **(e)**. *** $P < 0.001$ compared with NPs at pH 7.4.

In addressing the reviewer's **Major Concerns #2 and #3**, we have modified the relevant section of the **Revised Manuscript** as follows:

Results and Discussion; Cellular Uptake of ATRAM-BSA-PLGA NPs:

"The uptake of ATRAM-BSA-PLGA NPs in cancer cells was assessed using confocal fluorescence microscopy. Human breast cancer MCF-7 cells were incubated with Dox-TPP

loaded ATRAM-BSA-PLGA NPs for 1 and 4 h at 37 °C (Figure 4d). Substantially higher cellular internalization of the Dox-TPP cargo was observed at pH 6.5 compared to pH 7.4 at both 1 and 4 h incubation times. Moreover, the greater amount of intracellular Dox-TPP at acidic pH showed strong colocalization with mitochondria. Similarly, in human cervical cancer HeLa cells we observed a marked enhancement in cellular uptake, and mitochondrial localization, of the Dox-TPP cargo of ATRAM-BSA-PLGA NPs at acidic pH compared to physiological pH (Figure S11). Quantification of cellular uptake using flow cytometry confirmed the confocal microscopy results, with higher cellular internalization of ATRAM-BSA-PLGA NPs in MCF-7 cells measured at acidic pH relative to physiological pH (amount of internalized NPs was > 5- and > 7-fold higher at pH 6.5 compared to pH 7.4 at 1 and 4 h incubation, respectively) (Figure 4e,f). As a control, cellular uptake of Dox-TPP loaded BSA-PLGA NPs in MCF-7 cells was assessed using confocal fluorescence microscopy (Figure S12) and flow cytometry (Figure 4e,f). In the absence of the ATRAM peptide, poor uptake of the NPs was observed at both pHs (7.4 and 6.5) and incubation times (1 and 4 h). Thus, the pH-dependent membrane insertion of ATRAM facilitates cellular internalization of the coupled NPs preferentially in cells that reside within a mildly acidic environment.”

Supporting Information; Experimental Section; Intracellular Imaging and Colocalization:

“Cells (MCF-7, HeLa or differentiated THP-1) were seeded at a density of 2×10^5 cells/well in 500 μ L complete medium in 4-chambered 35 mm glass bottom Cellview cell culture dishes (Greiner Bio-One, Monroe, NC). After culturing for 24 h, the medium was replaced fresh medium (pH 7.4 or 6.5) containing 2.5 μ g/mL Dox-TPP loaded PLGA, BSA-PLGA or ATRAM-BSA-PLGA NPs and incubated for 1–4 h. For some experiments, MCF-7 cells were pre-incubated for 1 h at 4 °C in serum-free DMEM, pre-treated for 1 h at 37 °C with 10 mM sodium azide/6 mM 2-deoxy-D-glucose in serum- and glucose-free DMEM, or pretreated for 30 min at 37 °C with the following drugs in serum-free DMEM: 10 μ M chlorpromazine; 5 mM methyl- β -cyclodextrin; 5 μ M filipin; or 5 μ M amiloride. After addition of the NPs, the cells were maintained for 1 h at 4 °C or in presence of inhibitors at 37 °C. 30 min prior to imaging, the medium was replaced with fresh medium containing 50 nM MitoTracker Green or vehicle. Finally, immediately prior to imaging, the medium was once again replaced with fresh medium to remove any extracellular markers. Imaging was done on an Olympus Fluoview FV-1000 confocal laser scanning microscope, using a 63 \times Plan-Apo/1.3 NA oil immersion objective with DIC capability. Image processing was done using the Fiji image processing software.(Schindelin et al. 2012)”

Supporting Information; Experimental Section; Quantification of Cellular Uptake:

“Cellular uptake of Dox-TPP loaded ATRAM-BSA-PLGA NPs at pH 7.4 or 6.5 was measured using flow cytometry. Cells (MCF-7 or differentiated THP-1) were seeded at a density of 2×10^4 cells/well in 500 μ L complete medium in 24-well plates. After culturing for 24 h, the cells were washed with PBS at 37 °C and the medium was replaced with fresh medium (pH 7.4 or 6.5) containing 2.5 μ g/mL Dox-TPP loaded PLGA, BSA-PLGA or ATRAM-BSA-PLGA NPs and incubated for 1–4 h at 37 °C. Subsequently, the cells were washed three times with ice-cold PBS to remove the extracellular NPs, and then treated with trypsin-EDTA for 5 min to detach the cells and remove cell surface-bound peptide. Finally, the cells were centrifuged (1,000 \times g for 5 min at 4 °C) and re-suspended in 500 μ L ice-cold PBS with 10% FBS. Data collection (10,000 cells/sample, gated on live cells by forward/side scatter and propidium iodide (PI) exclusion) was done immediately afterwards on a BD FACSAria III cell sorter (BD Biosciences, San Jose, CA), and analysis was performed using the BD FACSDiva software.”

Concern #4. Cytotoxic effects of NPs (Figure 6a). Cytotoxicity of ATRAM-BSA-PLGA NPs loaded with dox is demonstrated. However, the empty NP controls utilized here are not correct, since they do not have the ATRAM peptide which enhances internalization. While previous studies indicate that this peptide is not cytotoxic, the increased NP uptake that it induces could lead to cytotoxic events. A control of empty ATRAM-BSA-PLGA NPs should be included.

Response: As suggested by the reviewers, we have carried out additional control experiments. MCF-7 cells were incubated with empty ATRAM-BSA-PLGA NPs (i.e. without Dox-TPP) for 48 h at pH 7.4 (Figure 6a) or 6.5 (Figure S14), and cell viability was assessed using the MTS assay. Treatment with drug-free ATRAM-BSA-PLGA NPs up to a concentration of 75 $\mu\text{g}/\text{mL}$ did not result in a significant loss of MCF-7 cell viability. These results are consistent with the previously reported lack of toxicity of the ATRAM peptide at either physiological or acidic pH (*Biochemistry* 2015, 54, 6567-6575), and strongly suggest that the designed ATRAM-BSA-PLGA NPs are nontoxic and compatible for use as a pH-responsive drug delivery system.

The results of the new control experiments suggested by the reviewer are presented in the updated Figure 6a and new Figure S14:

Figure 6. Mechanism of pH-dependent cytotoxicity of Dox-TPP loaded ATRAM-BSA-PLGA NPs. **(a)** Cell viability of MCF-7 cells treated with PLGA, BSA-PLGA or ATRAM-BSA-PLGA NPs (all without Dox-TPP) for 48 h. **(b-f)** Effect of Dox-TPP loaded ATRAM-BSA-PLGA NPs on viability MCF-7 (b), HeLa (c), MIA PaCa-2 (d), Neuro-2a (e), and 4T1 (f) cancer cells after 48 h incubation at pH 7.4 or 6.5. Cell viability in (a-f) was assessed using the MTS assay, with the % viability determined from the ratio of the absorbance of the treated cells to the control cells. * $P < 0.05$, ** $P < 0.01$ compared with pH 7.4. **(g)** Flow cytometry analysis of annexin V/propidium iodide (PI) staining of MCF-7 cells that were either untreated (control; left panel), or treated with ATRAM-BSA-PLGA NPs loaded with 0.5 $\mu\text{g}/\text{mL}$ Dox-TPP for 12 h at pH 6.5 (right panel). The bottom left quadrant (annexin V-/PI-) represents live cells; bottom right (annexin V+/PI-), early apoptotic cells; top right (annexin V+/PI+), late apoptotic cells; and top left (annexin V-/PI+), necrotic cells. **(h)** A summary of the incidence of early/late apoptosis and necrosis in the MCF-7 cells treated with Dox-TPP loaded ATRAM-BSA-PLGA NPs determined from the flow cytometry analysis of annexin V/PI staining in (g). * $P < 0.05$ compared with controls.

Figure S14. Cell viability of MCF-7 cells treated with drug-free ATRAM-BSA-PLGA NPs for 48 h at pH 6.5. Cell viability was assessed using the MTS assay, with the % viability determined from the ratio of the absorbance of the treated cells to the control cells. ns, non-significant ($P > 0.05$).

In addressing the reviewer's concern, we have modified the relevant sections of the **Revised Manuscript** as follows:

Results and discussion; Cytotoxic Effects of Drug-Loaded ATRAM-BSA-PLGA NPs:

“The effectiveness of ATRAM-BSA-PLGA NPs in delivering chemotherapeutics was assessed using the MTS assay. PLGA NPs without drug did not have a significant adverse effect on MCF-7 cell viability (Figure 6a). This is in agreement with the reported biocompatibility of PLGA, which is approved by the FDA for use in drug formulations. (Rezvantlab et al. 2018) Likewise, treatment with drug-free BSA-PLGA up to a reasonably high concentration of 75 µg/mL did not result in a significant loss of MCF-7 cell viability. Furthermore, drug-free ATRAM-BSA-PLGA NPs did not exhibit significant cytotoxicity up to a concentration of 75 µg/mL at pH 7.4 (Figure 6a) or 6.5 (Figure S14), which is consistent with the previously reported lack of toxicity of the ATRAM peptide at either physiological or acidic pH. (Wyatt et al. 2018) Taken together, the cell viability results suggest that the designed ATRAM-BSA-PLGA NPs are nontoxic and compatible for use as a pH-responsive drug delivery system.”

Supporting Information; Experimental Section; Cell Viability/Toxicity Assays:

“Cells were seeded at a density of 2×10^4 cells/well in 100 µL complete medium in standard 96-well plates. After culturing for 24 h, the medium was replaced with fresh medium (pH 7.4 or 6.5) containing 2.5–75 µg/mL drug-free NPs, or NPs loaded with 0.05–2 µg/mL Dox-TPP or paclitaxel (PTX), and incubated for 48 h at 37 °C. Thereafter, the medium was replaced with fresh medium, and 20 µL MTS reagent was added to each well. The MTS reagent was incubated for 4 h at 37 °C, and absorbance of the soluble formazan product ($\lambda = 490$ nm) of MTS reduction was measured on a BioTek Synergy H1MF Multi-Mode Microplate-Reader, with a reference wavelength of 650 nm to subtract background. Wells treated with peptide-free carrier were used as control, and wells with medium alone served as a blank. MTS reduction was determined from the ratio of the absorbance of the treated wells to the control wells.”

Concern #5. NP uptake in macrophages (Figure 7). At which pH were these experiments performed? A condition at low pH should be included, because it is possible that at the lower pH of the tumor microenvironment the NPs are taken up by resident macrophages, which could be a limitation or an opportunity depending on the drug delivered, but in any case should be addressed.

Response: The cellular uptake and cytotoxicity of Dox-TPP loaded NPs in macrophages (differentiated human monocytic leukemia THP-1 cells) experiments were done at pH 7.4. We have attempted to do these experiments at pH 6.5 as suggested by the reviewer. However, efforts to culture THP-1 cells at pH

6.5 were largely unsuccessful. Examination of the morphology of the cells under the microscope revealed extensive damage/death of the cells at low pH. Moreover, the MTS response of these cells showed significant variability between experiments (and even between different control samples/wells in the same experiment), reflecting the uncontrolled effects of low pH on the cells. Consequently, we decided not to include these experiments as any data generated with these damaged cells would be highly unreliable and, likely, irreproducible. However, since the vast majority of immune cells (including macrophages) that the NPs will encounter will be in circulation (i.e. at physiological pH), we believe that the data presented in **Figure 7** represents the most relevant condition for assessing the tumor-targeting capabilities of the ATRAM-BSA-PLGA NPs.

Concern #6. Figure 8, antitumor efficacy in vivo: as in the in vitro cytotoxicity studies, a control with the exact same drug-free NP (including the peptide) should be desirable.

Response: To address the reviewer's concern, we performed additional *in vivo* control studies. 4T1 tumor-bearing mice were injected intravenously (once every 3 days, for a total of 10 doses) with drug-free ATRAM-BSA-PLGA NPs (115 mg/kg). Body weight and tumor volume were recorded every 2 days. Comparable increase in tumor volume was observed for the control (saline) and drug-free ATRAM-BSA-PLGA NP treated (**Figure S18**) groups. These results clearly show that, similar to the drug-free BSA-PLGA NPs (**Figure 8c,d**), drug-free ATRAM-BSA-PLGA NPs did not affect tumor growth (**Figure S16**).

The results of the additional control experiment are presented in the new **Figure S18**:

Figure S18. Effects of drug-free ATRAM-BSA-PLGA NPs on tumor volume and body weight of 4T1 tumor-bearing mice. ns, non-significant ($P > 0.05$).

The relevant sections of the **Revised Manuscript** now read as follows:

Results and discussion; *In Vivo* Tumor Inhibition by Dox-TPP Loaded ATRAM-BSA-PLGA NPs:

“In order to assess the antitumor efficacy of the Dox-TPP loaded ATRAM-BSA-PLGA NPs, 4T1 tumor-bearing mice were randomized into sex treatment groups, which were injected intravenously with saline, drug-free BSA-PLGA or ATRAM-BSA-PLGA NPs, free Dox or Dox-TPP loaded BSA-PLGA or ATRAM-BSA-PLGA NPs, over a period of 30 days (Figure 8b). As expected, drug-free BSA-PLGA and ATRAM-BSA-PLGA NPs did not affect tumor growth (Figures 8c,d and S18a). Free Dox lead to a moderate slowing of tumor growth, with the tumor volume and weight following treatment ~78 and 55%, respectively, those of the saline treated controls (Figure 8c,d). A more pronounced effect was observed with the Dox-TPP loaded BSA-PLGA NPs (tumor volume and weight after treatment were ~60 and 38%, respectively, those of controls) (Figure 8c,d). However, the most effective treatment was the Dox-TPP loaded ATRAM-

BSA-PLGA NPs, which completely inhibited tumor growth. Indeed, Dox-TPP loaded ATRAM-BSA-PLGA NPs reduced the tumors from an initial volume of $26 \pm 2 \text{ mm}^3$ to $23 \pm 2 \text{ mm}^3$ after treatment (Figure 8c), while the tumor mass was ~24% that of the saline treated controls (Figure 8d). Histological analysis of tumor tissues using hematoxylin and eosin (H&E) staining confirmed the enhanced antitumor efficacy of the Dox-TPP ATRAM-BSA-PLGA NPs compared to the other treatment group (Figure 8e). **Importantly, treatment with Dox-TPP loaded ATRAM-BSA-PLGA NPs substantially prolonged survival compared to free Dox (Figure 8f) over the 90-day duration of the experiment.**"

Supporting Information; Experimental Section; *In Vivo* Tumor Inhibition Studies:

"Once the tumor volume reached $\sim 25 \text{ mm}^3$, the mice were randomized into six treatment groups ($n = 8$ per group), which were injected intravenously (once every 3 days, for a total of 10 doses) with: (1) saline; (2) drug-free BSA-PLGA NPs (115 mg/kg); (3) drug-free ATRAM-BSA-PLGA NPs (115 mg/kg); (4) free Dox (2 mg/kg); (5) Dox-TPP loaded BSA-PLGA NPs (115 mg/kg, with the loading capacity of Dox-TPP at 1.8 wt%); (6) Dox-TPP loaded ATRAM-BSA-PLGA NPs (115 mg/kg, with the loading capacity of Dox-TPP at 1.8 wt%). The dose of doxorubicin administered was based on previously published work.(Palanikumar et al. 2018) Body weight and tumor volume were recorded every 2 days, and survival ($n = 4$ per group) was monitored for a total of 90 days."

Minor concerns

Question 1. The z potential values reported for the different NP formulations do not align with colloidal stable NPs (too close to 0). Can the authors comment on how with such values the NPs are still stable for a period of 72 hours.

Response: The zeta potential of the PLGA NPs changed from -28 mV to -16 mV after conjugation with BSA (to form the non-crosslinked BSA-PLGA NPs; **Figure 1e**). Crosslinking of the BSA shell did not alter the zeta potential significantly (**Figure 4b**). However, conjugation of the ATRAM peptide to the BSA-PLGA NPs (to generate the pH-responsive ATRAM-BSA-PLGA NPs) changed the zeta potential of the NPs to -5 mV at pH 7.4 (**Figure 4c**). The ATRAM-BSA-PLGA NPs have a reasonable negative surface charge at physiological pH, as this is within the range of zeta potential values reported for other highly stable NPs.(Oh et al. 2018; Palanikumar et al. 2017). At pH 6.5, protonation of the glutamic acid residues of ATRAM (which drives the pH-responsiveness of peptides), increases the zeta potential of the ATRAM-BSA-PLGA NPs further to +10 mV (**Figure 4c**). This promotes efficient internalization by cancer cells within the acidic tumor microenvironment.(Rosenblum et al. 2018; Palanikumar et al. 2017)

The crosslinked BSA-PLGA NPs showed good colloidal stability in 10 mM phosphate buffer solution (pH 7.4) and cell culture medium containing 10% fetal bovine serum (FBS) over 72 h (**Figures 1f,g**). The NPs also maintained a similar surface charge in cell culture medium containing 10% FBS (**Figure S5c**).

We have carried out additional control experiments in which we monitored the size of the crosslinked BSA-PLGA NPs under conditions of high serum content and high ionic strength, both of which can affect the colloidal stability of nanocarriers. The NPs were stable in the presence of high serum content (50% FBS; **Figure S6a**), which suggests that the NPs are able to maintain a reasonable size in circulation that will allow them to localize to tumors and then readily internalize into cancer cells to deliver their chemotherapeutic cargo. The NPs were also stable in the presence of high ionic strength (1×PBS

[137 mM NaCl, 2.7 mM KCl, 8 mM Na₂HPO₄, and 2 mM KH₂PO₄] with added 5–50 mM NaCl; **Figure S6b**), which indicates that the NPs will remain stable and retain their chemotherapeutic cargo in cellular milieu until exposure to an appropriate stimulus (e.g. low pH). This high stability of the NPs is likely due to thermodynamically favorable interactions between specific domains of BSA with the surface of the PLGA NPs, which are reinforced by crosslinking of the shell, leading to colloidal stabilization and acquisition of stealth properties (i.e. prevention of serum protein adsorption).(Schöttler et al. 2016; Oh et al. 2018; Treuel et al. 2014; Mortimer et al. 2014)

In addressing the reviewer's concern, the zeta potential values (along with the hydrodynamic radii) of the different NP formulations used in this study are now presented in a table in the **Revised Supporting Information** (please see response to reviewer's **Minor Concern #2**).

The relevant sections of the **Revised Manuscript** have been modified accordingly and read as follows:

Results and Discussion; Preparation and Characterization of BSA-PLGA Nanoparticles (NPs):

- "BSA was enriched over the PLGA surface, forming a corona with a thickness of 10 nm and increasing the hydrodynamic diameter of the NPs from 105 nm to 130 nm (**Figure 1c, Table S2**). The zeta potential of the PLGA NPs changed from -28 mV to -16 mV after conjugation with BSA (**Figure 1e, Table S2**)."
- "The BSA-PLGA NPs showed good colloidal stability in 10 mM phosphate buffer solution (pH 7.4) and cell culture medium containing 10% fetal bovine serum (FBS) over 72 h (**Figure 1f,g**). Moreover, the BSA-PLGA NPs maintained a similar surface charge in cell culture medium containing 10% FBS (**Figure S5c**). Next, we monitored the size of the BSA-PLGA NPs under conditions of high serum content and high ionic strength, both of which are reported to affect the colloidal stability of nanocarriers.(L. Moore et al. 2015) **Increasing the serum content to 50% resulted in a modest increase of ~30 nm in the hydrodynamic diameter of the BSA-PLGA NPs (Figure S6a) relative to that in 10% FBS, suggesting that the NPs are able to maintain a reasonable size in circulation that will allow them to localize to tumors and then readily internalize into cancer cells to deliver their chemotherapeutic cargo. Likewise, increasing the ionic strength to the levels found in cells and higher (1× PBS [137 mM NaCl, 2.7 mM KCl, 8 mM Na₂HPO₄, and 2 mM KH₂PO₄] with added 5–50 mM NaCl)(Gao et al. 2012) did not significantly alter the size of the BSA-PLGA NPs (Figure S6b), indicating that the NPs will remain stable and retain their chemotherapeutic cargo in cellular milieu until exposure to an appropriate stimulus (e.g. low pH). Taken together, these results show that the crosslinked BSA shell confers a high degree of stability on the NPs. This is likely due to thermodynamically favorable interactions between specific domains of BSA with the surface of the PLGA NPs, which are reinforced by crosslinking of the shell, leading to colloidal stabilization and acquisition of stealth properties (i.e. prevention of serum protein adsorption).(Schöttler et al. 2016; Oh et al. 2018; Treuel et al. 2014; Mortimer et al. 2014)"**

Results and Discussion; Conjugation of pH-Responsive ATRAM Peptide to BSA-PLGA NPs:

"Conjugation to ATRAM was further confirmed by change in the zeta potential of the NPs at pH 7.4 from -16 mV (for crosslinked BSA-PLGA) to -5 mV (for ATRAM-BSA-PLGA) (**Figure 4b,c and Table S2**), which is within the range of zeta potential values reported for other highly stable NPs at physiological pH.(Oh et al. 2018; Palanikumar et al. 2017) At pH 6.5, due to protonation of the glutamic acid residues that drives the pH-responsiveness of ATRAM, the zeta potential increased further to +10 mV, suggesting that the ATRAM-BSA-PLGA NPs would be efficiently

internalized by cancer cells within the acidic tumor microenvironment.(Rosenblum *et al.* 2018; Palanikumar *et al.* 2017)”

Concern #2. Size values from DLS experiments should also be reported on a table format. It is difficult to appreciate the specific values from the graphs.

Response: In addressing the reviewer’s **Minor Concerns #1** and **#2**, the hydrodynamic diameters and zeta potentials of the different nanoparticle formulations used in this study are now reported in the new **Table S2** of the **Revised Supporting Information**:

Table S2. Diameters and zeta potentials of PLGA, BSA-PLGA (with and without crosslinking) and ATRAM-BSA-PLGA NPs nanoparticles.

Nanoparticle	Diameter (nm)	Zeta Potential
PLGA	105 ± 2	- 28
BSA-PLGA (without crosslinking)	135 ± 4	- 16
Crosslinked BSA-PLGA	130 ± 4	- 16
ATRAM-BSA-PLGA	131 ± 3	-5 (pH 7.4) +10 (pH 6.5)

Concern #3. The color code utilized in the heatmaps displayed in Figure 2 are not color-blind friendly. This should be changed to ensure all readers can interpret the data.

Response: We thank the reviewer for pointing this out! A color-blind friendly color code is now used for the heat maps of serum proteins identified in the control (FBS and BSA) samples and adsorbed to the NPs in the updated **Figure 2**:

Figure 2. Quantitative proteomic analysis of serum protein adsorption to the NPs. **(a)** Heat map representation of identified serum proteins in the control BSA (1 mg/mL solution) and FBS (10% in cell culture medium) samples. **(b)** Heat map representation of serum proteins adsorbed to the NPs under the following conditions: PLGA (1) and BSA-PLGA NPs (3) incubated in 10 mM phosphate buffer for 72 h; PLGA (2) and BSA-PLGA NPs (4) incubated in cell culture medium containing 10% FBS for 72 h.

The digests for both the control and NP samples were analyzed by liquid chromatography tandem mass spectrometry (LC-MS/MS) and protein abundance was determined using the label-free quantification (LFQ) intensities. As shown in the color scale bar, the purple and yellow (gold) colors indicate high and low LFQ intensities (\log_2 (LFQ)), (Tyanova, Temu, and Cox 2016) respectively, while dark brown indicates that the protein concentration is below the detection limit. The proteins corresponding to the UniProt Knowledgebase (UniProtKB) accession numbers shown in the figure are given in **Table S3.**”

Concern #4. The term Fluorescence Activated Cell Sorting (FACS) is used incorrectly here. To my understanding, the authors have performed flow cytometry, but have not sorted the cells.

Response: We thank the reviewer for bringing this oversight to our attention! As per the reviewer’s suggestion, we have changed “Fluorescence Activated Cell Sorting (FACS)” to “flow cytometry” throughout the **Revised Manuscript** and **Revised Supporting Information.**

Response to Reviewer #2

Palanikumar *et al.* present the generation of a BSA coated nanoparticle (NP) that contains a pH activated ATRAM peptide for enhancing delivery into solid tumors due the lower pH environment. The manuscript is well-written and exceptionally well-reasoned study on enhancing NP delivery of chemotherapeutics by incorporating multiple molecular design elements. Impressively, the introduction elaborately (and politely) lays out the failures of the NP field to harness the presumed EPR in humans (vs. rodent models where it is much more prevalent). Overall, the studies are well controlled and the conclusions supported by the primary data. I have several concerns listed below that, given the high caliber of this group, the authors should be able to readily address. In conclusion, this is an exciting study on enhancing delivery of NPs in vivo that after addressing most of my concerns below should greatly exceed the bar for publication in CB.

Response: We thank the reviewer for the positive comments regarding the manuscript. We are grateful for the feedback provided by the reviewer and have worked diligently to address all of their concerns. In the following pages, we respond point-by-point to the reviewer's comments. We believe, and hope the reviewer agrees, that the paper is much improved in content and clarity.

Concern #1. Fig. 1e/f. The NPs become quite large when exposed to 10% FBS, likely ~200+ nm diameter, and that will dramatically negatively impact their delivery into solid tumors where they are swimming upstream against the tumor produced water flow. This is a significant concern with all NPs. However, to their credit, the authors are showing data that most NP groups do not. Question: Can you keep the size of the NPs to ~100 nm in serum, especially 100%?

Response: In addressing the reviewer's concern, we measured the size of the crosslinked BSA-PLGA NPs in solutions of increasing FBS content. The highest concentration we were able to reliably measure the hydrodynamic diameter of the NPs in was 50% serum. Beyond that, scattering from the sample made the measurements unreliable. We observed a modest increase of ~30 nm in the diameter of the NPs in 50% FBS (**Figure S6a**) relative to 10% FBS (**Figure 1g**). This suggests that in circulation, the NPs are able to maintain a reasonable size that allows them to localize to tumors and readily deliver their chemotherapeutic cargo to cancer cells. This is borne out by the potent antitumor effects of the drug-loaded NPs (**Figure 8**).

The results of the additional control experiments are presented in the new **Figure S6a**:

“Figure S6. Effects of high serum concentration and high ionic strength on colloidal stability of crosslinked BSA-PLGA NPs. Size analysis of BSA-PLGA NPs incubated for 72 h in: (a) cell culture medium containing 50% fetal bovine serum (FBS), or (b) 1×PBS (137 mM NaCl, 2.7 mM KCl, 8 mM Na₂HPO₄, and 2 mM KH₂PO₄) containing increasing concentrations of NaCl.”

The relevant paragraph of the **Results and Discussion** of the **Revised Manuscript** is as follows:

Results and Discussion; Preparation and Characterization of BSA-PLGA Nanoparticles (NPs):

“The BSA-PLGA NPs showed good colloidal stability in 10 mM phosphate buffer solution (pH 7.4) and cell culture medium containing 10% fetal bovine serum (FBS) over 72 h (Figure 1f,g). Moreover, the BSA-PLGA NPs maintained a similar surface charge in cell culture medium containing 10% FBS (Figure S5c). Next, we monitored the size of the BSA-PLGA NPs under conditions of high serum content and high ionic strength, both of which are reported to affect the colloidal stability of nanocarriers.(L. Moore et al. 2015). Increasing the serum content to 50% resulted in a modest increase of ~30 nm in the hydrodynamic diameter of the BSA-PLGA NPs (Figure S6a) relative to that in 10% FBS, suggesting that the NPs are able to maintain a reasonable size in circulation that will allow them to localize to tumors and then readily internalize into cancer cells to deliver their chemotherapeutic cargo. Likewise, increasing the ionic strength to the levels found in cells and higher (1× PBS [137 mM NaCl, 2.7 mM KCl, 8 mM Na₂HPO₄, and 2 mM KH₂PO₄] with added 5–50 mM NaCl)(Gao et al. 2012) did not significantly alter the size of the BSA-PLGA NPs (Figure S6b), indicating that the NPs will remain stable and retain their chemotherapeutic cargo in cellular milieu until exposure to an appropriate stimulus (e.g. low pH). Taken together, these results show that the crosslinked BSA shell confers a high degree of stability on the NPs. This is likely due to thermodynamically favorable interactions between specific domains of BSA with the surface of the PLGA NPs, which are reinforced by crosslinking of the shell, leading to colloidal stabilization and acquisition of stealth properties (i.e. prevention of serum protein adsorption).(Schöttler et al. 2016; Oh et al. 2018; Treuel et al. 2014; Mortimer et al. 2014)”

Concern #2. Fig. 3. The cross-linked BSA-PLGA NPs need to be added to panel 3a to have all of the NPs comparable on the same graph. Likewise, panel 3c needs the pH 5-0 control added here. Plus the authors need to add labels of " cross-linked BSA-PLGA NPs" to panels 3b/c/d. Panel 3d needs to no GSH control added.

Response: As suggested by the reviewer, drug release profile for crosslinked BSA-PLGA NPs was added to **Figure 3a**. In **Figure 3c**, we included the release profile at pH 4.0 (as rhodamine B is protonated and becomes cationic at this pH) as a control instead of pH 5.0. At pH 4.0, significant release of rhodamine B is observed. This supports our conclusion that release of the Dox-TPP cargo under acidic conditions likely occurs due to a combination of degradation of the PLGA core (due to hydrolysis of the ester bonds in the polymer chains), and dissociation of the now hydrophilic chemotherapeutic from the hydrophobic PLGA core. In addition, the label of ‘crosslinked BSA-PLGA NPs’ was added to **Figure 3b–d**.

The relevant sections of the **Revised Manuscript** and **Revised Supporting Information** are as follows:

Results and Discussion; Drug Release Profile of BSA-PLGA NPs:

“Since endocytosis is a common cellular uptake route for nanocarriers,(Palanikumar, Kim, et al. 2015) during their endocytic entry into cells NPs will be exposed to increasingly acidic environments as they are trafficked from weakly acidic early/maturing endosomes (pH 6.0–5.5) to more acidic late endosomes/lysosomes (pH 5.0–4.5).(Fennelly and Amaravadi 2017; Piao and Amaravadi 2016) Therefore, we probed the effects of an acidic environment on the release of Dox-TPP from crosslinked BSA-PLGA NPs (Figure 3b). At pH 6.5, negligible release of Dox-TPP was observed, indicating that the crosslinked BSA-PLGA NPs will effectively retain their

chemotherapeutic cargo within the mildly acidic microenvironment of malignant solid tumors (pH 6.5–6.9). (Palanikumar *et al.* 2017; Persi *et al.* 2018) Lowering the pH to within the range reported for early endosomes (pH 5.8) triggered the release of a significant amount of the Dox-TPP cargo ($45 \pm 5\%$ release at 24 h). The release under acidic conditions occurs due to degradation of the PLGA core (as a result of hydrolysis of the ester bonds in the polymer chains). (Rezvantab *et al.* 2018) Additionally, as the pH decreases, dissociation of the increasingly hydrophilic Dox-TPP from the hydrophobic PLGA core likely contributes to the acid-triggered release of the drug. (Liu *et al.* 2018; Palanikumar *et al.* 2017) Lowering the pH even further to that of late endosomes/lysosomes (pH 5.0), led to much higher release of Dox-TPP ($79 \pm 4\%$ at 24 h) (Figure 3b). As a control, we monitored release of the hydrophobic dye rhodamine B from crosslinked BSA-PLGA NPs (Figure 3c). Negligible release of rhodamine B was observed at pH 7.4 or 5.8. However, at pH 4.0, where rhodamine B becomes cationic due to protonation of its carboxylic acid group ($pK_a \sim 4.2$). (Yu *et al.* 2013) a significant amount of the dye is released from the NPs ($44 \pm 2\%$ release at 24 h). This supports the notion that protonation of the cargo contributes to its acid-triggered release. Thus, the acidic microenvironment of endocytic compartments is expected to facilitate the efficient release of the Dox-TPP cargo following internalization of the crosslinked BSA-PLGA NPs into cancer cells.”

Supporting Information; Experimental Section; Synthesis and Characterization of BSA-PLGA NPs:

“Cargo (Dox-TPP or rhodamine B) release from the NPs, in the absence or presence of a stimulus – e.g. pH or glutathione (GSH) – was monitored using the dialysis method. (Huang *et al.* 2018) Briefly, 1 mL of 200 $\mu\text{g}/\text{mL}$ Dox-TPP loaded NPs (PLGA, non-crosslinked BSA-PLGA or crosslinked BSA-PLGA) was placed in a dialysis bag (molecular-weight cutoff: 3 kDa), with or without GSH (0.5 or 10 mM), and fully submerged into 25 mL of release medium – 10 mM phosphate buffer (pH 7.4 or 6.5) or 10 mM acetate buffer (pH 5.8, 5.0 or 4.0) – followed by stirring at 100 rpm. At the designated time points, 1 mL of the release medium was removed for analysis, and replenished with the same volume of fresh buffer. Dox-TPP fluorescence in the release sample was measured ($\lambda_{\text{ex/em}} = 480/580 \text{ nm}$) and amount of Dox-TPP released was determined using a standard calibration curve.

The updated **Figure 3** of the **Revised Manuscript** is as follows:

“**Figure 3.** Drug release profiles of the BSA-PLGA NPs in the absence and presence of a stimulus. (a) Stimulus-free release from Dox-TPP loaded PLGA or BSA-PLGA NPs (without and with shell crosslinking) in 10 mM phosphate buffer (pH 7.4). (b,c) pH-triggered release from Dox-TPP loaded (b) and rhodamine B-loaded (c) BSA-PLGA NPs with shell crosslinking. (d) Glutathione (GSH) mediated Dox-TPP release from crosslinked BSA-PLGA NPs in 10 mM phosphate buffer (pH 7.4).”

Concern #3. Fig 4. ATRAM is a highly hydrophobic peptide and it is rather remarkable that the only two charged Glu residues in ATRAM have a pKa of 6.5 vs. the expected ~4.2. Beyond their prior publications, the authors need to comment and/or preferably add pKa data to support this point.

Response: The pKa of titratable residues is affected by the dielectric constant as well as ionic interactions. When ATRAM is in contact with the plasma membrane of a cancer cell, the peptide's environment is very different from bulk water, due to a large change in the dielectric constant. The pKa value the reviewer mentions is the average value observed for highly hydrated Glu side-chains. This environment is very different to the surface of the bilayer. In fact, it is commonly observed that the pKa of the Glu side-chain increases when in a hydrophobic environment. (Caputo and London 2004; Harms *et al.* 2009; Pace, Grimsley, and Scholtz 2009) This is the case for ATRAM, which explains the higher pKa.

To address the reviewer's concern, the relevant paragraph of the **Results and Discussion** of the **Revised Manuscript** has been modified and expanded as follows:

Results and Discussion; Conjugation of pH-Responsive ATRAM Peptide to BSA-PLGA NPs:

“In order to specifically target tumor cells, the BSA-PLGA NPs were functionalized with the acidity-triggered rational membrane (ATRAM) peptide. (Nguyen *et al.* 2015) The highly soluble ATRAM peptide interacts with lipid membranes in a pH-dependent manner: at physiological or basic pH, ATRAM binds weakly to the membrane surface in a largely unstructured conformation, whereas under acidic conditions protonation of ATRAM's glutamic acid residues increases the overall hydrophobicity of the peptide and leads to its insertion into lipid bilayers as a transmembrane α -helix. (Nguyen *et al.* 2015; Kalmouni, Al-Hosani, and Magzoub 2019) The pKa values of ionizable groups in proteins are highly dependent on the environment: for the glutamic acid side-chain, this value ranges from ~4.0 in solution to > 8.0 in hydrophobic environments. (Caputo and London 2004; Harms *et al.* 2009; Pace, Grimsley, and Scholtz 2009) Crucially, the measured membrane insertion pKa of ATRAM is 6.5, (Nguyen *et al.* 2015; 2019; Nguyen, Dixon, and Barrera 2019), making the peptide ideal for targeting cancer cells that reside within the acidic microenvironment of solid tumors, (Palanikumar *et al.* 2017; Persi *et al.* 2018) which results from the high glycolytic rate of tumor cells. (Cairns, Harris, and Mak 2011) Indeed, ATRAM was shown to efficiently target tumors in mice. (Wyatt *et al.* 2018)”

Concern #4. Fig 4b needs the zeta potential of the non-ATRAM conjugated control cross-linked BSA-PLGA NP.

Response: As per the reviewer's suggestion, the zeta potential of crosslinked BSA-PLGA NPs (without ATRAM) has been added to the updated **Figure 4**.

The relevant section of the **Results and Discussion** has been modified accordingly and now reads as follows:

Results and discussion; Conjugation of pH-Responsive ATRAM Peptide to BSA-PLGA NPs:

“Conjugation to ATRAM was further confirmed by change in the zeta potential of the NPs at pH 7.4 from -16 mV (for crosslinked BSA-PLGA) to -5 mV (for ATRAM-BSA-PLGA) (Figure 4b,c and Table S2), which is within the range of zeta potential values reported for other highly stable NPs at physiological pH. (Oh *et al.* 2018; Palanikumar *et al.* 2017) At pH 6.5, due to protonation

of the glutamic acid residues that drives the pH-responsiveness of ATRAM, the zeta potential increased further to +10 mV, suggesting that the ATRAM-BSA-PLGA NPs would be efficiently internalized by cancer cells within the acidic tumor microenvironment. (Rosenblum *et al.* 2018; Palanikumar *et al.* 2017)”

The updated **Figure 4** of the **Revised Manuscript** is now as follows:

“Figure 4. Cellular uptake of ATRAM-BSA-PLGA NPs is strongly pH-dependent. **(a)** Size analysis for ATRAM-BSA-PLGA NPs in aqueous solution. **(b,c)** Zeta potential measurements of BSA-PLGA **(b)** and ATRAM-BSA-PLGA **(c)** NPs in aqueous solution. **(d)** Confocal laser scanning microscopy images of MCF-7 cells incubated for 1 or 4 h with Dox-TPP loaded ATRAM-BSA-PLGA NPs at physiological (*top panels*) or acidic (*lower panels*) pH. Quantification of colocalization of the Dox-TPP cargo with mitochondria using Pearson’s correlation coefficient (*r*; *right panels*). Scale bar = 10 μm. **(e,f)** Flow cytometry analysis of cellular uptake of Dox-TPP loaded BSA-PLGA and ATRAM-BSA-PLGA NPs in MCF-7 cells: **(e)** plot of side scatter (SSC) vs fluorescence signal for MCF-7 cells that were either

untreated (ctrl), or treated with Dox-TPP loaded BSA-PLGA (*left panel*) or ATRAM-BSA-PLGA (*right panel*) NPs for 1 or 4 h at pH 7.4 or 6.5; (f) quantification of cellular uptake of BSA-PLGA and ATRAM-BSA-PLGA NPs at different incubation times and pHs from the data in (e). *** $P < 0.001$ compared with NPs at pH 7.4.”

Concern #5. Fig 6b-f. TO test for pH induced toxicity from the NP vs. solely from the DOX-TPP cargo, he authors need to repeat these experiments using the control of ATRAM conjugated cross-linked BSA-PLGA NPs that are not loaded with the DOX-TPP.

Response: As suggested by the reviewer, we have carried out additional control experiments. MCF-7 cells were incubated with empty ATRAM-BSA-PLGA NPs (i.e. without Dox-TPP) for 48 h at pH 7.4 (**Figure 6a**) or 6.5 (**Figure S14**), and cell viability was assessed using the MTS assay. Treatment with drug-free ATRAM-BSA-PLGA NPs up to a reasonably high concentration of 75 $\mu\text{g/mL}$ did not result in a significant loss of MCF-7 cell viability at either pH. These results are consistent with the previously reported lack of toxicity of the ATRAM peptide at either physiological or acidic pH (*Biochemistry* **2015**, *54*, 6567-6575), and strongly suggest that the designed ATRAM-BSA-PLGA NPs are nontoxic and compatible for use as a pH-responsive drug delivery system.

The results of the new control experiments suggested by the reviewer are presented in the updated **Figure 6a** and the new **Figure S14**.

“Figure 6. Mechanism of pH-dependent cytotoxicity of Dox-TPP loaded ATRAM-BSA-PLGA NPs. (a) Cell viability of MCF-7 cells treated with PLGA, BSA-PLGA or ATRAM-BSA-PLGA NPs (all without Dox-TPP) for 48 h. (b-f) Effect of Dox-TPP loaded ATRAM-BSA-PLGA NPs on viability MCF-7 (b), HeLa (c), MIA PaCa-2 (d), Neuro-2a (e), and 4T1 (f) cancer cells after 48 h incubation at pH 7.4 or 6.5. Cell viability in (a-f) was assessed using the MTS assay, with the % viability determined from the ratio of the absorbance of the treated cells to the control cells. * $P < 0.05$, ** $P < 0.01$ compared with pH 7.4.

(g) **Flow cytometry** analysis of annexin V/propidium iodide (PI) staining of MCF-7 cells that were either untreated (control; *left panel*), or treated with ATRAM-BSA-PLGA NPs loaded with 0.5 $\mu\text{g}/\text{mL}$ Dox-TPP for 12 h at pH 6.5 (*right panel*). The bottom left quadrant (annexin V-/PI-) represents live cells; bottom right (annexin V+/PI-), early apoptotic cells; top right (annexin V+/PI+), late apoptotic cells; and top left (annexin V-/PI+), necrotic cells. (h) A summary of the incidence of early/late apoptosis and necrosis in the MCF-7 cells treated with Dox-TPP loaded ATRAM-BSA-PLGA NPs determined from the **flow cytometry** analysis of annexin V/PI staining in (g). **P < 0.05 compared with controls.*”

“**Figure S14.** Cell viability of MCF-7 cells treated with drug-free ATRAM-BSA-PLGA NPs for 48 h at pH 6.5. Cell viability was assessed using the MTS assay, with the % viability determined from the ratio of the absorbance of the treated cells to the control cells. ns, non-significant ($P > 0.05$).”

In addressing the reviewer’s concern, we have modified the relevant sections of the **Revised Manuscript** and **Revised Supporting Information** as follows:

Results and discussion; Cytotoxic Effects of Drug-Loaded ATRAM-BSA-PLGA NPs:

“The effectiveness of ATRAM-BSA-PLGA NPs in delivering chemotherapeutics was assessed using the MTS assay. PLGA NPs without drug did not have a significant adverse effect on MCF-7 cell viability (Figure 6a). This is in agreement with the reported biocompatibility of PLGA, which is approved by the FDA for use in drug formulations.(Rezvantlab et al. 2018) Likewise, treatment with drug-free BSA-PLGA up to a reasonably high concentration of 75 $\mu\text{g}/\text{mL}$ did not result in a significant loss of MCF-7 cell viability. Furthermore, drug-free ATRAM-BSA-PLGA NPs did not exhibit significant cytotoxicity up to a concentration of 75 $\mu\text{g}/\text{mL}$ at pH 7.4 (Figure 6a) or 6.5 (Figure S14), which is consistent with the previously reported lack of toxicity of the ATRAM peptide at either physiological or acidic pH.(Wyatt et al. 2018) Taken together, the cell viability results suggest that the designed ATRAM-BSA-PLGA NPs are nontoxic and compatible for use as a pH-responsive drug delivery system.”

Supporting Information; Experimental Section; Cell Viability/Toxicity Assays:

“Cells were seeded at a density of 2×10^4 cells/well in 100 μL complete medium in standard 96-well plates. After culturing for 24 h, the medium was replaced with fresh medium (pH 7.4 or 6.5) containing 2.5–75 $\mu\text{g}/\text{mL}$ drug-free NPs, or NPs loaded with 0.05–2 $\mu\text{g}/\text{mL}$ Dox-TPP or paclitaxel (PTX), and incubated for 48 h at 37 $^{\circ}\text{C}$. Thereafter, the medium was replaced with fresh medium, and 20 μL MTS reagent was added to each well. The MTS reagent was incubated for 4 h at 37 $^{\circ}\text{C}$, and absorbance of the soluble formazan product ($\lambda = 490$ nm) of MTS reduction was measured on a BioTek Synergy H1MF Multi-Mode Microplate-Reader, with a reference wavelength of 650 nm to subtract background. Wells treated with peptide-free carrier were used as control, and wells with medium alone served as a blank. MTS reduction was determined from the ratio of the absorbance of the treated wells to the control wells.”

Concern #6. Fig 7. The control of cross-linked BSA-PLGA NP is missing and needs to be added into this figure.

Response: As per the reviewer's recommendation, new control experiments were performed to assess the uptake and cytotoxicity of Dox-TPP loaded crosslinked BSA-PLGA NPs in THP-1 cells. Whereas treatment of THP-1 cells with PLGA NPs led to accumulation of a significant amount of the Dox-TPP cargo intracellularly over time (**Figure 7a**), minimal intracellular Dox-TPP was observed in THP-1 cells treated with crosslinked BSA-PLGA (**Figure S17**) or ATRAM-BSA-PLGA (**Figure 7b**) NPs for the duration of the experiment. Quantification of cellular uptake using flow cytometry showed that internalization of crosslinked BSA-PLGA or ATRAM-BSA-PLGA NPs in THP-1 cells was ~5 fold lower compared to PLGA NPs (**Figure 7c,d**). Moreover, exposure to Dox-TPP loaded PLGA NPs induced considerable toxicity in THP-1 cells, while Dox-TPP loaded crosslinked BSA-PLGA or ATRAM-BSA-PLGA NPs did not have a significant effect on THP-1 cell viability (**Figure 7f**). Finally, in contrast to Dox-TPP loaded PLGA NPs, Dox-TPP loaded crosslinked BSA-PLGA and ATRAM-BSA-PLGA NPs induced production of negligible levels (< 5 pg/mL) of TNF- α and Il-1 β by THP-1 cells, demonstrating the lack of immunogenicity of the hybrid NPs. Taken together, our results strongly suggest that ATRAM-BSA-PLGA NPs are able to effectively evade recognition and uptake by macrophages (**Figure 7e**).

The results of the new control experiments suggested by the reviewer are presented in the updated **Figure 7** and new **Figure S17**:

“Figure 7. ATRAM-BSA-PLGA NPs evade uptake by differentiated human monocytic leukemia THP-1 cells. (a,b) Confocal laser scanning microscopy images of THP-1 cells incubated with Dox-TPP loaded

PLGA (a) or ATRAM-BSA-PLGA (b) NPs for 1–4 h at pH 7.4. Scale bar = 5 μm . (c,d) Flow cytometry analysis of cellular uptake of Dox-TPP loaded PLGA, crosslinked BSA-PLGA and ATRAM-BSA-PLGA NPs in THP-1 cells: (c) plot of side scatter (SSC) vs fluorescence signal for THP-1 cells that were either untreated (ctrl), or treated with Dox-TPP loaded PLGA, crosslinked BSA-PLGA or ATRAM-BSA-PLGA NPs for 2 h at pH 7.4; (d) quantification of cellular uptake of the NPs from the data in (c). (e) Schematic representation of sequestration of PLGA NPs, but not crosslinked BSA-PLGA or ATRAM-BSA-PLGA NPs, by macrophages (e.g. monocytes). (f) Cell viability of THP-1 cells treated with Dox-TPP loaded PLGA, crosslinked BSA-PLGA or ATRAM-BSA-PLGA NPs for 48 h. Cell viability was assessed using the MTS assay, with the % viability was determined from the ratio of the absorbance of the treated cells to the control cells. (g) Release of inflammatory cytokines, tumor necrosis factor-alpha (TNF- α) and interleukin-1 beta (IL-1 β), by THP-1 cells exposed to Dox-TPP loaded PLGA, crosslinked BSA-PLGA or ATRAM-BSA-PLGA NPs for 24 h. Cells treated with lipopolysaccharide (LPS) were used as a positive control for inflammation. TNF- α and IL-1 β levels in the culture medium were assayed using a commercial ELISA kit. * $P < 0.05$, ** $P < 0.01$ or non-significant (ns, $P > 0.05$) compared with controls.”

“**Figure S17.** Confocal laser scanning microscopy images of differentiated human monocytic leukemia THP-1 cells treated with Dox-TPP loaded crosslinked BSA-PLGA NPs for 1–4 h at pH 7.4. Scale bar = 10 μm .”

The relevant sections of the **Revised Manuscript** and **Revised Supporting Information** have been modified accordingly and read as follows:

Results and Discussion; Macrophage Recognition and Immunogenicity of ATRAM-BSA-PLGA NPs:

“Cellular uptake of the NPs in differentiated human monocytic leukemia THP-1 cells, a widely used model of monocyte/macrophage activation,(Chanput, Mes, and Wichers 2014) was assessed using confocal fluorescence microscopy. Treatment of THP-1 cells with naked PLGA NPs led to accumulation of a significant amount of the Dox-TPP cargo intracellularly over time, indicating that the PLGA NPs are readily taken up by macrophages (Figure 7a). In contrast, minimal intracellular Dox-TPP was observed for THP-1 cells treated with crosslinked BSA-PLGA (Figure S17) or ATRAM-BSA-PLGA (Figure 7b) NPs for the duration of the experiment. Quantification

of cellular uptake using flow cytometry showed that internalization of crosslinked BSA-PLGA or ATRAM-BSA-PLGA NPs in THP-1 cells was ~5 fold lower compared to PLGA NPs (Figure 7c,d). Moreover, exposure to Dox-TPP loaded PLGA NPs induced considerable toxicity in THP-1 cells, whereas Dox-TPP loaded crosslinked BSA-PLGA or ATRAM-BSA-PLGA NPs did not have a significant effect on THP-1 cell viability (Figure 7f).

To assess the inflammatory potential of ATRAM-BSA-PLGA NPs, we quantified the production of tumor necrosis factor-alpha (TNF- α) and interleukin-1 beta (Il-1 β) by differentiated human THP-1 monocytes exposed to the NPs (Figure 7g). TNF- α and Il-1 β are inflammatory cytokines primarily produced by macrophages/monocytes during acute inflammation.(Tsarouchas et al. 2018) The macrophage activator lipopolysaccharide (LPS)(Meng and Lowell 1997) was used as a positive control. Treatment of THP-1 cells with PLGA NPs loaded with 0.5 μ g/mL Dox-TPP resulted in production of ~39 and ~58 pg/mL of TNF- α and Il-1 β , respectively. This was comparable to the ~40 and ~60 pg/mL of TNF- α and Il-1 β , respectively, produced by THP-1 cells due to exposure to the positive control LPS. **In contrast, Dox-TPP loaded crosslinked BSA-PLGA and ATRAM-BSA-PLGA NPs induced production of negligible levels (< 5 pg/mL) of TNF- α and Il-1 β by THP-1 cells, which demonstrates the lack of immunogenicity of the hybrid NPs. Taken together, these results show that ATRAM-BSA-PLGA NPs are able to effectively evade recognition and uptake by macrophages (Figure 7e)."**

Supporting Information; Experimental Section; Intracellular Imaging and Colocalization:

"Cells (MCF-7, HeLa or differentiated THP-1) were seeded at a density of 2×10^5 cells/well in 500 μ L complete medium in 4-chambered 35 mm glass bottom Cellview cell culture dishes (Greiner Bio-One, Monroe, NC). **After culturing for 24 h, the medium was replaced fresh medium (pH 7.4 or 6.5) containing 2.5 μ g/mL Dox-TPP loaded PLGA, BSA-PLGA or ATRAM-BSA-PLGA NPs and incubated for 1–4 h.** For some experiments, MCF-7 cells were pre-incubated for 1 h at 4 $^{\circ}$ C in serum-free DMEM, pre-treated for 1 h at 37 $^{\circ}$ C with 10 mM sodium azide/6 mM 2-deoxy-D-glucose in serum- and glucose-free DMEM, or pretreated for 30 min at 37 $^{\circ}$ C with the following drugs in serum-free DMEM: 10 μ M chlorpromazine; 5 mM methyl- β -cyclodextrin; 5 μ M filipin; or 5 μ M amiloride. After addition of the NPs, the cells were maintained for 1 h at 4 $^{\circ}$ C or in presence of inhibitors at 37 $^{\circ}$ C. 30 min prior to imaging, the medium was replaced with fresh medium containing 50 nM MitoTracker Green or vehicle. Finally, immediately prior to imaging, the medium was once again replaced with fresh medium to remove any extracellular markers. Imaging was done on an Olympus Fluoview FV-1000 confocal laser scanning microscope, using a 63 \times Plan-Apo/1.3 NA oil immersion objective with DIC capability. Image processing was done using the Fiji image processing software.(Schindelin et al. 2012)"

Supporting Information; Experimental Section; Quantification of Cellular Uptake:

"Cellular uptake of Dox-TPP loaded ATRAM-BSA-PLGA NPs at pH 7.4 or 6.5 was measured using **flow cytometry**. Cells (MCF-7 or differentiated THP-1) were seeded at a density of 2×10^4 cells/well in 500 μ L complete medium in 24-well plates. **After culturing for 24 h, the cells were washed with PBS at 37 $^{\circ}$ C and the medium was replaced with fresh medium (pH 7.4 or 6.5) containing 2.5 μ g/mL Dox-TPP loaded PLGA, BSA-PLGA or ATRAM-BSA-PLGA NPs and incubated for 1–4 h at 37 $^{\circ}$ C.** Subsequently, the cells were washed three times with ice-cold PBS to remove the extracellular NPs, and then treated with trypsin-EDTA for 5 min to detach the cells and remove cell surface-bound peptide. Finally, the cells were centrifuged (1,000 \times g for 5 min at 4 $^{\circ}$ C) and re-suspended in 500 μ L ice-cold PBS with 10% FBS. Data collection (10,000 cells/sample, gated on live cells by forward/side scatter and propidium iodide (PI) exclusion) was done immediately afterwards on a BD FACSAria III cell sorter (BD Biosciences, San Jose, CA), and analysis was performed using the BD FACSDiva software."

Supporting Information; Experimental Section; Cell Viability/Toxicity Assays:

“Cells were seeded at a density of 2×10^4 cells/well in 100 μL complete medium in standard 96-well plates. After culturing for 24 h, the medium was replaced with fresh medium (pH 7.4 or 6.5) containing 2.5–75 $\mu\text{g/mL}$ drug-free NPs, or NPs loaded with 0.05–2 $\mu\text{g/mL}$ Dox-TPP or paclitaxel (PTX), and incubated for 48 h at 37 $^\circ\text{C}$. Thereafter, the medium was replaced with fresh medium, and 20 μL MTS reagent was added to each well. The MTS reagent was incubated for 4 h at 37 $^\circ\text{C}$, and absorbance of the soluble formazan product ($\lambda = 490 \text{ nm}$) of MTS reduction was measured on a BioTek Synergy H1MF Multi-Mode Microplate-Reader, with a reference wavelength of 650 nm to subtract background. Wells treated with peptide-free carrier were used as control, and wells with medium alone served as a blank. MTS reduction was determined from the ratio of the absorbance of the treated wells to the control wells.”

Supporting Information; Experimental Section; Inflammatory Cytokine Assay:

“Differentiated THP-1 cells were seeded at a density of 2×10^4 cells/well in 100 μL complete medium in standard 96-well plates. After culturing for 24 h, the medium was replaced with medium containing PLGA, BSA-PLGA or ATRAM-BSA-PLGA NPs loaded with 0.5 $\mu\text{g/mL}$ Dox-TPP and incubated for 24 h at 37 $^\circ\text{C}$. Cells treated with lipopolysaccharide (LPS) were used as a positive control, while untreated cells served as a negative control. Thereafter, the cell culture medium was assayed for secretion of the inflammatory cytokines, tumor necrosis factor-alpha (TNF- α) and interleukin-1 beta (IL-1 β), using commercial ELISA kits. The total TNF- α and IL-1 β levels were determined from the absorbance ($\lambda = 450 \text{ nm}$) measured on a BioTek Synergy H1MF Multi-Mode Microplate-Reader using a standard TNF- α concentration calibration curve.”

Concern #7. Fig 8f should be updated to the current length of the experiment since the manuscript was first submitted.

Response 7: As suggested by the reviewer, the survival analysis has been updated to reflect the current length of the experiment (90 days).

The relevant sections of the **Revised Manuscript** and **Revised Supporting Information** are as follows:

Results and Discussion; *In Vivo* Tumor Inhibition by Dox-TPP Loaded ATRAM-BSA-PLGA NPs:

“In order to assess the antitumor efficacy of the Dox-TPP loaded ATRAM-BSA-PLGA NPs, 4T1 tumor-bearing mice were randomized into sex treatment groups, which were injected intravenously with saline, drug-free BSA-PLGA or ATRAM-BSA-PLGA NPs, free Dox or Dox-TPP loaded BSA-PLGA or ATRAM-BSA-PLGA NPs, over a period of 30 days (Figure 8b). As expected, drug-free BSA-PLGA and ATRAM-BSA-PLGA NPs did not affect tumor growth (Figures 8c,d and S18a). Free Dox lead to a moderate slowing of tumor growth, with the tumor volume and weight following treatment ~ 78 and 55%, respectively, those of the saline treated controls (Figure 8c,d). A more pronounced effect was observed with the Dox-TPP loaded BSA-PLGA NPs (tumor volume and weight after treatment were ~ 60 and 38%, respectively, those of controls) (Figure 8c,d). However, the most effective treatment was the Dox-TPP loaded ATRAM-BSA-PLGA NPs, which completely inhibited tumor growth. Indeed, Dox-TPP loaded ATRAM-BSA-PLGA NPs reduced the tumors from an initial volume of $26 \pm 2 \text{ mm}^3$ to $23 \pm 2 \text{ mm}^3$ after treatment (Figure 8c), while the tumor mass was $\sim 24\%$ that of the saline treated controls (Figure 8d). Histological analysis of tumor tissues using hematoxylin and eosin (H&E) staining confirmed the enhanced antitumor efficacy of the Dox-TPP ATRAM-BSA-PLGA NPs compared to the other treatment group (Figure 8e). **Importantly, treatment with Dox-TPP loaded ATRAM-**

BSA-PLGA NPs substantially prolonged survival compared to free Dox (Figure 8f) over the 90-day duration of the experiment.”

Supporting Information; Experimental Section; *In Vivo* Tumor Inhibition Studies:

“Once the tumor volume reached $\sim 25 \text{ mm}^3$, the mice were randomized into six treatment groups ($n = 8$ per group), which were injected intravenously (once every 3 days, for a total of 10 doses) with: (1) saline; (2) drug-free BSA-PLGA NPs (115 mg/kg); (3) drug-free ATRAM-BSA-PLGA NPs (115 mg/kg); (4) free Dox (2 mg/kg); (5) Dox-TPP loaded BSA-PLGA NPs (115 mg/kg, with the loading capacity of Dox-TPP at 1.8 wt%); (6) Dox-TPP loaded ATRAM-BSA-PLGA NPs (115 mg/kg, with the loading capacity of Dox-TPP at 1.8 wt%). The dose of doxorubicin administered was based on previously published work.(Palanikumar *et al.* 2018) Body weight and tumor volume were recorded every 2 days, and survival ($n = 4$ per group) was monitored for a total of 90 days.”

The updated **Figure 8f** of the **Revised Manuscript** is as follows:

“Figure 8. Inhibition of 4T1 tumor growth by Dox-TPP loaded ATRAM-BSA-PLGA NPs. **(a)** *In vivo* pharmacokinetics of Dox-TPP loaded in ATRAM-BSA-PLGA NPs. The Dox concentration in plasma of mice ($n = 6$ per group) treated with free Dox (2 mg/kg) or Dox-TPP loaded BSA-PLGA NPs (115 mg/kg, with the loading capacity of Dox-TPP at 1.8 wt%) was quantified using HPLC(Alhareth *et al.* 2012) at different time points over 48 h. **(b)** Treatment schedule for the tumor reduction studies. Once the tumor volume reached $\sim 25 \text{ mm}^3$, the mice were randomized into the different treatment groups ($n = 8$ per group), which were injected intravenously with: (1) saline; (2) BSA-PLGA NPs (115 mg/kg); (3)

free Dox (2 mg/kg); (4) Dox-TPP loaded BSA-PLGA NPs (115 mg/kg, with the loading capacity of Dox-TPP at 1.8 wt%); (5) Dox-TPP loaded ATRAM-BSA-PLGA NPs (115 mg/kg, with the loading capacity of Dox-TPP at 1.8 wt%). Injections were done every 3 days for a total of 10 doses, with the first day of treatment defined as day 0. (c) Tumor volume growth curves for the different treatment groups over 30 days of treatment. (d) Tumor mass analysis for the different treatment groups. After 30 days of treatment, 4 mice per treatment group were sacrificed and the tumor tissues were isolated and imaged (*left panel*) and subsequently weighed to determine the tumor mass (*right panel*). (e) Hematoxylin and eosin (H&E)-stained images of tumor sections from the different treatment groups following 30 days of treatment. Images on the right are magnified views of the boxed regions in the images on the left. Scale bar = 50 μm . (f) Survival curves for the saline, free Dox and Dox-TPP loaded ATRAM-BSA-PLGA NPs treatment groups ($n = 4$ per group) over 90 days. * $P < 0.05$, ** $P < 0.01$, *** $P < 0.001$ or non-significant (ns, $P > 0.05$) for comparison with controls and amongst the different treatment groups.”

Response to Reviewer #3

The manuscript “pH-Responsive High Stability Polymeric Nanoparticles for Targeted Delivery of Anticancer Therapeutics” from Palanikumar, L., et. al. highlights the development of biocompatible and biodegradable pH-responsive nanoparticles that display: 1) low non-specific release of drugs, 2) high selectivity towards cancer cells, 3) higher circulation stability, 4) selective triggered release of therapeutics, and 5) high applicability for in vivo treatment of tumors, decreasing their volume and improving survival rate; all key characteristics of an ideal delivery system. The authors also looked into the specifics of the uptake mechanism of this particular system by cancer cells, the cytotoxic pathway leading to cancer cell eradication, and the structural characteristics of the nanoparticles that lead to their anticancer properties, providing a complete assessment of the proposed nanoparticle design.

The project has been carried out with great attention to detail, and extensive tests to prove the advantages of these pH-responsive nanoparticles for cancer treatment, including their actual application in an in vivo scenario, and as such I consider it significantly relevant for a publication in *Communications Biology*. However, I do have many concerns and comments that I think should be addressed prior publication, to strengthen the scientific value of the manuscript.

Response: We thank the reviewer for the positive comments regarding the manuscript. We are grateful for the feedback provided by the reviewer and have worked diligently to address all of their concerns. In the following pages, we respond point-by-point to the reviewer’s comments. We believe, and hope the reviewer agrees, that the paper is much improved in content and clarity.

Concern #1. The contents from Scheme 1 should be split into the actual figures of the manuscript at the points where the different topics are discussed. For instance, panel a and c can be made into one panel, and added to Figure 1, panel b to Figure 3, and panel d to Figure 5. I find it quite confusing to see a Scheme with many panels right up front, and then it being referred to repeatedly throughout the manuscript to explain different things.

Response: Following the reviewer’s suggestion, the contents of **Scheme 1** were distributed to the relevant figures:

1. **Panels a** (*Schematic representation of preparation of ATRAM peptide conjugated BSA-PLGA nanoparticles*) and **c** (*Schematic representation of the synthesized Dox-TPP drug, which is designed to target the mitochondria of cancer cells to induce apoptosis*) were combined and moved to **Figure 1** (*Characterization of the BSA-PLGA NPs*).
2. **Panel b** (*Schematic representation of sequestration of PLGA NPs, but not crosslinked BSA-PLGA or ATRAM-BSA-PLGA NPs, by macrophages*) was moved to **Figure 7** (*ATRAM-BSA-PLGA NPs evade uptake by differentiated human monocytic leukemia THP-1 cells*).
3. **Panel d** (*Schematic representation of efficient cellular uptake, by both energy-independent and -dependent mechanisms, of the ATRAM-BSA-PLGA NPs at acidic tumoral pH, followed by intracellular release of the Dox-TPP cargo*) was added to **Figure 5** (*Determination of cellular uptake mechanisms of ATRAM-BSA-PLGA NPs*).

Scheme 1 has now been removed from the **Revised Manuscript**. The updated figures are as follows:

Figure 1. Characterization of the BSA-PLGA nanoparticles (NPs). (a) Schematic representation of preparation of Dox-TPP loaded ATRAM-conjugated BSA-PLGA nanoparticles (b,d) Transmission electron microscopy (TEM, *left panels*) and scanning transmission electron microscopy (STEM, *right panels*) images of PLGA (b) and BSA-PLGA (d) NPs. Scale bar = 50 nm. Size analysis (c) and zeta potential measurements (e) for PLGA and BSA-PLGA NPs in 10 mM phosphate buffer at pH 7.4. (f,g) Colloidal stability analysis for BSA-PLGA NPs in 10 mM phosphate buffer at pH 7.4 (f) and cell culture medium containing 10% fetal bovine serum (FBS) (g). Inset: images of higher concentrations of nanoparticles dispersed in buffer (f) and serum (g) solutions to show the colloidal dispersity after 72 h.”

“Figure 5. Determination of cellular uptake mechanisms of ATRAM-BSA-PLGA NPs. **(a–f)** Confocal laser scanning microscopy images of MCF-7 cells treated with Dox-TPP loaded ATRAM-BSA-PLGA NPs for 1 h, at acidic pH 6.5. Uninhibited uptake in control cells **(a)** was compared to uptake in cells that were pretreated with sodium azide and 2-deoxy-D-glucose to deplete cellular ATP **(b)**. Alternatively, the cells were pretreated with endocytosis inhibitors: chlorpromazine (Chlor; clathrin-dependent endocytosis) **(c)**, methyl- β -cyclodextrin (M β CD; lipid raft-mediated endocytosis) **(d)**, filipin (Flp; caveolae-dependent endocytosis) **(e)** or amiloride (Aml, macropinocytosis inhibitor) **(f)**. Scale bar = 10 μ m. **(g)** **Flow Cytometry** analysis of cellular uptake of Dox-TPP loaded ATRAM-BSA-PLGA NPs in MCF-7 cells under conditions in **(a–f)**. *** $P < 0.05$ compared with controls.** **(h)** **Schematic representation of efficient cellular uptake, by both energy-independent and -dependent mechanisms, of the ATRAM-BSA-PLGA NPs at acidic tumoral pH, followed by intracellular release of the Dox-TPP cargo.**”

“Figure 7. ATRAM-BSA-PLGA NPs evade uptake by differentiated human monocytic leukemia THP-1 cells. **(a,b)** Confocal laser scanning microscopy images of THP-1 cells incubated with Dox-TPP loaded PLGA **(a)** or ATRAM-BSA-PLGA **(b)** NPs for 1–4 h at pH 7.4. Scale bar = 5 μm . **(c,d)** Flow cytometry analysis of cellular uptake of Dox-TPP loaded PLGA, crosslinked BSA-PLGA and ATRAM-BSA-PLGA NPs in THP-1 cells: **(c)** plot of side scatter (SSC) vs fluorescence signal for THP-1 cells that were either untreated (ctrl), or treated with Dox-TPP loaded PLGA, crosslinked BSA-PLGA or ATRAM-BSA-PLGA NPs for 2 h at pH 7.4; **(d)** quantification of cellular uptake of the NPs from the data in **(c)**. **(e)** Schematic representation of sequestration of PLGA NPs, but not crosslinked BSA-PLGA or ATRAM-BSA-PLGA NPs, by macrophages (e.g. monocytes). **(f)** Cell viability of THP-1 cells treated with Dox-TPP loaded PLGA, crosslinked BSA-PLGA or ATRAM-BSA-PLGA NPs for 48 h. Cell viability was assessed using the MTS assay, with the % viability was determined from the ratio of the absorbance of the treated cells to the control cells. **(g)** Release of inflammatory cytokines, tumor necrosis factor-alpha (TNF- α) and interleukin-1 beta (IL-1 β), by THP-1 cells exposed to Dox-TPP loaded PLGA, crosslinked BSA-PLGA or ATRAM-BSA-PLGA NPs for 24 h. Cells treated with lipopolysaccharide (LPS) were used as a positive control for inflammation. TNF- α and IL-1 β levels in the culture medium were assayed using a commercial ELISA kit. * $P < 0.05$, ** $P < 0.01$ or non-significant (ns, $P > 0.05$) compared with controls.”

Concern #2a. How exactly the Dox loading was performed? And where is it expected to be located in the particle: at the surface, on the inside, or both? If Dox can also be exposed on the surface of the nanoparticles, then it can also react during the EDC coupling as it also contains a carboxylic acid. How can this affect the total efficiency of this system (Dox that is not exposed or reacts with EDC vs Dox that reacts with EDC and which efficacy can lower due to the modification)?

Response 2: Dox-TPP was synthesized by conjugating the hydrophobic chemotherapeutic doxorubicin (Dox) to triphenylphosphonium (TPP), which selectively targets mitochondria due to its high lipophilicity and stable cationic charge, by amide bond formation between the amino group in Dox and the carboxyl group in TPP. The resulting mitochondria-targeting Dox-TPP drug is highly hydrophobic.(Han et al. 2014) Dox-TPP loaded polylactic-co-glycolic acid (PLGA) nanoparticles (NPs) were prepared by a simultaneous emulsion solvent diffusion method,(Kocbek et al. 2007) which leads to incorporation of the hydrophobic Dox-TPP within the hydrophobic core of the PLGA NPs. The PLGA core stably encapsulates hydrophobic drugs, but exhibits poor encapsulation efficiency and stability for hydrophilic compounds.(Makadia and Siegel 2011) This means that most of the Dox-TPP is not exposed, and therefore unlikely to be modified, during the EDC coupling of BSA to the surface of the PLGA NPs, and we do not expect a significant adverse effect of the EDC coupling reaction on the efficiency of the system.

Concern #2b. It is clear that there is a high crosslinking density of the BSA layer, as measured, but what is the pore size of this shell? Does it completely avoid leakage because the pores are smaller than the molecules or controls the leakage because of diffusion delay through a crowded network?

Response: To crosslink the BSA shell, we tested various concentrations of the bifunctional compound glutaraldehyde (GA), and the extent of crosslinking was determined by the trinitrobenzenesulfonic acid (TNBS) assay.(Silva et al. 2004) For instance, using 0.1 wt% of GA crosslinked 30% of reactive lysine residues (~35 lysine residues have primary amines out of a total of 59 residues) in BSA. However, this crosslinking density did not significantly reduce the uncontrolled Dox-TPP release from the BSA-PLGA NPs.

Supporting Figure for Reviewers. Dox-TPP release profile from BSA-PLGA NPs with a lower BSA shell crosslinking density.

Increasing the GA concentration to 0.25 wt% crosslinked 72% of reactive lysine residues in BSA (**Figure S5a**), achieving a consistent polydispersity index of 0.02. (Langer et al. 2003) Such a high crosslinking density confers encapsulation stability, (Jiwpanich et al. 2010) yielding BSA-PLGA NPs that did not exhibit any stimulus-free leakage of loaded drug over 24 h (**Figure 3a**).

As for the reviewer's question regarding the pore size of the crosslinked BSA shell of our NPs, we are unable to provide a definitive answer. A common procedure for determining the pore size of mesoporous material is the Barrett-Joyner-Halenda (BJH) method. (Barrett, Joyner, and Halenda 1951) The BJH method calculates pore size distributions from experimental isotherms using the Kelvin model of pore filling (the method uses the modified Kelvin equation to relate the amount of adsorbate removed from the pores of the material, as the relative pressure (P/P_0) is decreased from a high to low value, to the size of the pores). While the BJH method is used to determine pore size of mesoporous inorganic NPs (e.g. silica), the method is not suitable for organic NPs due to destruction or collapse of polymer/protein-based material under the high pressure/temp conditions of the experiment. This was the case with the BSA-PLGA NPs when we attempted to measure the pore size using the BJH method to address the reviewer's concern. This inability to obtain meaningful data using methods such as BJH may explain the almost complete lack of information on the size of pores in crosslinked organic NPs in the literature.

Based on the above, we cannot definitively state whether the inhibition of stimulus-free leakage of loaded drug is "because the pores are smaller than the molecules" or due to "diffusion delay through a crowded network". However, there are several studies on drug release kinetics from BSA NPs with crosslinking densities that are similar to (or even higher than) that of our BSA-PLGA NPs. These BSA NPs showed significant stimulus-free release of drugs of similar size to Dox-TPP but with a more hydrophilic character, which suggests that the pores in the crosslinked structure do not completely impede the escape of the encapsulated cargo. (Arriagada et al. 2019; Zhang et al. 2019; S et al. 2014) Indeed, analysis of the release kinetics from these crosslinked BSA NPs, by fitting several kinetic models (zero-order, first-order, Higuchi, and Korsmeyer-Peppas), indicates that the release is due, in large part, to 'Fickian diffusion of the cargo from the hydrophobic core in the albumin nanosystem'. (Arriagada et al. 2019) This suggests that the inhibition of stimulus-free leakage from our BSA-PLGA NPs is largely a consequence of stiffening of the BSA shell, due to the crosslinking, substantially hindering diffusion of the hydrophobic Dox-TPP cargo from the hydrophobic PLGA core.

Concern #2c. In the TEM pics of the PLGA+BSA layer (Figure 1, page 8), I can indeed see one particle with a layer, but the ones around it do not seem to contain such layer, so it will be nice to see a picture with more than one particle showing the layer. For a symmetrical system like this one, this should be visible in many particles using TEM. Also, for particle's layers/shells made with proteins, the TEM beam might be too destructive to give accurate structural information; CryoTEM characterization is preferred to get more details of the structure. In terms of BSA coating, and based on the pictures provided, how uniform is particle coating, are all particles coated?

Response: As the reviewer correctly points out, the TEM beam can be rather destructive to samples composed of proteins/organic polymers, as is the case with our BSA-PLGA NPs, and often doesn't allow for accurate structural characterization of such samples. This was the case in the TEM images presented in the previous (original) version of the manuscript (old **Figure 1a,c**), where the samples underwent inelastic destruction due to TEM beam damage, leading to the apparent polydispersity of the NPs (see response to reviewer's **Concern #2d**) as well as distortion of their structural details.

We have repeated the TEM analysis, and have included new STEM (scanning TEM) analysis, of the PLGA and BSA-PLGA NPs using a fresh set of grids. The beam energy was regulated so as not to cause damage to the NPs while imaging in TEM and STEM modes. The TEM images were acquired using a 200 kV beam with spot size 5, gun lens 6 and a dose of 1.13–1.16 A/m², to ensure the beam current which did not cause any damage to the sample. The STEM images were acquired in HAADF (high-angle annular dark-field) mode with spot size 9, gun lens 4 and a screen current of less than 0.2 nA. This mode was helpful for clear depiction of the core-shell structure of the BSA-PLGA NPs, as all of the inelastically scattered beam was collected for the image formation.

The TEM and STEM results are presented in the updated **Figure 1b,d** of the **Revised Manuscript**. **Figure 1b** clearly shows monodispersed spherical PLGA NPs in the TEM (*left panel*) and STEM (*right panel*) images. **Figure 1d** shows the core-shell structure of the BSA-PLGA NPs, wherein the shell appears lighter than the grey core in the TEM image (*left panel*), which results from the higher density of the core-forming PLGA compared to the BSA forming the shell. This difference in densities is depicted in the STEM image (*right panel*) as a bright core as surrounded by a fainter shell.

The TEM and STEM images were acquired from regions of the grid with low densities of PLGA and BSA-PLGA NPs to avoid overlapping NPs and allow for discernment of the structural details. However, examination of multiple NPs from different regions of the grid confirmed uniformity in terms of size, shape and morphology.

We agree with the reviewer that CryoTEM is well-suited for NPs that undergo beam damage. However, in this case we decided against utilizing CryoTEM as the technique is time-consuming and labor-intensive, which raised the possibility that we would be unable to complete the necessary experiments within the timeframe allowed for revisions. Instead, we elected to employ the more accessible TEM and STEM techniques, particularly since using the above-mentioned imaging settings allowed us to minimize the apparent beam damage to the samples.

The new imaging settings were added to the updated **Experimental Section** of the **Revised Supporting Information**:

Supporting Information; Experimental Section; Synthesis and Characterization of BSA-PLGA NPs:

“BSA on the surface of the NPs was crosslinked with 500 μ M glutaraldehyde (GA) for 30 min, and excess and unreacted GA was removed by dialyzing three times against phosphate buffer. The synthesized crosslinked BSA-PLGA NPs were characterized using transmission electron microscopy (TEM) and scanning transmission electron microscopy (STEM) (FEI Talos F200X Transmission Electron Microscope). The beam energy was regulated so as to minimize damage to the samples while imaging in TEM and STEM modes. TEM images were acquired using a 200 kV beam with spot size 5, gun lens 6 and a dose of 1.13–1.16 A/m². STEM images were acquired in HAADF (high-angle annular dark-field) mode with spot size 9, gun lens 4 and a screen current of less than 0.2 nA. This mode was helpful for clear depiction of the core-shell structure of the BSA-PLGA NPs, as all of the inelastically scattered beam was collected for the image formation. Further characterization of the NPs was done using dynamic light scattering (DLS), Fourier-transform infrared spectroscopy (FTIR, Agilent Cary 600 Series FTIR Spectrometer) and zeta potential measurements.”

The updated **Figure 1** of the **Revised Manuscript** is as follows:

“Figure 1. Characterization of the BSA-PLGA nanoparticles (NPs). **(a)** Schematic representation of preparation of Dox-TPP loaded ATRAM-conjugated BSA-PLGA nanoparticles. **(b,d)** Transmission electron microscopy (TEM, *left panels*) and scanning transmission electron microscopy (STEM, *right panels*) images of PLGA **(b)** and BSA-PLGA **(d)** NPs. Scale bar = 50 nm. **(c,e)** Size analysis **(c)** and zeta potential measurements **(e)** for PLGA and BSA-PLGA NPs in 10 mM phosphate buffer at pH 7.4. **(f,g)** Colloidal stability analysis for BSA-PLGA NPs in 10 mM phosphate buffer at pH 7.4 **(f)** and cell culture medium containing 10% fetal bovine serum (FBS) **(g)**. Inset: images of higher concentrations of nanoparticles dispersed in buffer **(f)** and serum **(g)** solutions to show the colloidal dispersity after 72 h.”

Concern #2d. Furthermore, in relation to size there does not seem to be an agreement between the TEM pictures and the DLS data, because I can see polydispersity in TEM, while DLS gives a sharp peak. Why is this? If the real sample actually has smaller and larger sizes, like the TEM pictures, then this could also affect the particle uptake mechanism and even display two different internalization pathways, just like the results observed by the authors, which will be dependent on the size.

Response: All new NP formulations (PLGA, BSA-PLGA without shell crosslinking and crosslinked BSA-PLGA) were characterized using DLS, and consistently showed monodispersity. The polydispersity observed in the TEM images presented in the previous (original) version of the manuscript (old **Figure 1a,c**), was a result of the NPs undergoing inelastic destruction due to TEM beam damage. TEM and STEM images acquired with a fresh set of grids using new imaging conditions (described in the response to reviewer’s **Concern #2c**) clearly show monodispersity of the NPs in terms of size, shape

and morphology (updated **Figure 1b,d** of the **Revised Manuscript**). Thus, there is no disagreement between the DLS and TEM data.

Cellular internalization of ATRAM-BSA-PLGA NPs occurs via two different pathways, namely direct translocation and clathrin-mediated endocytosis (**Figure 5** and **S13**). If the cellular uptake of the NPs is simply a function of their size, as the reviewer suggests, we would not expect that uptake to be pH-dependent (**Figures 4** and **5**). The pH-dependence of the internalization of the NPs confirms that it is facilitated by the pH-responsive ATRAM peptide. The cellular uptake of pH-responsive peptides such as ATRAM is proposed to involve pH-dependent insertion into the cell membrane – where the peptide undergoes a conformation change from a membrane surface adsorbed unstructured peptide at physiological pH to a transmembrane α -helix at acidic pH – followed by translocation of the peptide across the lipid bilayer.(Nguyen et al. 2015; 2019) However, there is evidence that, following pH-dependent insertion, a fraction of these peptides is also taken up by endocytosis, particularly when the peptide is coupled to a sizeable cargo, such as a liposome or a nanoparticle.(Nguyen et al. 2019) Therefore, the utilization of two different uptake pathways by the hybrid NPs is due to the ATRAM peptide, which facilitates the cellular internalization of the coupled NPs at low pH.

Concern #2e. Could you comment further as to why the colloidal stability of particles with BSA on the surface is higher than PEG coated systems? I would have thought that particles covered with proteins will be less stable due to both steric and electrostatic effects

Response: Interactions with blood components, such as serum proteins, are reported to cause substantial leakage of loaded drugs from self-assembled polymeric structures.(Zhao et al. 2016; Palanikumar et al. 2018) Moreover, the formation of a serum protein corona on receptor-targeting NPs during *in vivo* circulation adversely affects target recognition and results in nonspecific distribution.(Zhao et al. 2016; Dai, Walkey, and Chan 2014) To overcome these issues, we have designed hybrid NPs that consist of a drug-loaded biocompatible and biodegradable polylactic-co-glycolic acid (PLGA)(Makadia and Siegel 2011) core ‘wrapped’ with a shell composed of bovine serum albumin (BSA).

Analysis using the BCA protein assay revealed that the concentration of BSA conjugated to PLGA NPs was ~7.3 wt% of the NPs, which is close to the reported 8 wt% of PEGylation over the surface of NPs required to provide a backfilling effect for long term circulation, as well as prevent the formation of a serum protein corona that can cause unexpected changes in cellular interactions and uptake, biodistribution, and immunogenicity. Quantitative proteomic analysis confirmed that the BSA shell successfully prevents the formation of a serum protein corona the surface of the NPs (**Figure 2**).

Encapsulation stability and drug release analyses of the BSA-PLGA NPs in buffer, cell culture medium (with FBS) and in cells (**Figures 3** and **S7**), revealed that the BSA shell significantly reduces drug release from the PLGA core. This is likely due to thermodynamically favorable interactions between specific domains of BSA with the surface of the PLGA NPs leading to colloidal stabilization and acquisition of stealth properties (i.e. prevention of serum protein adsorption).(Schöttler et al. 2016; Oh et al. 2018; Treuel et al. 2014; Mortimer et al. 2014) However, presence of the BSA shell alone is not sufficient to completely abolish the drug release.

This prompted us to crosslink the BSA shell using glutaraldehyde (GA).(Nakamura et al. 1998) The crosslinked BSA-PLGA NPs are characterized by very high encapsulation stability and do not exhibit any stimulus-free leakage of loaded drugs (**Figures 3** and **S7,8**). Moreover, these NPs showed good colloidal stability in buffer, medium, and under conditions of high serum content (**Figure 1f,g** and **S6**).

This strongly suggests that crosslinking of the shell reinforces the aforementioned thermodynamically favorable interactions between the BSA shell and PLGA NPs, yielding a high degree of colloidal stability. However, we have refrained from directly comparing the stability of the crosslinked BSA-PLGA NPs with that of PEG coated systems, particularly since the sheer diversity of those systems makes such a comparison difficult.

Concern #2f. The cell environment has high ionic strength, which is known to significantly affect both the zeta potential and the colloidal stability of particles. Even though the stability of the PLGA-BSA particles was measured in presence of proteins and 10 mM PBS, it will be also good to show how increasing ionic strength to values near those found in cells will affect the colloidal properties of this particular system.

Response: Following the reviewer's suggestion, we have carried out additional control experiments in which we assessed the effect of higher ionic strengths (i.e. at the levels found in cells and higher) on the stability of the crosslinked BSA-PLGA NPs (**Figure S6a**). Incubating the BSA-PLGA NPs in solutions of 1×PBS (which contains 137 mM NaCl, 2.7 mM KCl, 8 mM Na₂HPO₄, and 2 mM KH₂PO₄) with added increasing concentrations of NaCl (5–50 mM), did not significantly alter the hydrodynamic diameter of the NPs (**Figure S6b**). This indicates that the BSA-PLGA NPs will remain stable (i.e. they will not prematurely release their chemotherapeutic cargo) in cellular milieu until exposure to an appropriate stimulus (e.g. low pH).

The results of the additional control experiments are presented in the new **Figure S6a** of the **Revised Supporting Information**:

“Figure S6. Effects of high serum concentration and high ionic strength on colloidal stability of crosslinked BSA-PLGA NPs. Size analysis of BSA-PLGA NPs incubated for 72 h in: (a) cell culture medium containing 50% fetal bovine serum (FBS), or (b) 1×PBS (137 mM NaCl, 2.7 mM KCl, 8 mM Na₂HPO₄, and 2 mM KH₂PO₄) containing increasing concentrations of NaCl.”

The relevant paragraph of the **Results and Discussion** of the **Revised Manuscript** has been modified accordingly and now reads as follows:

Results and Discussion; Preparation and Characterization of BSA-PLGA Nanoparticles (NPs):

“The BSA-PLGA NPs showed good colloidal stability in 10 mM phosphate buffer solution (pH 7.4) and cell culture medium containing 10% fetal bovine serum (FBS) over 72 h (**Figure 1f,g**). Moreover, the BSA-PLGA NPs maintained a similar surface charge in cell culture medium containing 10% FBS (**Figure S5c**). Next, we monitored the size of the BSA-PLGA NPs under conditions of high serum content and high ionic strength, both of which are reported to affect the colloidal stability of nanocarriers. (L. Moore et al. 2015) **Increasing the serum content to 50% resulted in a modest increase of ~30 nm in the hydrodynamic diameter of the BSA-PLGA NPs**

(Figure S6a) relative to that in 10% FBS, suggesting that the NPs are able to maintain a reasonable size in circulation that will allow them to localize to tumors and then readily internalize into cancer cells to deliver their chemotherapeutic cargo. Likewise, increasing the ionic strength to the levels found in cells and higher (1× PBS [137 mM NaCl, 2.7 mM KCl, 8 mM Na₂HPO₄, and 2 mM KH₂PO₄] with added 5–50 mM NaCl)(Gao *et al.* 2012) did not significantly alter the size of the BSA-PLGA NPs (Figure S6b), indicating that the NPs will remain stable and retain their chemotherapeutic cargo in cellular milieu until exposure to an appropriate stimulus (e.g. low pH). Taken together, these results show that the crosslinked BSA shell confers a high degree of stability on the NPs. This is likely due to thermodynamically favorable interactions between specific domains of BSA with the surface of the PLGA NPs, which are reinforced by crosslinking of the shell, leading to colloidal stabilization and acquisition of stealth properties (i.e. prevention of serum protein adsorption).(Schöttler *et al.* 2016; Oh *et al.* 2018; Treuel *et al.* 2014; Mortimer *et al.* 2014)”

Concern #3a. First sentence of “Encapsulation Stability...”, change “to assess the stability...” to “to assess the encapsulation stability...”.

Response: We thank the reviewer for their careful reading of the manuscript! As per the reviewer’s suggestion, we have now modified the relevant sentence to:

Results and Discussion; Encapsulation Stability of BSA-PLGA NPs:

“Förster resonance energy transfer (FRET) experiments were used to assess the encapsulation stability of PLGA, non-crosslinked BSA-PLGA and crosslinked BSA-PLGA NPs (Figure S7).”

Concern #3b. It is mentioned that “Cellular FRET results (Figure S7) were consistent with the solution experiments.”, however different conditions are tested (BSA-PLGA with FRET pair and PLGA with FRET pair versus BSA-PLGA FRET pair and BSA-PLGA no complete FRET pair), not the same exact particles, which does not represent consistency in results. Yes, this new data shows that the FRET pair or DiI is kept inside the particles when they are exposed to cells, and that is positive, but it does not show improvement of dye retention in BSA coated vs non-coated like the previous data. A better control would have been to have the DiI loaded particle exposed to cells with DiO in the surrounding solution, if there was leakage, because you need close proximity of the dyes to have FRET, you should see FRET signal, if there is no leakage, you should see little to no signal. Another control could have been using PLGA-FRET pair just like in the solution experiments.

Response: As per the reviewer’s recommendation, we have performed an additional control experiment to confirm that the cellular FRET results are consistent with the solution experiments. Incubation of MCF-7 cells with the DiO/DiI co-loaded PLGA NPs resulted in intracellular accumulation of the fluorophores over 4 h, but no detectable FRET signal, indicating degradation of the NPs and separation of the FRET pair (**Figure S8a**). In contrast, treatment of with DiO/DiI co-loaded crosslinked BSA-PLGA NPs for 4 h lead to a strong intracellular FRET signal, indicating close proximity of DiO and DiI (**Figure S8b**). Finally, MCF-7 cells treated with control DiI-loaded crosslinked BSA-PLGA NPs did not show any FRET signal (**Figure S8c**). These results demonstrate the high encapsulation stability of the crosslinked BSA-PLGA NPs, which are able to retain loaded dyes even in the presence of high concentrations of serum proteins or in a cellular environment.

In addressing the reviewer’s concern, the relevant sections of the **Revised Manuscript** and **Revised Supporting Information** have been modified as follows:

Results and Discussion; Encapsulation Stability of BSA-PLGA NPs:

“Cellular FRET results (Figure S8) were consistent with the solution experiments. Incubation of MCF-7 cells with the DiO/DiI co-loaded PLGA NPs resulted in intracellular accumulation of the fluorophores over 4 h (Figure S8a). However, no FRET signal was detectable, which indicates degradation of the NPs and release of the FRET pair. In contrast, treatment of with DiO/DiI co-loaded crosslinked BSA-PLGA NPs for 4 h led to a strong FRET signal from subcellular compartments, indicating close proximity of DiO and DiI within intact NPs (Figure S8b). As a control, MCF-7 cells treated with crosslinked BSA-PLGA NPs loaded with only the acceptor dye, DiI, did not show any FRET signal (Figure S8c). These results demonstrate the high encapsulation stability of the crosslinked BSA-PLGA NPs, which are able to retain loaded dyes even in the presence of high concentrations of serum proteins or in a cellular environment.”

Supporting Information; Experimental Section; Cellular Förster Resonance Energy Transfer (FRET):

“MCF-7 cells were seeded at a density of 2×10^5 cells/well in 500 μ L complete DMEM in 4-chambered 35 mm glass bottom Cellview cell culture dishes. After culturing for 24 h, the medium was replaced fresh DMEM containing 0.1 mg/mL NPs (PLGA or crosslinked BSA-PLGA) co-loaded with 0.61% DiO/DiI or loaded with 0.70% DiI, and incubated for 4 h. Immediately prior to imaging, the medium was replaced with fresh DMEM to remove any extracellular NPs. FRET images were acquired on an Olympus Fluoview FV-1000 confocal laser scanning microscope, using a 63 \times Plan-Apo/1.3 NA oil immersion objective with DIC capability. Images were recorded in the DiO channel ($\lambda_{\text{ex}} = 488$ nm, $\lambda_{\text{em}} = 543$ nm; DiI channel ($\lambda_{\text{ex}} = 543$ nm, $\lambda_{\text{em}} = 633$ nm); FRET channel ($\lambda_{\text{ex}} = 488$ nm, $\lambda_{\text{em}} = 633$ nm). The FRET ratio was calculated as: $\text{FRET ratio} = I_{\text{FRET}} / (I_{\text{FRET}} + I_{\text{DiO}})$. The exposure time was 200 ms for the DiO and FRET channels and 100 ms for the DiI channel.”

The results of the new control experiment are shown in the new **Figure S8** in the **Revised Supporting Information**:

Figure S8. Encapsulation stability of the BSA-PLGA NPs in cells. Confocal laser scanning microscopy images of MCF-7 cells incubated with DiO/DiI co-loaded PLGA (a) or crosslinked BSA-PLGA (b) NPs,

or DiI loaded crosslinked BSA-PLGA NPs (c), for 4 h at pH 7.4. Confocal microscope fluorescence parameters: DiO, $\lambda_{ex/em} = 488/543$ nm; DiI, $\lambda_{ex/em} = 543/633$ nm; FRET, $\lambda_{ex/em} = 488/633$ nm; FRET ratio = $I_{FRET}/(I_{FRET} + I_{DiO})$. Scale bar = 10 μ m.

Concern #3c. The encapsulation stability experiments are done comparing PLGA-BSA and PLGA particles, but why not showing too results for PLGA-BSA non-crosslinked vs crosslinked? That will be more conclusive as to show the need for the crosslinking step.

Response: As per the reviewer's recommendation, we performed additional encapsulation stability experiments with non-crosslinked BSA-PLGA NPs co-loaded with DiO/DiI in buffer and cell culture medium containing 10 and 20% serum (**Figure S7**). The results of these new control experiments show that, similar to PLGA NPs, non-crosslinked BSA-PLGA NPs are unstable in serum containing solutions, whereas crosslinked BSA-PLGA NPs are stable under the same conditions. Thus, these new control experiments provide more conclusive evidence for the importance of the BSA shell crosslinking for stability of the NPs.

The results of the new control experiment are shown in the updated **Figure S7** of the **Revised Supporting Information**:

“Figure S7. Encapsulation stability of the BSA-PLGA NPs in solution. (a) Schematic representation of the FRET-based encapsulation stability analysis for PLGA, non-crosslinked BSA-PLGA and crosslinked

BSA-PLGA NPs co-loaded with DiO (donor, $\lambda_{em} = 505$ nm) and DiI (acceptor, $\lambda_{em} = 565$ nm) fluorophores. **(b–d)** FRET efficiency analysis for PLGA (*left*), non-crosslinked BSA-PLGA (*middle*) and crosslinked BSA-PLGA (*right*) NPs in 10 mM phosphate buffer solution at pH 7.4 (**b**), or cell culture medium containing 10% (**c**) or 20% (**d**) FBS. The excitation wavelength was set at 450 nm, and the fluorescence spectra were recorded at different time points over 48 h. **(e)** Summary of time-dependence of FRET efficiency ($=I_{DiI}/(I_{DiI} + I_{DiO})$) for PLGA, non-crosslinked BSA-PLGA and crosslinked BSA-PLGA NPs under the conditions described in **(c,d)**.”

The relevant sections of the **Revised Manuscript** and **Revised Supporting Information** have been modified and now read as follows:

Results and Discussion; Encapsulation Stability of BSA-PLGA NPs:

“Förster resonance energy transfer (FRET) experiments were used to assess the encapsulation stability of PLGA, non-crosslinked BSA-PLGA and crosslinked BSA-PLGA NPs (Figure S7). The NPs were co-loaded with the FRET donor (3,3'-dioctadecyloxycarbocyanine perchlorate, DiO) and acceptor (1,1'-dioctadecyl-3,3,3',3'-tetramethylindocarbocyanine perchlorate, DiI) pair (loading capacity 1 wt%) (Figure S7a). In the case of PLGA and non-crosslinked BSA-PLGA NPs, the acceptor dye, DiI, fluorescence intensity at 565 nm remained constant in 10 mM phosphate buffer (Figure S7b), but decreased significantly in cell culture medium containing FBS (Figure S7c,d). On the other hand, for crosslinked BSA-PLGA NPs we observed no time-dependent decrease in DiI fluorescence intensity in either 10 mM phosphate buffer (Figure S7b) or cell culture medium containing 10% or 20% FBS (Figure S7c,d). The calculated time-dependent FRET efficiencies confirmed release of the FRET pair from PLGA and non-crosslinked BSA-PLGA NPs when incubated in serum-containing solutions due to destabilization of the carrier,(Palanikumar et al. 2018) whereas crosslinked BSA-PLGA NPs are stable under the same conditions (Figure S7e).”

Supporting Information; Experimental Section; Synthesis and Characterization of BSA-PLGA NPs:

“Encapsulation stability and drug release kinetics of the NPs were measured on PerkinElmer LS-55 Fluorescence Spectrometer. Encapsulation stability was assessed using Förster resonance energy transfer (FRET).(Rajdev and Ghosh 2019) The NPs (PLGA, non-crosslinked BSA-PLGA and crosslinked BSA-PLGA) were co-loaded with the FRET donor (3,3'-dioctadecyloxycarbocyanine perchlorate, DiO) and acceptor (1,1'-dioctadecyl-3,3,3',3'-tetramethylindocarbocyanine perchlorate, DiI) pair. Release of the dyes from the NPs, which results in decreased FRET, was monitored under different conditions (10 mM phosphate buffer, or cell culture medium containing 10% or 20% FBS) by measuring the acceptor DiI fluorescence ($\lambda_{em} = 565$ nm) upon excitation of the donor DiO ($\lambda_{ex} = 450$ nm).”

Concern #4a. Figure 3: why there is no release profile for extracellular cancer cell pH 6.0-6.5? This condition is shown for the ATRAM peptide but not for the release. It will be more consistent to show release in normal pH, pH outside cancer cell and pH inside cancer cell.

Response: As suggested by the reviewer, we have performed an additional control experiment in which the cargo release from crosslinked BSA-PLGA NPs was monitored at pH 6.5 (i.e. the extracellular tumoral pH). At this pH, negligible release of Dox-TPP was observed, indicating that the crosslinked BSA-PLGA NPs will effectively retain their chemotherapeutic cargo within the mildly acidic microenvironment of malignant solid tumors (pH 6.5–6.9),(Palanikumar et al. 2017; Persi et al. 2018)

until they are taken up by cancer cells and exposed to the intracellular stimuli (low pH and/or high levels of GSH) that would degrade the NPs and release the cargo.

The results of the new control experiment are shown in the updated **Figure 3b** of the **Revised Manuscript**:

“Figure 3. Drug release profiles of the BSA-PLGA NPs in the absence and presence of a stimulus. (a) Stimulus-free release from Dox-TPP loaded PLGA or BSA-PLGA NPs (without and with shell crosslinking) in 10 mM phosphate buffer (pH 7.4). (b,c) pH-triggered release from Dox-TPP loaded (b) and rhodamine B-loaded (c) BSA-PLGA NPs with shell crosslinking. (d) Glutathione (GSH) mediated Dox-TPP release from crosslinked BSA-PLGA NPs in 10 mM phosphate buffer (pH 7.4).”

In addressing the reviewer’s concern, the relevant sections of the **Revised Manuscript** and **Revised Supporting Information** have been modified and now read as follows:

Results and Discussion; Drug Release Profile of BSA-PLGA NPs:

“Since endocytosis is a common cellular uptake route for nanocarriers,(Palanikumar, Kim, et al. 2015) during their endocytic entry into cells NPs will be exposed to increasingly acidic environments as they are trafficked from weakly acidic early/maturing endosomes (pH 6.0–5.5) to more acidic late endosomes/lysosomes (pH 5.0–4.5).(Fennelly and Amaravadi 2017; Piao and Amaravadi 2016) Therefore, we probed the effects of an acidic environment on the release of Dox-TPP from crosslinked BSA-PLGA NPs (Figure 3b). At pH 6.5, negligible release of Dox-TPP was observed, indicating that the crosslinked BSA-PLGA NPs will effectively retain their chemotherapeutic cargo within the mildly acidic microenvironment of malignant solid tumors (pH 6.5–6.9).(Palanikumar et al. 2017; Persi et al. 2018)”

Supporting Information; Experimental Section; Synthesis and Characterization of BSA-PLGA NPs:

“Cargo (Dox-TPP or rhodamine B) release from the NPs, in the absence or presence of a stimulus – e.g. pH or glutathione (GSH) – was monitored using the dialysis method.(Huang et al. 2018) Briefly, 1 mL of 200 μ g/mL Dox-TPP loaded NPs (PLGA, non-crosslinked BSA-PLGA or crosslinked BSA-PLGA) was placed in a dialysis bag (molecular-weight cutoff: 3 kDa), with or

without GSH (0.5 or 10 mM), and fully submerged into 25 mL of release medium – 10 mM phosphate buffer (pH 7.4 or 6.5) or 10 mM acetate buffer (pH 5.8, 5.0 or 4.0) – followed by stirring at 100 rpm. At the designated time points, 1 mL of the release medium was removed for analysis, and replenished with the same volume of fresh buffer. Dox-TPP fluorescence in the release sample was measured ($\lambda_{\text{ex/em}} = 480/580$ nm) and amount of Dox-TPP released was determined using a standard calibration curve.”

Concern #4b. More explanation as to why the protonation of Dox-TPP leads to release at lower pH? Until this point in the manuscript, I only got the impression of higher release because of PLGA degradation, and not because of electrostatic repulsion of the drug with the carrier. This could be explained in the section related to Dox-TPP synthesis. I also guess that the conditions for encapsulation of Dox have to take this into account, and this is not mentioned. Also, the control (Rhodamine B) used is not non-pH-sensitive, it is pH sensitive but maybe in the range studied it stays the same. Rhodamine B is a cation at pH lower than 4, and a zwitterion at pH higher than 4, which means that for these experiments it was zwitterionic. However, there was not much release with this molecule, which is essentially charge-neutral, but good release with Paclitaxel, which is neutral and hydrophobic, after explaining the importance of the protonation step of Dox for release (besides PLGS degradation)? This is not clear. An explanation for no Rhodamine release vs Paclitaxel release is needed. Maybe even show with a negative drug what happens. Also, is loading efficiency of Rhodamine B and Paclitaxel comparable to Dox-TPP?

Response: The reviewer is correct in stating that release of the Dox-TPP cargo from the BSA-PLGA NPs under acidic conditions is primarily driven by degradation of the PLGA core, which results from hydrolysis of the ester bonds in the polymer chains.(Rezvantlab et al. 2018) However, as the pH decreases, Dox-TPP is protonated and becomes more hydrophilic, so it is reasonable to assume that this will lead to dissociation of the drug from the hydrophobic PLGA core. Thus, it is likely that dissociation of the protonated Dox-TPP from the (now degraded) hydrophobic PLGA core contributes to the drug’s acid-triggered release from the NPs.

To test this hypothesis, we carried out an additional control experiment. As the reviewer points out, rhodamine B is zwitterionic under the previously tested conditions (i.e. pH 7.4 and 5.8), where negligible release of the dye was observed (**old Figure 3c**). Therefore, we monitored release of rhodamine B from BSA-PLGA NPs at pH 4.0, where the dye becomes cationic due to protonation of its carboxylic acid group ($pK_a \sim 4.2$),(Yu et al. 2013). In this case, we observe significant release of rhodamine B from BSA-PLGA NPs ($44 \pm 2\%$ release at 24 h) (**new Figure 3c**). This strongly supports the notion that protonation of the cargo contributes to its acid-triggered release.

Regarding paclitaxel (PTX), we will address the release mechanism from the NPs (vs that of Dox-TPP and rhodamine B) in our response to the reviewer’s **Concern #7**.

In addressing the reviewer’s concern, the relevant paragraph of the **Revised Manuscript** has been modified to more clearly explain the proposed mechanism of acid-triggered Dox-TPP release from BSA-PLGA NPs, and now read as follows:

Results and Discussion; Drug Release Profile of BSA-PLGA NPs:

“Lowering the pH to within the range reported for early endosomes (pH 5.8) triggered the release of a significant amount of the Dox-TPP cargo ($45 \pm 5\%$ release at 24 h). The release under acidic

conditions occurs due to degradation of the PLGA core (as a result of hydrolysis of the ester bonds in the polymer chains). (Rezvantlab *et al.* 2018) Additionally, as the pH decreases, dissociation of the increasingly hydrophilic Dox-TPP from the hydrophobic PLGA core likely contributes to the acid-triggered release of the drug. (Liu *et al.* 2018; Palanikumar *et al.* 2017) Lowering the pH even further to that of late endosomes/lysosomes (pH 5.0), led to much higher release of Dox-TPP ($79 \pm 4\%$ at 24 h) (Figure 3b). As a control, we monitored release of the hydrophobic dye rhodamine B from crosslinked BSA-PLGA NPs (Figure 3c). Negligible release of rhodamine B was observed at pH 7.4 or 5.8. However, at pH 4.0, where rhodamine B becomes cationic due to protonation of its carboxylic acid group ($pK_a \sim 4.2$), (Yu *et al.* 2013) a significant amount of the dye is released from the NPs ($44 \pm 2\%$ release at 24 h). This supports the notion that protonation of the cargo contributes to its acid-triggered release. Thus, the acidic microenvironment of endocytic compartments is expected to facilitate the efficient release of the Dox-TPP cargo following internalization of the crosslinked BSA-PLGA NPs into cancer cells.”

The results of the new control experiment are shown in the updated **Figure 3c** of the **Revised Manuscript**:

“**Figure 3.** Drug release profiles of the BSA-PLGA NPs in the absence and presence of a stimulus. (a) Stimulus-free release from Dox-TPP loaded PLGA or BSA-PLGA NPs (without and with shell crosslinking) in 10 mM phosphate buffer (pH 7.4). (b,c) pH-triggered release from Dox-TPP loaded (b) and rhodamine B-loaded (c) BSA-PLGA NPs with shell crosslinking. (d) Glutathione (GSH) mediated Dox-TPP release from crosslinked BSA-PLGA NPs in 10 mM phosphate buffer (pH 7.4).”

Concern #5a. What is the pH of the solutions that contain GSH?

Response: The solution containing glutathione (GSH) was 10 mM phosphate buffer (pH 7.4). This is now explicitly stated in the legend for **Figure 3d**:

Figure Legends:

“**Figure 3. (d)** Glutathione (GSH) mediated Dox-TPP release from crosslinked BSA-PLGA NPs in 10 mM phosphate buffer (pH 7.4).”

Concern #5b. How GSH-BSA complexes can disrupt a covalently crosslinked shell, that is covalently attached to the particle? If the BSA was crosslinked via S-S bridges only, then I would understand this statement; but that is not the case, so how exactly that disruption occurs? Can this be proved by looking at NMR, IR of BSA crosslinked or not crosslinked in presence of GSH?.

Response: We agree with the reviewer that the mechanism of GSH-mediated disruption of the crosslinked BSA shell is not immediately obvious. Exposure of the BSA-PLGA NPs to GSH (0.5–10 mM) leads to significant release of the Dox-TPP cargo (**Figure 3d**). As GSH does not damage PLGA (please see response to **Concern #5c** below), this logically leads us to conclude that the target of GSH is the BSA shell, particularly since GSH is known to have a high affinity for BSA. Thus, one possible explanation for the observed GSH-mediated cargo release from the BSA-PLGA NPs is that the high binding affinity of GSH to BSA leads to formation of stable GSH-BSA complexes (Jahanban-Esfahlan and Panahi-Azar 2016) on the NP surface, which somehow destabilize the BSA shell. Another possibility is that the crosslinked BSA shell is further stabilized by disulfide bond formation between the exposed Cys34 of adjacent BSA molecules (Rombouts et al. 2015) brought into close proximity by crosslinking. GSH would reduce the BSA-BSA disulfide bonds, thereby weakening the BSA shell. Regardless of the exact mechanism, the results strongly suggest that the elevated levels of GSH within cancer cells will promote disruption of the BSA shell of the NPs and release of the Dox-TPP cargo intracellularly.

While we appreciate the reviewer's suggestion, it is not currently obvious to us how NMR or IR may definitively prove the proposed mechanism. However, we hope to explore the use of these techniques, along with others at our disposal, in follow-up studies to provide direct evidence that supports (or disproves) the proposed mechanism of GSH-mediated disruption of the crosslinked BSA shell.

In addressing the reviewer's concern, the relevant paragraph of the **Revised Manuscript** has been modified, and now read as follows:

Results and Discussion; Drug Release Profile of BSA-PLGA NPs:

“Cancer cells exhibit persistently high levels of reactive oxygen species (ROS) due to the presence of oncogenic mutations that promote aberrant metabolism and protein translation. (Cairns, Harris, and Mak 2011) Since high levels of ROS can damage macromolecules, inactivate enzymes, and trigger senescence and apoptosis, cancer cells counteract the detrimental effects of ROS by producing elevated levels of antioxidant molecules, such as the tripeptide glutathione (GSH; γ -glutamyl-cysteinyl-glycine). (Cairns, Harris, and Mak 2011) Cellular GSH is primarily localized to the cytosol, mitochondria, endoplasmic reticulum and nucleus, and intracellular concentrations of the antioxidant are orders of magnitude higher than those in extracellular fluids. (Montero et al. 2013) Moreover, studies have reported several fold higher cytosolic levels of GSH in tumors cells compared to healthy cells. (X. Wang et al. 2013; Palanikumar et al. 2017) Given these differences, we tested whether GSH represents a convenient stimulus for destabilization of the NPs and release of their cargo within cancer cells. Exposure of the crosslinked BSA-PLGA NPs to 0.5 mM GSH at pH 7.4, which is within the range reported for healthy mammalian cells, (Deponte 2013) triggered the release of a modest amount of the Dox-TPP cargo ($18 \pm 3\%$ release at 24 h) (Figure 3d). However, addition of 10 mM GSH at pH 7.4, which is within the range reported for malignant cells, (X. Wang et al. 2013; Bansal and Simon 2018) led to substantial release of Dox-TPP from the NPs ($47 \pm 3\%$ release at 24 h). As GSH does not adversely affect PLGA, (Paka and Ramassamy 2017) this suggests that the target of GSH is the BSA shell, particularly since GSH is known to have a high affinity for BSA. (Jahanban-Esfahlan and Panahi-Azar 2016) A possible explanation for the

observed GSH-mediated cargo release from the BSA-PLGA NPs is that the high binding affinity of GSH to BSA leads to formation of stable GSH-BSA complexes (Jahanban-Esfahlan and Panahi-Azar 2016) on the NP surface, which destabilize the BSA shell. Another possibility is that the crosslinked BSA shell is further stabilized by disulfide bond formation between the exposed Cys34 of adjacent BSA molecules (Rombouts et al. 2015) brought into close proximity by crosslinking. GSH would reduce the BSA-BSA disulfide bonds, thereby weakening the BSA shell. Regardless of the exact mechanism, the results strongly suggest that the elevated levels of GSH within cancer cells will promote disruption of the BSA shell of the NPs and release of the Dox-TPP cargo intracellularly.”

Concern #5c. In the GSH experiments, how is it known that the PLGA is not damaged during GSH exposure, and is only BSA shell damage leading to release ?

Response: A number of PLGA-based NP systems have recently been reported in which PLGA directly interacts with GSH. These include NPs with GSH-functionalized PLGA (*Mol. Pharmaceutics* **2017**, 14, 93-106), NPs in which a GSH layer is ‘wrapped’ around a PLGA core (*Int. J. Nanomedicine* **2019**, 14, 1533–1549), and NPs in which GSH is incorporated within the PLGA core (*Sci. Rep.* **2019**, 9, 11098 ; *J. Biomed.* **2018**, 3, 50-59). In all of these cases, the presence of GSH did not affect the drug loading efficiency or drug release profile, size, polydispersity, zeta potential, or stability of the PLGA-based formulations. This leads us to conclude that GSH does not damage the PLGA core, but rather destabilizes the BSA shell, of the BSA-PLGA NPs.

Concern #6. Page 14: Figure 4: Why the cellular uptake was not shown with BSA-PLGA as control?

Response: To address the reviewer’s concern, we have performed additional control experiments. Cellular uptake in MCF-7 cells of Dox-TPP loaded BSA-PLGA NPs was assessed using confocal fluorescence microscopy (**Figure S12**) and flow cytometry (**Figure 4e,f**). In the absence of the ATRAM peptide, poor uptake of BSA-PLGA NPs was observed at both pHs (7.4 and 6.5) and incubation times (1 and 4 h). This confirms that the pH-dependent membrane insertion of ATRAM facilitates uptake of the coupled NPs into cancer cells at low pH.

The results of the new control experiments suggested by the reviewer are presented in the new **Figure S12** and updated **Figure 4e,f**.

“Figure S12. Confocal laser scanning microscopy images of MCF-7 cells incubated with Dox-TPP loaded BSA-PLGA NPs for 1 h at pH 7.4 (top panels) and pH 6.5 (lower panels). Scale bar = 10 μm.”

“Figure 4. Cellular uptake of ATRAM-BSA-PLGA NPs is strongly pH-dependent. (a) Size analysis for ATRAM-BSA-PLGA NPs in aqueous solution. (b,c) Zeta potential measurements of BSA-PLGA (b) and ATRAM-BSA-PLGA (c) NPs in aqueous solution. (d) Confocal laser scanning microscopy images of MCF-7 cells incubated for 1 or 4 h with Dox-TPP loaded ATRAM-BSA-PLGA NPs at physiological (*top panels*) or acidic (*lower panels*) pH. Quantification of colocalization of the Dox-TPP cargo with mitochondria using Pearson’s correlation coefficient (*r*; *right panels*). Scale bar = 10 μ m. (e,f) Flow cytometry analysis of cellular uptake of Dox-TPP loaded BSA-PLGA and ATRAM-BSA-PLGA NPs in MCF-7 cells: (e) plot of side scatter (SSC) vs fluorescence signal for MCF-7 cells that were either untreated (ctrl), or treated with Dox-TPP loaded BSA-PLGA (*left panel*) or ATRAM-BSA-PLGA (*right panel*) NPs for 1 or 4 h at pH 7.4 or 6.5; (f) quantification of cellular uptake of BSA-PLGA and ATRAM-BSA-PLGA NPs at different incubation times and pHs from the data in (e). * $P < 0.001$ compared with NPs at pH 7.4.”**

In addressing the reviewer’s concern, we have modified the relevant sections of the **Revised Manuscript** and **Revised Supporting Information** as follows:

Results and Discussion; Cellular Uptake of ATRAM-BSA-PLGA NPs:

“The uptake of ATRAM-BSA-PLGA NPs in cancer cells was assessed using confocal fluorescence microscopy. Human breast cancer MCF-7 cells were incubated with Dox-TPP loaded ATRAM-BSA-PLGA NPs for 1 and 4 h at 37 $^{\circ}$ C (Figure 4d). Substantially higher cellular

internalization of the Dox-TPP cargo was observed at pH 6.5 compared to pH 7.4 at both 1 and 4 h incubation times. Moreover, the greater amount of intracellular Dox-TPP at acidic pH showed strong colocalization with mitochondria. Similarly, in human cervical cancer HeLa cells we observed a marked enhancement in cellular uptake, and mitochondrial localization, of the Dox-TPP cargo of ATRAM-BSA-PLGA NPs at acidic pH compared to physiological pH (Figure S11). Quantification of cellular uptake using flow cytometry confirmed the confocal microscopy results, with higher cellular internalization of ATRAM-BSA-PLGA NPs in MCF-7 cells measured at acidic pH relative to physiological pH (amount of internalized NPs was > 5- and > 7-fold higher at pH 6.5 compared to pH 7.4 at 1 and 4 h incubation, respectively) (Figure 4e,f). As a control, cellular uptake of Dox-TPP loaded BSA-PLGA NPs in MCF-7 cells was assessed using confocal fluorescence microscopy (Figure S12) and flow cytometry (Figure 4e,f). In the absence of the ATRAM peptide, poor uptake of the NPs was observed at both pHs (7.4 and 6.5) and incubation times (1 and 4 h). Thus, the pH-dependent membrane insertion of ATRAM facilitates cellular internalization of the coupled NPs preferentially in cells that reside within a mildly acidic environment.”

Supporting Information; Experimental Section; Intracellular Imaging and Colocalization:

“Cells (MCF-7, HeLa or differentiated THP-1) were seeded at a density of 2×10^5 cells/well in 500 μ L complete medium in 4-chambered 35 mm glass bottom Cellview cell culture dishes (Greiner Bio-One, Monroe, NC). After culturing for 24 h, the medium was replaced fresh medium (pH 7.4 or 6.5) containing 2.5 μ g/mL Dox-TPP loaded PLGA, BSA-PLGA or ATRAM-BSA-PLGA NPs and incubated for 1–4 h. For some experiments, MCF-7 cells were pre-incubated for 1 h at 4 $^{\circ}$ C in serum-free DMEM, pre-treated for 1 h at 37 $^{\circ}$ C with 10 mM sodium azide and 6 mM 2-deoxy-D-glucose in serum- and glucose-free DMEM, or pretreated for 30 min at 37 $^{\circ}$ C with the following drugs in serum-free DMEM: 10 μ M chlorpromazine; 5 mM methyl- β -cyclodextrin; 5 μ M filipin; or 5 μ M amiloride. After addition of the NPs, the cells were maintained for 1 h at 4 $^{\circ}$ C or in presence of inhibitors at 37 $^{\circ}$ C. 30 min prior to imaging, the medium was replaced with fresh medium containing 50 nM MitoTracker Green or vehicle. Finally, immediately prior to imaging, the medium was once again replaced with fresh medium to remove any extracellular markers. Imaging was done on an Olympus Fluoview FV-1000 confocal laser scanning microscope, using a 63 \times Plan-Apo/1.3 NA oil immersion objective with DIC capability. Image processing was done using the Fiji image processing software.(Schindelin et al. 2012)”

Supporting Information; Experimental Section; Quantification of Cellular Uptake:

“Cellular uptake of Dox-TPP loaded ATRAM-BSA-PLGA NPs at pH 7.4 or 6.5 was measured using flow cytometry. Cells (MCF-7 or differentiated THP-1) were seeded at a density of 2×10^4 cells/well in 500 μ L complete medium in 24-well plates. After culturing for 24 h, the cells were washed with PBS at 37 $^{\circ}$ C and the medium was replaced with fresh medium (pH 7.4 or 6.5) containing 2.5 μ g/mL Dox-TPP loaded PLGA, BSA-PLGA or ATRAM-BSA-PLGA NPs and incubated for 1–4 h at 37 $^{\circ}$ C. Subsequently, the cells were washed three times with ice-cold PBS to remove the extracellular NPs, and then treated with trypsin-EDTA for 5 min to detach the cells and remove cell surface-bound peptide. Finally, the cells were centrifuged (1,000 \times g for 5 min at 4 $^{\circ}$ C) and re-suspended in 500 μ L ice-cold PBS with 10% FBS. Data collection (10,000 cells/sample, gated on live cells by forward/side scatter and propidium iodide (PI) exclusion) was done immediately afterwards on a BD FACSAria III cell sorter (BD Biosciences, San Jose, CA), and analysis was performed using the BD FACSDiva software.”

Concern #7. Page 19: What is the release mechanism in the case of Paclitaxel? (From Concern #4b. An explanation for no Rhodamine release vs Paclitaxel release is needed. Also, is loading efficiency of Rhodamine B and Paclitaxel comparable to Dox-TPP?)

Response: The release of Dox-TPP from crosslinked BSA-PLGA NPs was monitored over 24 h using time-dependent fluorescence measurements (Wilhelm *et al.* 2016) under different conditions of pH (5.0 – 7.4) and GSH (0.5–10 mM), which were meant to mimic the conditions the NPs are exposed to upon internalization (by either endocytosis or direct translocation) into cancer cells (**Figure 3b,d**). Based on these experiments, we proposed a mechanism in which low pH (within late endosomes/lysosomes) degrades the PLGA core (as a result of hydrolysis of the ester bonds in the polymer chains) (Rezvantlab *et al.* 2018) and releases Dox-TPP from the NPs. We also suggested that protonation of the drug at low pH leads to its dissociation from the hydrophobic PLGA core and likely contributes to the acid-triggered release. (Liu *et al.* 2018; Palanikumar *et al.* 2017). Control experiments with rhodamine B loaded BSA-PLGA NPs at pHs from 7.4 to 4.0 (where rhodamine B becomes cationic due to protonation of its carboxylic acid group ($pK_a \sim 4.2$)), (Yu *et al.* 2013) supported our hypothesis that protonation of the cargo contributes to its acid-triggered release (**Figure 3c**).

ATRAM-BSA-PLGA NPs loaded with the hydrophobic chemotherapeutic paclitaxel (PTX) (loading capacity was comparable to that of Dox-TPP; **Table S1**), were used as a control to ascertain the generality of the hybrid NPs as a delivery platform for anticancer drugs. In this case, we tested the effect of the PTX-loaded NPs on MCF-7 cell viability at pH 7.4 and 6.5 using the MTS assay. The results showed that, as observed with Dox-TPP-loaded NPs (**Figure 6b**), treatment with PTX-loaded NPs for 48 h led to significantly higher cytotoxicity in MCF-7 cells at pH 6.5 compared to pH 7.4 (**Figure S16**). This confirmed that the ATRAM-BSA-PLGA NPs exhibit selectivity for cancer cells in an acidic environment, such as that of tumors, compared to normal cells under physiological conditions.

Thus, in the case of PTX loaded NPs, we did not measure the release of the drug *per se*, but rather the effect of released drug on cell viability. The release mechanism for PTX from the NPs is again likely to involve acid-triggered release within endocytic compartments due to degradation of the PLGA core. Unlike Dox-TPP and rhodamine B, PTX does not undergo protonation at low pH, so dissociation of PTX likely does not contribute significantly to its acid-triggered release. However, it is important to note that whereas the drug release was measured over 24 h, the duration of the cell viability assays was much longer time (48 h), which is expected to lead far more extensive degradation (hydrolysis) of the PLGA core, leading to significant release of PTX (even in the absence of drug protonation-driven dissociation from the PLGA core).

Supporting Information; Table S1. Drug loading capacity of BSA-PLGA NPs.

Chemotherapeutic Drug	Loading Capacity (wt%)
Doxorubicin-Triphenylphosphonium (Dox-TPP)	1.8
Paclitaxel (PTX)	2.0
Rhodamine B	1.6

The loading capacity of the crosslinked BSA-PLGA NPs was determined using *Equation 2*.

Concern #8. Page 21: The controls used for tumor inhibition, can also be used as control for the studies with different cancer cell lines, particularly Dox loaded BSA-PLGA nanoparticles.

Response: As suggested by the reviewer, we carried out additional control experiments in which we measured the effects of Dox-TPP loaded crosslinked BSA-PLGA NPs on MCF-7 and HeLa cell viability. In the absence of the ATRAM peptide, no significant toxicity was observed with the NPs loaded with up to 2 $\mu\text{g/mL}$ Dox-TPP at either pH 7.4 or 6.5. This confirms that the pH-responsive ATRAM is required for the pH-dependent cytotoxicity of the coupled NPs.

The results of the new control experiments are presented in the new **Figure S15**:

“Figure S15. Cell viability of MCF-7 (a) and HeLa (b) cells treated with Dox-TPP loaded BSA-PLGA NPs for 48 h at pH 7.4 or 6.5. Cell viability was assessed using the MTS assay, with the % viability determined from the ratio of the absorbance of the treated cells to the control cells. ns, non-significant ($P > 0.05$).”

In addressing the reviewer’s concern, the relevant sections of the **Revised Manuscript** and **Revised Supporting Information** have been modified and now read as follows:

Results and Discussion; Cytotoxic Effects of Drug-Loaded ATRAM-BSA-PLGA NPs:

“Multiple human and animal cancer cell lines were treated with Dox-TPP loaded ATRAM-BSA-PLGA NPs at physiological and acidic pH, and cell viability was quantified using the MTS assay. Treatment of human cancer cell lines, MCF-7 (Figure 6b), HeLa (Figure 6c), and pancreatic carcinoma MIA PaCa-2 (Figure 6d), with the Dox-TPP loaded NPs resulted in substantially greater loss of viability at pH 6.5 compared to pH 7.4, particularly at higher Dox-TPP concentrations ($> 0.5 \mu\text{g/mL}$). For example, exposure to ATRAM-BSA-PLGA NPs loaded with 2 $\mu\text{g/mL}$ Dox-TPP at pH 7.4 decreased viability of MCF-7 cells to $65 \pm 3\%$; however, exposure of MCF-7 cells to the same NPs at pH 6.5 resulted in a much lower viability of $8 \pm 2\%$. Likewise, incubation of mouse cancer cell lines, neuroblastoma Neuro-2a (Figure 6e) and breast cancer 4T1 (Figure 6f), with Dox-TPP loaded NPs led to significantly higher toxicity at pH 6.5 compared to 7.4. As a control, we measured the effects of Dox-TPP loaded BSA-PLGA NPs on MCF-7 and HeLa cell viability (Figure S15). No significant toxicity was observed with BSA-PLGA NPs loaded with up to 2 $\mu\text{g/mL}$ Dox-TPP at either pH 7.4 or 6.5. This confirms that the pH-responsive ATRAM is required for the both the pH-dependent uptake (Figure 4d–f) and cytotoxicity of the coupled NPs (Figure 6b–f). Although the Dox-TPP loaded ATRAM-BSA-PLGA NPs induced pH-dependent toxicity in all cell lines tested, the greatest effect was observed in MCF-7 and MIA PaCa-2 cells, suggesting that the pH-responsive hybrid NPs could be particularly effective in targeting human breast and pancreatic cancers.”

Supporting Information; Experimental Section; Cell Viability/Toxicity Assays:

“Cells were seeded at a density of 2×10^4 cells/well in 100 μ L complete medium in standard 96-well plates. After culturing for 24 h, the medium was replaced with fresh medium (pH 7.4 or 6.5) containing 2.5–75 μ g/mL drug-free NPs, or NPs loaded with 0.05–2 μ g/mL Dox-TPP or paclitaxel (PTX), and incubated for 48 h at 37 °C. Thereafter, the medium was replaced with fresh medium, and 20 μ L MTS reagent was added to each well. The MTS reagent was incubated for 4 h at 37 °C, and absorbance of the soluble formazan product ($\lambda = 490$ nm) of MTS reduction was measured on a BioTek Synergy H1MF Multi-Mode Microplate-Reader, with a reference wavelength of 650 nm to subtract background. Wells treated with peptide-free carrier were used as control, and wells with medium alone served as a blank. MTS reduction was determined from the ratio of the absorbance of the treated wells to the control wells.”

REVIEWERS' COMMENTS:

Reviewer #1 (Remarks to the Author):

Palanikumar and colleagues present here a revised version of their manuscript in which they have provided new data as well as substantial changes to the text that answer all previously stated concerns. I believe this now makes a very compelling work that merits publication in Communications Biology, however I have a minor comment that I would strongly encourage the authors to implement.

Concern #5 (first revision). The authors indicate that internalization studies in macrophages were performed and pH=7.4 because these cells were challenging to culture at lower pHs. While I find the justification given by the authors acceptable, I think this should be indicated in the text and discussed as a caveat of the study. Even though most immune cells exposed to the particles will be in circulation, there will also be immune cells in the tumor microenvironment that could encounter these particles at lower pHs.

Reviewer #2 (Remarks to the Author):

The authors have addressed all of my concerns. This is a very nicely done study, and showing the excessive amount of the supporting data is a standard that I wish all labs working on nanoparticles would do.

Reviewer #3 (Remarks to the Author):

The manuscript "pH-Responsive High Stability Polymeric Nanoparticles for Targeted Delivery of Anticancer Therapeutics" from Palanikumar, L., et. al. highlights the development of biocompatible and biodegradable pH-responsive nanoparticles that display low non-specific release of drugs with high selectivity towards cancer cells, high circulation stability, and high applicability for in vivo treatment of tumors, covering all key characteristics of an ideal delivery system. The updated version of the manuscript is significantly improved, answering all the concerns raised previously, and providing stronger evidence of the advantages of these pH-responsive nanoparticles for cancer treatment, with the appropriate controls as comparison. I highly appreciate the efforts of the authors to address fully all the questions and concerns related to the previous version, and in light of all the improvements made, I consider it highly relevant and ready for publication in Communications Biology.

As recommended, we have addressed **Reviewer #1**'s remaining concern:

Concern #5 (first revision). The authors indicate that internalization studies in macrophages were performed and pH=7.4 because these cells were challenging to culture at lower pHs. While I find the justification given by the authors acceptable, I think this should be indicated in the text and discussed as a caveat of the study. Even though most immune cells exposed to the particles will be in circulation, there will also be immune cells in the tumor microenvironment that could encounter these particles at lower pHs.

In addressing **Reviewer #1**'s concern, we have included a discussion on tumor-associated macrophages and the reasons for not being able to perform the experiment suggested by the reviewer (as we had done in responding to the reviewer's concern during the previous round of revisions).

The paragraphs added to the address **Reviewer #1**'s remaining concern are as follows:

Results; Macrophage Recognition and Immunogenicity of ATRAM-BSA-PLGA NPs:

“Opsonization of NPs and their subsequent uptake by monocytes and macrophages of the mononuclear phagocyte system (MPS) leads to accumulation of the NPs in healthy organs (e.g. spleen and liver) rather than at the target solid malignant tumors (tumor-associated macrophages are discussed in Supplementary Note 4).(Blanco, Shen, and Ferrari 2015) To overcome this issue, the surface of NPs is often functionalized with neutral molecules, e.g. PEG, known to resist protein adsorption and MPS clearance.(Rattan et al. 2017) However, studies have reported that surface modification with compounds such as PEG may trigger an immune reaction.(van Witteloostuijn, Pedersen, and Jensen 2016)”

Supplementary Information; Supplementary Note 4:

“Tumor-associated macrophages play an important role in cancer survival, proliferation and metastasis.(Mantovani et al. 2017) To probe the possible interactions of ATRAM-BSA-PLGA NPs with macrophages within the tumor microenvironment, we attempted to measure the cellular uptake and cytotoxicity of the Dox-TPP loaded NPs in THP-1 cells at pH 6.5. Unfortunately, these experiments were largely unsuccessful due to poor growth of the cells at low pH. Examination of the morphology under the microscope revealed extensive cell damage/death. Moreover, the MTS response of these cells showed significant variability between experiments (and even between different control samples/wells in the same experiment), reflecting the uncontrolled effects of low pH on the cells. Thus, we are unable to comment on the potential interactions between the NPs and tumor-associated macrophages. However, since the vast majority of immune cells (including macrophages) encountered will be in circulation (i.e. at physiological pH),(Rattan et al. 2017; Oh et al. 2018) the data presented in Figure 7 represents the most relevant condition for assessing the tumor-targeting capabilities of the ATRAM-BSA-PLGA NPs.”